# Proximal-Based Generative Modeling for Bayesian Inverse Problems

**Boyang Zhang** [1 2]   **Zhiguo Wang** [3]   **Ya-Feng Liu** [4 5]

## Abstract

Score-based diffusion models demonstrate superior performance in generative tasks but encounter fundamental bottlenecks in inverse problems due to the analytical intractability of the time-dependent likelihood score. To bridge this gap, we propose a novel proximal-based generative modeling (PGM) framework that rigorously circumvents explicit likelihood evaluation. Our framework is built upon a theoretical equivalence between Gaussian convolution in diffusion processes and Moreau-Yosida regularization in nonsmooth optimization. This enables a new sampling mechanism driven by the proposed Moreau score, which admits a closed-form expression via proximal operators. Moreover, we introduce Moreau score matching to learn the proximal operators that rely solely on samples drawn from the prior distribution. Theoretically, PGM eliminates the early-stopping bias inherent in the score-based diffusion model and achieves non-asymptotic convergence. Experiments demonstrate that PGM significantly surpasses state-of-the-art methods in reconstruction quality and sampling time.

## 1. Introduction

Inverse problems entail the estimation of an unknown signal of interest, $\boldsymbol{x} \in \mathbb{R}^d$, from observed measurement data, $\boldsymbol{y} \in \mathbb{R}^m$. These problems are ubiquitous in signal processing (Tropp & Wright, 2010; Ma et al., 2018; Hu et al., 2020; Zhang et al., 2023), computational imaging (Jin et al., 2017;

---

[1]School of Advanced Interdisciplinary Sciences, University of Chinese Academy of Sciences, Beijing, China [2]Academy of Mathematics and Systems Science, Chinese Academy of Sciences, Beijing, China [3]School of Mathematics, Sichuan University, Chengdu, China [4]Ministry of Education Key Laboratory of Mathematics and Information Networks, Beijing, China [5]School of Mathematical Sciences, Beijing University of Posts and Telecommunications, Beijing, China. Correspondence to: Ya-Feng Liu <yafengliu@bupt.edu.cn>.

*Proceedings of the 43rd International Conference on Machine Learning*, Seoul, South Korea. PMLR 306, 2026. Copyright 2026 by the author(s).

Bertero et al., 2021; Daras et al., 2024), and communications (Soltani et al., 2019; Yu et al., 2022; Liu et al., 2024). From a Bayesian perspective, solving an inverse problem is equivalent to sampling from a posterior distribution of the composite form (Mou et al., 2022; Renaud et al., 2025):

$$\pi(\boldsymbol{x}) \propto \exp\{-U(\boldsymbol{x})\} \text{ with } U(\boldsymbol{x}) = \beta f_{\boldsymbol{y}}(\boldsymbol{x}) + g(\boldsymbol{x}), \quad (1)$$

where $\beta$ is a hyperparameter, $f_{\boldsymbol{y}}(\boldsymbol{x})$ represents the data fidelity derived from the likelihood $p(\boldsymbol{y}|\boldsymbol{x})$ and $g(\boldsymbol{x})$ represents the prior derived from $p(\boldsymbol{x})$ which may be nonsmooth.

To address the Bayesian inverse problem, two primary methodological paradigms have been developed: optimization-based methods (Haber et al., 2000; Afonso et al., 2010; Ye et al., 2019) and sampling-based methods (Stuart, 2010; Knapik et al., 2011; Ehrhardt et al., 2024; Janati et al., 2025; Habring et al., 2026). Optimization-based methods seek to reconstruct the unknown signal by identifying the most likely one under the posterior distribution, which leads to the maximum a posteriori (MAP) estimator. Although optimization-based methods provide a single point estimate, they do not capture the full structure of the posterior distribution, and thus may fail to reflect uncertainty. In contrast, sampling-based methods aim to generate a collection of samples from the posterior distribution, enabling principled uncertainty quantification.

Sampling from such complex, high-dimensional distributions remains a fundamental challenge (Welling & Teh, 2011; Xu et al., 2018; Song et al., 2020). Traditional Markov chain Monte Carlo (MCMC) methods, such as the unadjusted Langevin algorithm (ULA) (Durmus & Moulines, 2017) and the Metropolis-adjusted Langevin algorithm (MALA) (Roberts & Tweedie, 1996), offer theoretical guarantees for smooth distributions (Ma et al., 2019). While recent advances in proximal (Pereyra, 2016; Salim et al., 2019; Pillai, 2024) and mirror Langevin dynamics (Hsieh et al., 2018; Ahn & Chewi, 2021) have extended these capabilities to nonsmooth settings, they rely on explicit knowledge of the potential functions. In the literature, the traditional approach of using hand-crafted, explicit priors is slowly being replaced by rich, learned priors (Kingma & Welling, 2013; Goodfellow et al., 2014; Ho et al., 2020; Chung et al., 2022a). Despite their strong ability to approximate complex prior distributions, these learned implicit priors make it hard to generate samples from the posterior distribution.

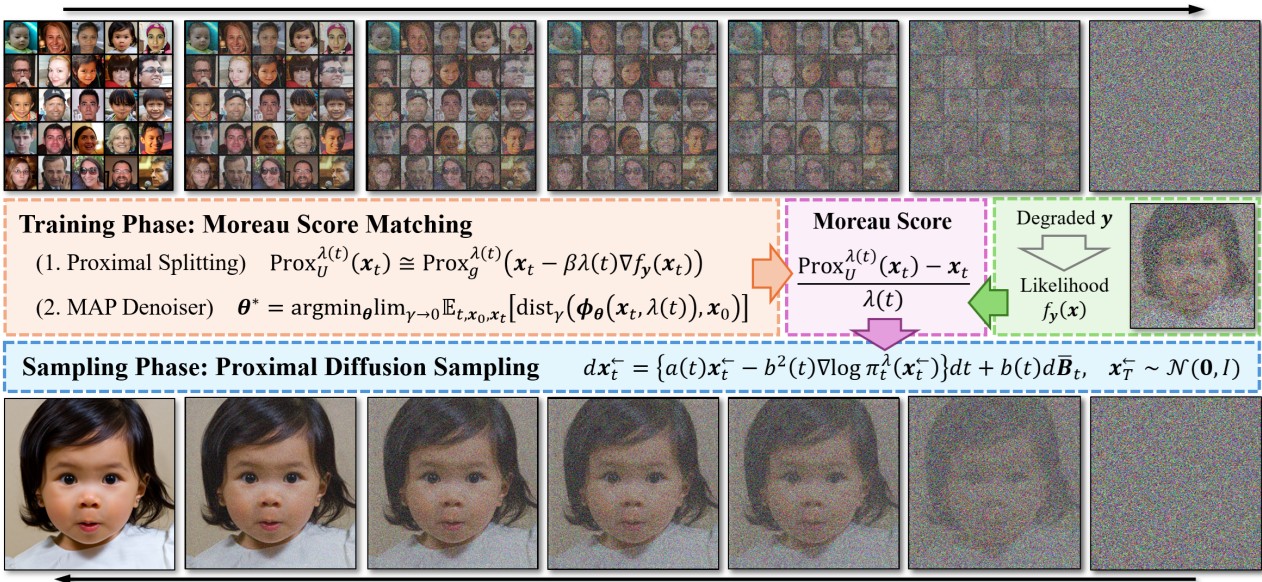

*Figure 1.* **A sketch map for PGM.** Training phase: a proximal splitting is applied to provide an approximation of the Moreau score, where a network is trained to learn the proximal operator in an unsupervised manner. Sampling phase: the traditional score function is replaced by the Moreau score, which admits an explicit, smooth, and asymptotically equivalent formulation via proximal operators.

Recent years have witnessed considerable interest in leveraging diffusion models as structural priors for solving inverse problems (Chung et al., 2022a; Song et al., 2023a; Mardani et al., 2023; Alkhouri et al., 2024; Chang et al., 2025; Zhang et al., 2026; Habring et al., 2026; Yuan et al., 2026). The emergence of score-based diffusion models has marked a paradigm shift, demonstrating the ability to learn generative models directly from data without requiring an explicit analytical form of the target density (Ho et al., 2020; Song et al., 2020). By estimating the score function via denoising score matching and simulating reverse-time stochastic differential equations (SDEs), these methods have achieved state-of-the-art performance in image generation (Dhariwal & Nichol, 2021; Rombach et al., 2022; Song et al., 2023a; Tolooshams et al., 2025). Despite this success, applying diffusion models to Bayesian inverse problems remains challenging in practice, due to the difficulty of accurately estimating the time-dependent posterior score $\nabla \log p_t(\boldsymbol{x}_t|\boldsymbol{y})$ (Meng & Kabashima, 2022; Kawar et al., 2022). As a direct extension, conditional diffusion models explicitly train a score network architecture $\boldsymbol{s}_{\boldsymbol{\theta}}(\boldsymbol{x}_t, t, \boldsymbol{y})$ to learn the conditional score $\nabla_{\boldsymbol{x}} \log p_t(\boldsymbol{x}_t|\boldsymbol{y})$ with paired training data (Saharia et al., 2022a;b).

To enhance the generalizability and reduce training costs, zero-shot methods modify the sampling trajectory to enforce measurement consistency, without requiring any retraining or fine-tuning of the score network parameters. Specifically, by Bayes' rule,

$$\nabla_{\boldsymbol{x}} \log p_t(\boldsymbol{x}_t|\boldsymbol{y}) = \underbrace{\nabla_{\boldsymbol{x}} \log p_t(\boldsymbol{x}_t)}_{\text{prior score}} + \underbrace{\nabla_{\boldsymbol{x}} \log p(\boldsymbol{y}|\boldsymbol{x}_t)}_{\text{log-likelihood}}, \quad (2)$$

where the time-dependent log-likelihood $\nabla_{\boldsymbol{x}} \log p(\boldsymbol{y}|\boldsymbol{x}_t)$ generally lacks a closed-form expression. Numerous studies have focused on approximating this intractable term (Song et al., 2023b; Yu et al., 2023; Li & Wang, 2025). For linear inverse problems, subspace and null-space projection is applied to modify the sampling trajectory by forcing the intermediate sample to directly satisfy the measurement consistency (Lugmayr et al., 2022; Kawar et al., 2022; Wang et al., 2022; Zirvi et al., 2025). For nonlinear inverse problems, a series of approximations on $p(\boldsymbol{x}_0|\boldsymbol{x}_t)$ is applied to obtain the time-dependent log-likelihood (Chung et al., 2022a; Song et al., 2023b; Boys et al., 2023; Wu et al., 2024a; Xu et al., 2025). However, these approximations require expensive iterative computations during sampling and lack theoretical convergence guarantees.

For Bayesian inverse problems, the primary objective is to efficiently and accurately sample from the posterior distribution $\pi(\boldsymbol{x})$. This goal does not necessarily require using the time-dependent posterior score in the reverse-time SDE. A natural and fundamental question therefore arises:

*Can we construct an alternative score function that replaces the time-dependent posterior score while still guaranteeing correct sampling from the posterior distribution?*

Motivated by proximal-based sampling methods, we develop a new generative modeling paradigm, a novel proximal-based generative modeling (PGM) framework, that bypasses the computation of the intractable time-dependent likelihood term.

Our contributions are threefold:

- **A novel framework of bridging optimization and diffusion.** We establish a rigorous theoretical connection between the posterior score and the proposed Moreau score. Leveraging this relationship, we introduce a proximal-based generative modeling and derive a new reverse-time SDE driven explicitly by the Moreau score, which admits a closed-form expression via proximal operators and can be evaluated efficiently.

- **Unsupervised Moreau score matching.** We introduce Moreau score matching, a training objective that learns the Moreau score for unknown priors in a completely unsupervised manner. Once the proximal operator is learned, it can be used as a pre-trained operator for any sampling processes or further fine-tuning. This allows us to perform posterior sampling via proximal diffusion sampling, unifying score-based generative modeling with proximal Langevin dynamics.

- **Improved theoretical and experimental results.** We extend classical convergence results for diffusion models to the context of solving inverse problems. Notably, our convergence guarantees are stronger, as they successfully eliminate the early-stopping error based on the properties of the Moreau approximation. Extensive experiments demonstrate that PGM consistently outperforms state-of-the-art methods in generation quality, memory consumption, and sampling time.

## 2. Background

### 2.1. Score-Based Diffusion Models

We begin with a brief review of score-based diffusion models (Song et al., 2020). The forward process progressively perturbs data into noise via Gaussian kernels, a transformation described by the SDE of the following form:

$$d\boldsymbol{x}_t^{\rightarrow} = a(t)\boldsymbol{x}_t^{\rightarrow}dt + b(t)d\boldsymbol{B}_t, \qquad (3)$$

where $\boldsymbol{B}_t$ denotes a standard Wiener process. The goal of diffusion models is to learn the corresponding reverse SDE, which takes the form

$$d\boldsymbol{x}_t^{\leftarrow} = \left\{a(t)\boldsymbol{x}_t^{\leftarrow} - b^2(t)\nabla\log p_t(\boldsymbol{x}_t^{\leftarrow})\right\}dt + b(t)d\bar{\boldsymbol{B}}_t, \qquad (4)$$

where $d\bar{\boldsymbol{B}}_t$ is a reverse-time Wiener process and $\nabla\log p_t(\boldsymbol{x}_t^{\leftarrow})$ is the score function. In practice, the score is approximated by a neural network $\boldsymbol{s}_{\boldsymbol{\theta}}(\boldsymbol{x}, t)$ with parameters $\boldsymbol{\theta}$, trained via denoising score matching (Vincent, 2011):

$$\boldsymbol{\theta}^* = \arg\min_{\boldsymbol{\theta}} \mathbb{E}\left[\|\boldsymbol{s}_{\boldsymbol{\theta}}(\boldsymbol{x}_t, t) - \nabla\log p_t(\boldsymbol{x}_t|\boldsymbol{x}_0)\|_2^2\right], \quad (5)$$

where the expectation is taken over $t \sim \mathcal{U}[0, T]$, $\boldsymbol{x}_t \sim p(\boldsymbol{x}_t|\boldsymbol{x}_0)$, and $\boldsymbol{x}_0 \sim p_0(\boldsymbol{x}_0)$. Given a trained score model, the reverse-time SDE is simulated with standard numerical solvers (e.g., Euler–Maruyama) to generate samples.

### 2.2. Diffusion Models for Inverse Problems

Consider a given measurement $\boldsymbol{y} \in \mathbb{R}^m$ of the form

$$\boldsymbol{y} = \mathcal{A}(\boldsymbol{x}) + \boldsymbol{\xi}, \qquad (6)$$

where $\mathcal{A} : \mathbb{R}^d \to \mathbb{R}^m$ denotes a forward measurement operator, $\boldsymbol{x} \in \mathbb{R}^d$ is an unknown signal, and $\boldsymbol{\xi} \sim \mathcal{N}(\boldsymbol{0}, \sigma_\xi^2 I)$ is the additive Gaussian noise.

If $\mathcal{A}$ is a linear operator, it can be represented by a measurement matrix $A \in \mathbb{R}^{m\times d}$. The forward model becomes $\boldsymbol{y} = A\boldsymbol{x} + \boldsymbol{\xi}$. If $\mathcal{A}$ is a nonlinear function, the Jacobian $\nabla\mathcal{A}(\boldsymbol{x})$ depends on the state $\boldsymbol{x}$. In nearly all practical scenarios, $m \ll d$ (fewer measurements than pixels) or $\mathcal{A}$ has a nontrivial null-space. The problem is severely ill-posed, meaning the pseudo-inverse solution is either nonunique or wildly unstable to noise. Therefore, a prior is usually needed to constrain the solution.

Diffusion models can be adapted to solve such inverse problems by replacing the unconditional score in (4) with the posterior score $\nabla\log p_t(\boldsymbol{x}_t|\boldsymbol{y})$. For notational convenience, we denote $\pi_t(\boldsymbol{x}_t) := p_t(\boldsymbol{x}_t|\boldsymbol{y})$ as the time-dependent conditional distribution. As shown in (2), the posterior score $\nabla\log \pi_t(\boldsymbol{x}_t)$ decomposes into a time-dependent prior score $\nabla\log p_t(\boldsymbol{x}_t)$ that can be efficiently estimated via denoising score matching (5) and a time-dependent likelihood term

$$\nabla_{\boldsymbol{x}}\log p(\boldsymbol{y}|\boldsymbol{x}_t) = \nabla_{\boldsymbol{x}}\log\int_{\mathbb{R}^d} p(\boldsymbol{y}|\boldsymbol{x}_0)p(\boldsymbol{x}_0|\boldsymbol{x}_t)d\boldsymbol{x}_0, \quad (7)$$

whose evaluation constitutes the main technical challenge.

Numerous studies have sought to approximate this intractable time-dependent likelihood term (7). For Gaussian linear inverse problems, diffusion posterior sampling (DPS) (Chung et al., 2022a) adopts the likelihood approximation $p(\boldsymbol{y}|\boldsymbol{x}_t) \approx \mathcal{N}(A\hat{\boldsymbol{x}}_0(\boldsymbol{x}_t), \sigma_\xi^2 I)$, where $\hat{\boldsymbol{x}}_0(\boldsymbol{x}_t)$ denotes a posterior mean estimator. However, this Dirac delta approximation of $p(\boldsymbol{x}_0|\boldsymbol{x}_t)$ fundamentally biases the sampling distribution, pulling it away from the true Bayesian posterior. To obtain a better approximation, pseudoinverse-guided diffusion models (ΠGDM) (Song et al., 2023b) and Tweedie moment projected diffusion (TMPD) (Boys et al., 2023) leverage higher order information and obtain statistically principled approximations. This term acts as a force vector, gently pushing the unconditional SDE updates toward the measurement manifold based on the residual error.

However, these DPS-family approaches require backpropagation through a pre-trained model, making it computationally demanding and memory intensive. Numerous more advanced techniques such as plug-and-play methods, have since emerged, which do not require backpropagation (Wang et al., 2022; Zhu et al., 2023; Mardani et al., 2023; Wu et al., 2024a; Dou et al., 2025; Zhang et al., 2025a). A detailed discussion on recent works for solving inverse problems with diffusion models is provided in Appendix A.

## 2.3. Proximal Unadjusted Langevin Algorithm

Sampling algorithms play an increasingly important role in tackling Bayesian inverse problems (1). Among them, proximal methods provide a principled way to handle nonsmooth posterior potentials. In this subsection, we review the proximal unadjusted Langevin algorithm (P-ULA) and its connection to Moreau–Yosida regularization.

As the potential in (1) has a nonsmooth term $g$, we exploit smoothing techniques from optimization theory to tackle this challenge. For a proper, closed, convex function $g : \mathbb{R}^d \to \mathbb{R} \cup \{+\infty\}$, its proximal operator is defined as:

$$\text{Prox}_g^\lambda(\boldsymbol{x}) = \underset{\boldsymbol{u} \in \mathbb{R}^d}{\arg\min} \left\{ g(\boldsymbol{u}) + \frac{1}{2\lambda} \|\boldsymbol{u} - \boldsymbol{x}\|^2 \right\}, \quad (8)$$

where $\lambda > 0$ is a proximal parameter. The Moreau-Yosida regularization (also known as Moreau envelope) (Durmus et al., 2022) of $g$ is defined as:

$$g^\lambda(\boldsymbol{x}) = \underset{\boldsymbol{u} \in \mathbb{R}^d}{\min} \left\{ g(\boldsymbol{u}) + \frac{1}{2\lambda} \|\boldsymbol{u} - \boldsymbol{x}\|^2 \right\}. \quad (9)$$

It can be shown that the gradient of $g^\lambda$ is connected to the proximal operator via the relation

$$\nabla g^\lambda(\boldsymbol{x}) = \lambda^{-1}(\boldsymbol{x} - \text{Prox}_g^\lambda(\boldsymbol{x})). \quad (10)$$

This relation enables the optimization of nonsmooth functions with gradient-based methods, provided access to their proximal operator. Further properties of the Moreau envelope are detailed in Appendix C.

To handle a potentially nonsmooth term $g$, one can consider its Moreau envelope $g^\lambda$ and the surrogate distribution

$$p_0^\lambda(\boldsymbol{x}) \propto \underset{\boldsymbol{u} \in \mathbb{R}^d}{\sup} \left\{ p_0(\boldsymbol{u}) \exp \left\{ -\frac{\|\boldsymbol{u} - \boldsymbol{x}\|^2}{2\lambda} \right\} \right\},$$

which is smooth for any $\lambda > 0$ and converges to the original target $p_0$ as $\lambda \to 0$. A standard sampling method for such smooth surrogates is the P-ULA (Pereyra, 2016), which iterates as follows:

$$\boldsymbol{x}_{k+1} = \boldsymbol{x}_k + \delta \nabla \log p_0^\lambda(\boldsymbol{x}_k) + \sqrt{2\delta} \, \boldsymbol{\xi}_k, \quad (11)$$

where $\delta > 0$ is the stepsize and $\boldsymbol{\xi}_k \sim \mathcal{N}(\boldsymbol{0}, I)$.

Although this approach provides a principled mechanism for sampling from the target distribution $\pi(\boldsymbol{x})$, it requires evaluating $\nabla \log p_0^\lambda$, i.e., explicit access to the target density. In many practical applications, however, one only has access to samples from the target distribution of P-ULA. Notably, from the perspective of score-based modeling, (11) can be viewed as employing an alternative Moreau score function $\nabla \log p_0^\lambda(\boldsymbol{x}_k)$ to replace the original score $\nabla \log p_0(\boldsymbol{x}_k)$.

# 3. Proximal-Based Generative Modeling

In this section, we introduce a novel proximal-based generative modeling framework for inverse problems. Our core idea is to substitute the traditional score function in the reverse-time SDE with the Moreau score, thereby admitting an explicit, smooth, and asymptotically equivalent formulation.

## 3.1. Motivation: Moreau–Yosida–Gaussian Equivalence

To motivate our approach, we start with the VE-SDE forward process defined in (3), which yields

$$\boldsymbol{x}_t = \boldsymbol{x}_0 + \sqrt{\lambda(t)} \, \boldsymbol{\xi}_t, \quad \boldsymbol{\xi}_t \sim \mathcal{N}(\boldsymbol{0}, I), \quad (12)$$

where $\boldsymbol{x}_0 \sim \pi$ and $\pi$ is defined in (1). The corresponding marginal density is

$$\pi_t(\boldsymbol{x}_t) \propto \int_{\mathbb{R}^d} \pi(\boldsymbol{x}_0) \exp \left\{ -\frac{\|\boldsymbol{x}_0 - \boldsymbol{x}_t\|^2}{2\lambda(t)} \right\} d\boldsymbol{x}_0, \quad (13)$$

which means a Gaussian convolution of the target density.

In parallel, we define the Moreau approximation of $\pi$ as

$$\pi_t^\lambda(\boldsymbol{x}_t) \propto \underset{\boldsymbol{x}_0 \in \mathbb{R}^d}{\sup} \left\{ \pi(\boldsymbol{x}_0) \exp \left\{ -\frac{\|\boldsymbol{x}_0 - \boldsymbol{x}_t\|^2}{2\lambda(t)} \right\} \right\}. \quad (14)$$

The following lemma demonstrates that, for quadratic potentials, these two constructions are equivalent.

**Lemma 3.1.** *Let* $\pi(\boldsymbol{x}) \propto \exp\{-U(\boldsymbol{x})\}$, *where* $U$ *is a positive semidefinite quadratic function. Then for any* $\lambda(t) > 0$, *we have*

$$\pi_t^\lambda(\boldsymbol{x}_t) \propto \pi_t(\boldsymbol{x}_t). \quad (15)$$

Lemma 3.1 provides an optimization perspective on Gaussian noising: in the quadratic case, adding Gaussian noise (convolution) is equivalent, up to a normalization constant, to applying Moreau–Yosida regularization.

Inspired by this equivalence, we replace the traditional Stein score $\nabla \log \pi_t(\boldsymbol{x}_t)$ in the reverse-time dynamics with the Moreau score $\nabla \log \pi_t^\lambda(\boldsymbol{x}_t)$. This yields the reverse SDE:

$$d\boldsymbol{x}_t^\leftarrow = \left\{ -b^2(t) \nabla \log \pi_t^\lambda(\boldsymbol{x}_t^\leftarrow) \right\} dt + b(t) \, d\bar{\boldsymbol{B}}_t. \quad (16)$$

Using (1) and (10), the Moreau score admits the closed form

$$\nabla \log \pi_t^\lambda(\boldsymbol{x}_t) = \frac{\text{Prox}_U^{\lambda(t)}(\boldsymbol{x}_t) - \boldsymbol{x}_t}{\lambda(t)}, \quad (17)$$

where the proximal operator is defined by

$$\begin{aligned} \text{Prox}_U^{\lambda(t)}(\boldsymbol{x}_t) &= \underset{\boldsymbol{x}_0 \in \mathbb{R}^d}{\arg\min} \left\{ U(\boldsymbol{x}_0) + \frac{\|\boldsymbol{x}_0 - \boldsymbol{x}_t\|^2}{2\lambda(t)} \right\} \\ &= \underset{\boldsymbol{x}_0 \in \mathbb{R}^d}{\arg\min} \left\{ \beta f_{\boldsymbol{y}}(\boldsymbol{x}_0) + g(\boldsymbol{x}_0) + \frac{\|\boldsymbol{x}_0 - \boldsymbol{x}_t\|^2}{2\lambda(t)} \right\}. \end{aligned} \quad (18)$$

This constructs an alternative score function that replaces the time-dependent posterior score while still guaranteeing correct sampling from the posterior distribution. This provides the key intuition behind our approach and establishes a direct connection to score-based diffusion models.

Under the inverse model (6), we have $f_{\boldsymbol{y}}(\boldsymbol{x}) = -\log p(\boldsymbol{y}|\boldsymbol{x})$ and $g(\boldsymbol{x}) = -\log p(\boldsymbol{x})$. Then the objective in (18) can be reformulated as

$$\begin{aligned}
\text{Prox}_U^{\lambda(t)}(\boldsymbol{x}_t) &= \arg\max_{\boldsymbol{x}_0 \in \mathbb{R}^d} \{\log p(\boldsymbol{y}|\boldsymbol{x}_0) + \log p(\boldsymbol{x}_0|\boldsymbol{x}_t)\} \\
&= \arg\max_{\boldsymbol{x}_0 \in \mathbb{R}^d} \log p(\boldsymbol{x}_0|\boldsymbol{x}_t, \boldsymbol{y}),
\end{aligned} \quad (19)$$

where the last equality follows from the conditional independence of $\boldsymbol{y}$ and $\boldsymbol{x}_t$ given $\boldsymbol{x}_0$. Substituting (19) into (17) reveals a key advantage: in contrast to the conditional score defined in (2), the proposed Moreau score avoids evaluating the time-dependent likelihood term $\log p(\boldsymbol{y}|\boldsymbol{x}_t)$.

We now quantify the discrepancy between the traditional score $\nabla \log \pi_t(\boldsymbol{x}_t)$ and the Moreau score $\nabla \log \pi_t^\lambda(\boldsymbol{x}_t)$.

**Proposition 3.2.** Let $\boldsymbol{x}_t$ be generated by (12), and let $\boldsymbol{x}_0$ and $\boldsymbol{y}$ satisfy the inverse model (6).

(1) If the posterior $p(\boldsymbol{x}_0|\boldsymbol{x}_t, \boldsymbol{y})$ is symmetric unimodal, then

$$\nabla \log \pi_t(\boldsymbol{x}_t) = \nabla \log \pi_t^\lambda(\boldsymbol{x}_t).$$

(2) If $f_{\boldsymbol{y}}$ is convex, then

$$\left\| \nabla \log \pi_t(\boldsymbol{x}_t) - \nabla \log \pi_t^\lambda(\boldsymbol{x}_t) \right\| \leq \sqrt{\frac{2d}{\lambda(t)}}.$$

In particular, if $f_{\boldsymbol{y}}$ is $m$-strongly convex, then

$$\left\| \nabla \log \pi_t(\boldsymbol{x}_t) - \nabla \log \pi_t^\lambda(\boldsymbol{x}_t) \right\| \leq \sqrt{\frac{2d}{\beta m \lambda^2(t) + \lambda(t)}}.$$

Further, the discrepancy vanishes as $\beta \to \infty$.

Proposition 3.2 states that the Moreau score yields a computationally tractable and well controlled approximation to the traditional score. A generalized discussion of the Moreau-Yosida-Gaussian equivalence and the properties of the Moreau score is provided in Appendix D.

### 3.2. Moreau Score Matching

To sample from the target distribution $\pi$ via the reverse SDE (16), the Moreau score (17) is required. In this subsection, we introduce a Moreau score matching procedure to compute this quantity.

Recall from (1) that $U(\boldsymbol{x}) = \beta f_{\boldsymbol{y}}(\boldsymbol{x}) + g(\boldsymbol{x})$, where $f_{\boldsymbol{y}}(\boldsymbol{x})$ is derived from the likelihood and is assumed known for any given measurement $\boldsymbol{y}$. A standard proximal splitting for $\text{Prox}_U^{\lambda(t)}(\boldsymbol{x}_t)$ gives (Condat et al., 2023):

$$\text{Prox}_U^{\lambda(t)}(\boldsymbol{x}_t) \approx \text{Prox}_g^{\lambda(t)}(\boldsymbol{x}_t - \beta\lambda(t)\nabla f_{\boldsymbol{y}}(\boldsymbol{x}_t)). \quad (20)$$

Since $g$ is unknown, the proximal operator $\text{Prox}_g^{\lambda(t)}$ is unavailable, which forces us to learn this operator from data in an unsupervised mechanism. Notably, this only requires samples generated from the prior $p(\boldsymbol{x}_0) \propto \exp\{-g(\boldsymbol{x}_0)\}$, rather than the posterior $\pi$. A detailed characterization of the proximal splitting is provided in Appendix E.2.

We now turn to the computation of the proximal operator appearing in the Moreau score (17). As shown in (12), the transition kernel is

$$p(\boldsymbol{x}_t|\boldsymbol{x}_0) \propto \exp\left\{-\frac{\|\boldsymbol{x}_0 - \boldsymbol{x}_t\|^2}{2\lambda(t)}\right\}. \quad (21)$$

Substituting (21) and prior $p(\boldsymbol{x}_0)$ into the definition of the proximal operator in (8) yields

$$\text{Prox}_g^{\lambda(t)}(\boldsymbol{x}_t) = \arg\max_{\boldsymbol{x}_0 \in \mathbb{R}^d} p(\boldsymbol{x}_0|\boldsymbol{x}_t). \quad (22)$$

This implies that the proximal operator is equivalent to the maximum a posteriori (MAP) estimator of $\boldsymbol{x}_0$ given Gaussian corrupted observation $\boldsymbol{x}_t$ (Fang et al., 2023).

From the viewpoint of Bayesian decision theory, the MAP estimator is also obtained by minimizing the Bayes risk under sharply concentrated loss functions. Concretely, let $\boldsymbol{\phi}_{\boldsymbol{\theta}}(\boldsymbol{x}_t, \lambda(t))$, which is parameterized by $\boldsymbol{\theta}$, be an estimator of $\boldsymbol{x}_0$. For any $\zeta > 0$, we measure the discrepancy between $\boldsymbol{x}_0$ and its estimator using the loss function:

$$\text{dist}_\zeta(\boldsymbol{x}, \boldsymbol{x}') = 1 - \mathcal{N}(\boldsymbol{x} - \boldsymbol{x}'; \boldsymbol{0}, \zeta^2 I).$$

We proceed under the following regularity conditions:

**Assumption 3.3** (Properties of potential)**.** The function $f_{\boldsymbol{y}}$ is proper, closed, $L$-smooth, and $m$-strongly convex with $m \geq 0$. The function $g$ is convex with compact domain $\mathcal{X}$.

We now present the optimality result, identifying the MAP estimator as the minimizer of a specific Bayes risk.

**Proposition 3.4.** Under Assumption 3.3, let $\boldsymbol{x}_t$ be generated by (12). Consider the parameter $\boldsymbol{\theta}^*$ that minimizes the limiting Bayes risk:

$$\boldsymbol{\theta}^* = \arg\min_{\boldsymbol{\theta}} \lim_{\zeta \to 0} \mathbb{E}\left[\text{dist}_\zeta\left(\boldsymbol{\phi}_{\boldsymbol{\theta}}(\boldsymbol{x}_t, \lambda(t)), \boldsymbol{x}_0\right)\right], \quad (23)$$

where the expectation is taken over $t \sim \mathcal{U}[0, T]$, $\boldsymbol{x}_0 \sim \exp\{-g(\boldsymbol{x}_0)\}$, and $\boldsymbol{x}_t \sim p(\boldsymbol{x}_t|\boldsymbol{x}_0)$. Then, we have

$$\boldsymbol{\phi}_{\boldsymbol{\theta}^*}(\boldsymbol{x}_t, \lambda(t)) = \arg\max_{\boldsymbol{x}_0 \in \mathbb{R}^d} p(\boldsymbol{x}_0|\boldsymbol{x}_t) = \text{Prox}_g^{\lambda(t)}(\boldsymbol{x}_t).$$

A detailed proof of Proposition 3.4 is provided in Appendix E. Notably, Proposition 3.4 indicates that the proximal operator $\text{Prox}_g^{\lambda(t)}(\boldsymbol{x}_t)$ can be approximated with $\boldsymbol{\phi}_{\boldsymbol{\theta}^*}(\boldsymbol{x}_t, \lambda(t))$ by solving the optimization problem (23).

Substituting this result back into the splitting (20), we obtain an explicit, tractable formulation for the Moreau score:

$$\nabla \log \pi_t^{\lambda}(\boldsymbol{x}_t) \approx \frac{\boldsymbol{\phi}_{\boldsymbol{\theta}^*}\left(\boldsymbol{x}_t - \beta \lambda(t) \nabla f_{\boldsymbol{y}}(\boldsymbol{x}_t), \lambda(t)\right) - \boldsymbol{x}_t}{\lambda(t)}.$$

This enables the execution of reverse dynamics (16) to effectively generate samples from the target posterior $\pi(\boldsymbol{x}) \propto \exp\{-U(\boldsymbol{x})\}$. After training, the learned proximal operator $\boldsymbol{\phi}_{\boldsymbol{\theta}^*}(\boldsymbol{x}_t, \lambda(t))$ can be used as a pre-trained operator for any sampling processes or further fine-tuning, following the zero-shot posterior sampling paradigm like score-based diffusion models. Further, it can also serve as a plug-and-play prior, replacing classical mathematical denoisers in optimization and sampling literature.

In Section 4, we will discuss the error between the traditional score and the Moreau score.

### 3.3. Generalized Proximal Diffusion Sampling

The above Moreau score (14) is based on the VE-SDE forward process (12). To sample from (1), we extend the Moreau score to a general forward process (3), and then provide the proximal diffusion sampling process. Consider

$$\boldsymbol{x}_t = \mu(t)\boldsymbol{x}_0 + \sigma(t)\boldsymbol{\xi}_t, \quad \boldsymbol{\xi}_t \sim \mathcal{N}(\boldsymbol{0}, I), \qquad (24)$$

where

$$\mu(t) = \exp\left\{\int_0^t a(s)ds\right\},$$

$$\sigma^2(t) = \int_0^t b^2(s) \exp\left\{2\int_s^t a(r)dr\right\}ds.$$

We consider the reverse-time model

$$d\boldsymbol{x}_t^{\leftarrow} = \left\{a(t)\boldsymbol{x}_t^{\leftarrow} - b^2(t)\nabla \log \pi_t^{\lambda}(\boldsymbol{x}_t^{\leftarrow})\right\}dt + b(t)d\bar{\boldsymbol{B}}_t, \tag{25}$$

where the generalized surrogate distribution is defined by

$$\pi_t^{\lambda}(\boldsymbol{x}_t) \propto \sup_{\boldsymbol{x}_0 \in \mathbb{R}^d}\left\{\pi(\boldsymbol{x}_0)\exp\left\{-\frac{\|\boldsymbol{x}_0 - \boldsymbol{x}_t/\mu(t)\|^2}{2\lambda(t)}\right\}\right\},$$

and the time-varying parameter

$$\lambda(t) = \sigma^2(t)/\mu^2(t) = \int_0^t b^2(s)/\mu^2(s)ds.$$

Then the generalized Moreau score function is given by

$$\nabla \log \pi_t^{\lambda}(\boldsymbol{x}_t) = \frac{\mu(t)\text{Prox}_U^{\lambda(t)}(\boldsymbol{x}_t/\mu(t)) - \boldsymbol{x}_t}{\sigma^2(t)}. \tag{26}$$

---

**Algorithm 1** Proximal-Based Generative Modeling

**Require:** Realizations $\{\boldsymbol{x}^{(i)}\}_{i=1}^{N_s} \sim \exp\{-g\}$, objective $f_{\boldsymbol{y}}$, inverse temperature $\beta$, proximal schedule $\lambda(t)$, diffusion schedule $\mu(t)$, learning schedule $\{\zeta_{\text{Epoch}}\}$.
**Ensure:** Sample $\boldsymbol{x} \sim \exp\{-[\beta f_{\boldsymbol{y}}(\boldsymbol{x}) + g(\boldsymbol{x})]\}$.
1: **(Training Phase: Moreau Score Matching)**
2: Initialize proximal network $\boldsymbol{\phi}_{\boldsymbol{\theta}}(\boldsymbol{x}, \lambda)$.
3: **for** Epoch = 1 to MaxEpoch **do**
4:     Sample $\boldsymbol{x}_0 \sim \exp\{-g(\boldsymbol{x})\}$, $t \in \mathcal{U}[0, T]$.
5:     Generate $\boldsymbol{x}_t|\boldsymbol{x}_0 \sim \mathcal{N}(\boldsymbol{0}, \lambda(t)I)$.
6:     Take stochastic gradient descent on (23).
7: **end for**
8: **(Sampling Phase: Proximal Diffusion Sampling)**
9: Initialize $\bar{\boldsymbol{x}}_0 \sim p_T$.
10: **for** $k = 0$ to $K$ **do**
11:     Calculate $\boldsymbol{P}_k$ with (28) and coefficients with (29).
12:     Update $\bar{\boldsymbol{x}}_{k+1}$ with (27).
13: **end for**
14: **Return** Sample $\boldsymbol{x} = \bar{\boldsymbol{x}}_{K+1}$.

---

Compared with the original Moreau score in (17), the generalized expression in (26) involves a rescaling of the proximal input by $1/\mu(t)$ and substitutes the denominator with $\sigma^2(t)$.

Following the splitting (20) and Proposition 3.4, we use a proximal network $\boldsymbol{\phi}_{\boldsymbol{\theta}}(\boldsymbol{x}_t, \lambda(t))$ to approximate the real proximal $\text{Prox}_g^{\lambda(t)}$. The network parameter $\boldsymbol{\theta}$ is learned by solving the optimization problem (23) via stochastic gradient descent, using samples drawn from $t \sim \mathcal{U}[0, T]$, $\boldsymbol{x}_0 \sim \exp\{-g(\boldsymbol{x}_0)\}$, and $\boldsymbol{x}_t|\boldsymbol{x}_0 \sim \mathcal{N}(\boldsymbol{0}, \lambda(t)I)$.

Substituting (26) into (25), the corresponding reverse process is given by (RP) at the top of the next page. To numerically simulate this process, we employ an exponential interpolation discretization (Zhang & Chen, 2022), which leads to the following proximal diffusion sampling iteration:

$$\bar{\boldsymbol{x}}_{k+1} = \alpha_{1,k}\bar{\boldsymbol{x}}_k + \alpha_{2,k}\boldsymbol{P}_k + \alpha_{3,k}\boldsymbol{\xi}_k, \tag{27}$$

where $\boldsymbol{\xi}_k \sim \mathcal{N}(\boldsymbol{0}, I)$, $\tau_k = T - t_k$,

$$\boldsymbol{P}_k = \boldsymbol{\phi}_{\boldsymbol{\theta}^*}\left(\frac{\bar{\boldsymbol{x}}_k}{\mu(\tau_k)} - \beta \lambda(\tau_k)\nabla f\left(\frac{\bar{\boldsymbol{x}}_k}{\mu(\tau_k)}\right), \lambda(\tau_k)\right), \tag{28}$$

and the coefficients

$$\alpha_{1,k} = \frac{\lambda(\tau_{k+1})\mu(\tau_{k+1})}{\lambda(\tau_k)\mu(\tau_k)}, \alpha_{2,k} = \mu(\tau_{k+1})\left(1 - \frac{\lambda(\tau_{k+1})}{\lambda(\tau_k)}\right)$$

$$\alpha_{3,k} = \mu(\tau_{k+1})\sqrt{\lambda(\tau_{k+1})}\sqrt{1 - \frac{\lambda(\tau_{k+1})}{\lambda(\tau_k)}}. \tag{29}$$

A detailed analysis of the sampling process and discretization error is provided in Appendix G. Finally, the complete PGM framework is summarized in Algorithm 1.

$$dx_t^{\leftarrow} = \left\{ \left[ a(t) + \frac{b^2(t)}{\sigma^2(t)} \right] x_t^{\leftarrow} - \frac{b^2(t)\mu(t)}{\sigma^2(t)} \phi_{\boldsymbol{\theta}^*} \left( \frac{x_t^{\leftarrow}}{\mu(t)} - \beta\lambda(t)\nabla f_{\boldsymbol{y}} \left( \frac{x_t^{\leftarrow}}{\mu(t)} \right), \lambda(t) \right) \right\} dt + b(t)d\bar{\boldsymbol{B}}_t. \tag{RP}$$

## 4. Theoretical Results

In this section, we present quantitative bound on the Wasserstein-1 distance between the target distribution and the generated distribution induced by the proposed PGM. We begin by stating our main assumptions.

**Assumption 4.1.** We assume the following conditions hold:

1. (Bounded schedule) The diffusion schedule $\mu(t) \in [0,1]$ is nonincreasing and $m_\mu \leq |\nabla\log\mu(t)| \leq M_\mu \leq 1/2$. Moreover, $\lambda(t) \in [0, \bar{\lambda}]$ is nondecreasing and $1 \leq m_\lambda \leq |\nabla\log\lambda(t)| \leq M_\lambda$.

2. (Bounded stepsize) $\delta := \sup_k \{t_{k+1} - t_k\} < \infty$.

The first condition controls the growth of the coefficients in (29) and is satisfied by practical SDEs (e.g., VE-SDE and VP-SDE). The second condition provides a uniform upper bound on the stepsize and the relation $T \leq K\delta$, which helps control the discretization error. These assumptions are standard for analyzing the convergence of diffusion models (Lee et al., 2022; Li et al., 2024a; Xu et al., 2026).

Next, we first bound the discrepancy between the approximated Moreau score and the true score function.

**Proposition 4.2.** *Under Assumptions 3.3, consider the diffusion process* (3). *Denote*

$$\boldsymbol{P}_t = \phi_{\boldsymbol{\theta}^*} \left( \frac{x_t}{\mu(t)} - \beta\lambda(t)\nabla f_{\boldsymbol{y}}(\frac{x_t}{\mu(t)}), \lambda(t) \right). \tag{30}$$

*Then there exists $M \geq 0$ such that*

$$\left\| \nabla\log\pi_t(x_t) - \frac{\mu(t)\boldsymbol{P}_t - x_t}{\sigma^2(t)} \right\| \leq \frac{M(1 + \|x_t\|)}{\sigma^2(t)}. \tag{31}$$

Proposition 4.2 provides an upper bound on the score estimation error, which consists of two components: the proximal splitting error and the gap between the true score and the Moreau score. Both of them can be bounded with standard optimization techniques and the discrepancy from Proposition 3.2, respectively.

Let $\text{Law}(\bar{x}_k) = \pi_\infty R_k$ denote the distribution of $\bar{x}_k$ in (27), where $\bar{x}_0 \sim \pi_\infty := \lim_{t\to\infty} \text{Law}(x_t^{\leftarrow})$ and $R_k$ is the transition kernel associated with $p(\bar{x}_k|\bar{x}_0)$. Define the diameter of the constraint set as $\text{diam}(\mathcal{X}) = \sup\{\|x - x'\| : x, x' \in \mathcal{X}\}$. We then present the following non-asymptotic convergence result.

**Theorem 4.3.** Under Assumption 3.3 and 4.1, for sufficiently large $T$, there exist $D_1, D_2, D_3$ such that

$$\mathcal{W}_1\left(\text{Law}(\bar{x}_K), \pi\right) \leq D_1 \exp\left[-m_\mu T\right] + D_2 M + D_3 \delta^{1/2},$$

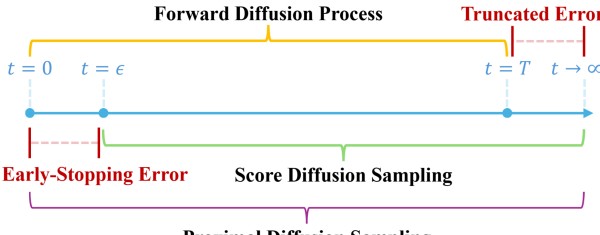

*Figure 2.* Sampling error decomposition.

where

$$D_1 = W_e(\sqrt{d} + \text{diam}(\mathcal{X})),$$

$$D_2 = \left(1 + W_e\lambda^{-1}\left(\frac{\text{diam}^2(\mathcal{X})M_\lambda}{m_\lambda - M_\mu}\right)\right)W_\lambda,$$

$$D_3 = \left(1 + W_e\lambda^{-1}\left(\frac{\text{diam}^2(\mathcal{X})M_\lambda}{m_\lambda - M_\mu}\right)\right)(W_f + W_\mu),$$

with some bounded constants $W_e$, $W_\lambda$, $W_\mu$, $W_f$.

It is worth noting that the error $\mathcal{W}_1\left(\text{Law}(\bar{x}_K), \pi\right)$ vanishes as $T \to \infty$, $M \to 0$, and $\delta \to 0$. Consequently, Theorem 4.3 extends classical convergence results for diffusion models (De Bortoli, 2022; Khalafi et al., 2024; Li et al., 2024b) to the context of solving inverse problems. Notably, our convergence guarantees are stronger, as they eliminate the early-stopping error of order $\mathcal{O}(\sqrt{\epsilon})$, which corresponds to the deviation between the green coverage region $t \in [\epsilon, \infty)$ and the blue arrow region $t \in [0, \infty)$ in Figure 2. It typically arises from partitioning the sampling interval into $[0, \epsilon]$ and $[\epsilon, T]$ to address nonsmoothness near $t = 0$. As $t$ approaches zero, the true score $\nabla\log\pi_t(x_t)$ diverges, creating a singularity that must be avoided. In our framework, this issue is naturally resolved by replacing the true score with the well-defined Moreau score $\nabla\log\pi_t^\lambda(x_t)$.

Theorem 4.3 provides the convergence of Algorithm 1, building a theoretical basis for the superior performance in some scenarios. However, the theoretical results are established mainly under convexity-type assumptions for analytical tractability, whereas the real-world problems involve more complex and generally nonconvex settings in practical scenarios. In Section 5, we execute experiments to validate our theoretical results with synthetic data and show practical performance with real-world datasets when assumptions are violated. In future work, we will extend our PGM framework and theoretical results to more general nonconvex settings.

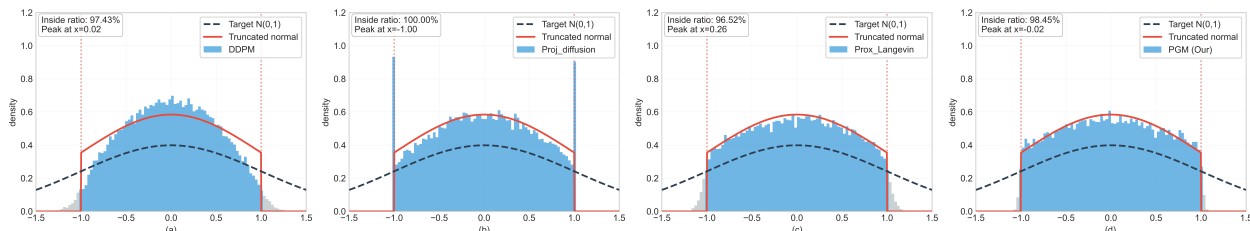

*Figure 3.* **Sampling from truncated normal distribution.** Score-based methods (a) DDPM and (b) Projected diffusion model fail to handle constraint. Proximal-based methods (c) proximal Langevin and (d) PGM (Our) perform better. PGM achieves better feasibility (inside-ratio= $98.45\%$) and optimality (peak at $x = -0.02$).

## 5. Experiments

In this section, we validate the practical performance of PGM through two experiments. To highlight the significance of the theoretical results, we test a toy example on quadratic programming. Further, to validate practical performance of PGM with real-world datasets, we apply PGM to inverse problems across four image restoration tasks: super-resolution, random inpainting, Gaussian deblurring, and nonlinear deblurring. Models are trained on the FFHQ (Kazemi & Sullivan, 2014), CelebA-HQ (Liu et al., 2015), and LSUN-Bedroom (Yu et al., 2015) datasets.

Additional implementation details and extended results are provided in Appendix B. Our code is available at https://github.com/boyangzhang2000/PGM.

### 5.1. Toy Example: Quadratic Programming

Consider the constrained quadratic optimization problem:

$$\min_{\boldsymbol{x} \in \mathbb{R}^d} \quad f(\boldsymbol{x}) = \frac{1}{2} \boldsymbol{x}^\top A \boldsymbol{x} + \boldsymbol{b}^\top \boldsymbol{x} \tag{32}$$
$$\text{s.t.} \quad \boldsymbol{x} \in \mathcal{X} \triangleq \{\boldsymbol{x} : \|x\|_2 \leq r\},$$

where $A, \boldsymbol{b}, r$ are hyperparameters. This corresponds to sampling from $\pi(\boldsymbol{x}) \propto \exp\{-\beta f(\boldsymbol{x})\} \mathbb{I}_{\mathcal{X}}(\boldsymbol{x})$.

To illustrate PGM's superiority in handling constraints, we first visualize sampling from truncated Gaussian $\pi(\boldsymbol{x}) \propto \mathcal{N}(0,1)\mathbb{I}_{[-1,1]}$ where $A = I, \boldsymbol{b} = \boldsymbol{0}, r = 1$. Under these settings, both traditional score and Moreau score (26) can be calculated explicitly. The methods for comparison include DDPM (Ho et al., 2020) and projected diffusion model (Christopher et al., 2024), together with proximal-based methods such as proximal Langevin (Pereyra, 2016) and PGM. Results are evaluated with feasibility, i.e., the proportion of samples satisfying the constraint (inside-ratio), and the optimality, i.e., the peak of the sampling distribution.

As shown in Figure 3, DDPM degrades in performance significantly due to the early-stopping error (see Fig.2). Projected diffusion model disrupts the structure of distribution and causes peaks at the boundary $\{-1, 1\}$. Proximal Langevin has better convergent performance, yet suffers from long mixing time. In contrast, PGM recovers the target

*Table 1.* Feasibility evaluation with various $K$.

| Metric | 10000 PGM samples, $\beta = 10$ | | | | |
|---|---|---|---|---|---|
| | $K = 0$ | $K = 1$ | $K = 5$ | $K = 10$ | $K = 20$ |
| Feasibility | 0.00% | 7.43% | 97.43% | 99.61% | 99.73% |

*Table 2.* Optimality and feasibility evaluation with various $\beta$.

| Metric | 10000 PGM samples, $K = 10$ | | | | |
|---|---|---|---|---|---|
| | $\beta = 0$ | $\beta = 0.1$ | $\beta = 1$ | $\beta = 2$ | $\beta = 10$ |
| Optimality | 0.2365 | 0.2094 | 0.0701 | 0.0438 | 0.0164 |
| Feasibility | 99.90% | 99.88% | 99.78% | 99.66% | 99.61% |

fast and achieves better feasibility and optimality.

To validate the effect of Moreau score matching, we randomly choose $A, \boldsymbol{b}, r$ and generate realizations uniformly from $\mathcal{X}$ for training. In sampling processes, we vary the number of iterations $K$ and evaluate the feasibility rate. Based on the results in Table 1, as $K$ increases, the sample distribution rapidly converges towards the constraint set, and the feasibility rate ultimately approaches 1. This is consistent with the convergence result in Theorem 4.3.

Further, we test the influence of the inverse temperature $\beta$ and calculate the optimality gap, i.e., the difference between the sample objective and the true optimum. Results in Table 2 indicate that as $\beta$ increases, the target distribution $\pi(\boldsymbol{x})$ will concentrate around the optimal solution of the problem, which causes the samples to gradually approach the constraint boundary and converge to the optimizer.

### 5.2. Memory and Sampling Efficiency

To demonstrate the efficiency of PGM, we monitored the memory usage during the solution of an inverse problem, and compared it with baseline methods. Table 3 reports the memory usage for the FFHQ dataset. Although the proposed PGM slightly increases the model size, it is significantly more efficient in terms of the memory usage because it avoids backpropagation through the posterior mean estimator. Moreover, PGM achieves a dramatic speedup in the sampling time, completing in 4s compared to 36s for DPS, representing a $9\times$ improvement in speed.

*Table 3.* Memory and sampling usage of different methods.

| Dataset | Method | # parameters | Model size | Total memory | Sampling time |
|---|---|---|---|---|---|
| FFHQ | DPS | 162,075,353 | 1953 MB | 5369 MB | 36 s |
| | DMPS | 162,075,353 | 1953 MB | 7168 MB | 43 s |
| | PSLD | 329,378,945 | 3969 MB | 9485 MB | 65 s |
| | ReSample | 329,378,945 | 3969 MB | 5009 MB | 54 s |
| | PGM(Our) | 221,673,987 | 2055 MB | **4682** MB | **4** s |

## 5.3. Quantitative Evaluation

We then present a comparative evaluation of human face reconstruction on the FFHQ and CelebA-HQ datasets. Results are compared with state-of-the-art diffusion-based methods including DPS (Chung et al., 2022a), manifold constrained gradients (MCG) (Chung et al., 2022b), plug-and-play using ADMM (ADMM-PnP) (Ahmad et al., 2020), denoising diffusion destoration models (DDRM) (Kawar et al., 2022), diffusion model posterior sampling (DMPS) (Meng & Kabashima, 2022), posterior sampling with latent diffusion (PSLD) (Rout et al., 2023), and Resample (Song et al., 2023a). The pixel level error (e.g., PSNR) and structure error (including SSIM and LPIPS) are measured to show the comparison results. For fair comparison, all methods use similar network backbone and have similar memory usage.

The quantitative results on the FFHQ dataset, as presented in Table 4, demonstrate the superior performance of PGM across all three linear image restoration tasks. This confirms its excellent generalizability and robustness to different types of image degradation problems. Further, PGM achieves a superior trade-off between numerical fidelity and perceptual similarity by simultaneously optimizing PSNR, SSIM, and LPIPS.

*Table 4.* Quantitative evaluation of samples for FFHQ.

| FFHQ | Super resolution 4× | | | Inpainting (random 70%) | | | Gaussian deblurring | | |
|---|---|---|---|---|---|---|---|---|---|
| Method | PSNR ↑ | SSIM ↑ | LPIPS ↓ | PSNR ↑ | SSIM ↑ | LPIPS ↓ | PSNR ↑ | SSIM ↑ | LPIPS ↓ |
| DPS | 28.47 | 0.793 | 0.175 | 32.32 | 0.897 | 0.106 | 27.70 | 0.774 | 0.169 |
| MCG | 23.74 | 0.673 | 0.223 | 24.89 | 0.731 | 0.178 | 25.33 | 0.668 | 0.371 |
| ADMM-PnP | 21.30 | 0.760 | 0.303 | 15.87 | 0.608 | 0.308 | 21.23 | 0.675 | 0.399 |
| DDRM | 27.51 | 0.753 | 0.257 | 24.97 | 0.680 | 0.287 | 26.51 | 0.702 | 0.299 |
| DMPS | 27.21 | 0.766 | 0.181 | 28.17 | 0.814 | 0.150 | 26.04 | 0.699 | 0.227 |
| Latent-DPS | 24.65 | 0.609 | 0.344 | 27.08 | 0.727 | 0.270 | 25.98 | 0.704 | 0.258 |
| PSLD-LDM | 27.22 | 0.705 | 0.267 | 25.61 | 0.630 | 0.270 | 20.08 | 0.400 | 0.422 |
| ReSample | 28.90 | 0.804 | 0.164 | 31.34 | 0.890 | 0.099 | 28.73 | 0.794 | 0.201 |
| PGM (Our) | **29.82** | **0.956** | **0.158** | **32.48** | **0.962** | **0.092** | **30.67** | **0.959** | **0.148** |

Results on the CelebA-HQ dataset in Table 5 further confirm the superiority of PGM. It achieves consistent state-of-the-art performance across all three restoration tasks, indicating that its advantages are not data-specific but stem from fundamental strengths in the proposed proximal-based generative modeling framework.

Consistent with previous results, the SSIM scores of PGM on the FFHQ and CelebA-HQ datasets are exceptionally high, i.e., all of the scores are above 0.95, approaching the theoretical maximum of 1.0. This indicates that the restored images are virtually indistinguishable from the ground truth

*Table 5.* Quantitative evaluation of samples for CelebA-HQ.

| CelebA-HQ | Super resolution 4× | | | Inpainting (random 70%) | | | Gaussian deblurring | | |
|---|---|---|---|---|---|---|---|---|---|
| Method | PSNR ↑ | SSIM ↑ | LPIPS ↓ | PSNR ↑ | SSIM ↑ | LPIPS ↓ | PSNR ↑ | SSIM ↑ | LPIPS ↓ |
| DPS | 28.41 | 0.782 | 0.173 | 32.48 | 0.899 | 0.102 | 28.36 | 0.772 | 0.175 |
| MCG | 25.92 | 0.740 | 0.193 | 29.53 | 0.847 | 0.134 | 15.85 | 0.536 | 0.517 |
| ADMM-PnP | 21.08 | 0.631 | 0.304 | 15.40 | 0.342 | 0.627 | 20.98 | 0.602 | 0.289 |
| DDRM | 29.49 | 0.817 | 0.151 | 27.69 | 0.798 | 0.166 | 26.88 | 0.747 | 0.193 |
| DMPS | 28.48 | 0.811 | 0.147 | 28.84 | 0.826 | 0.175 | 26.45 | 0.726 | 0.206 |
| Latent-DPS | 26.83 | 0.690 | 0.272 | 26.23 | 0.703 | 0.226 | 27.42 | 0.729 | 0.205 |
| PSLD-LDM | 27.61 | 0.704 | 0.209 | 27.07 | 0.689 | 0.260 | 24.21 | 0.548 | 0.323 |
| ReSample | 30.45 | 0.832 | 0.144 | 32.77 | 0.903 | 0.082 | 30.69 | 0.832 | 0.148 |
| PGM (Our) | **30.84** | **0.951** | **0.131** | **33.41** | **0.960** | **0.076** | **31.52** | **0.957** | **0.129** |

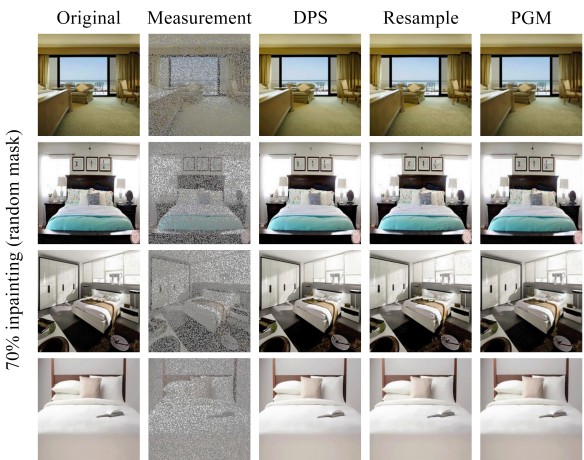

Original   Measurement   DPS   Resample   PGM

70% inpainting (random mask)

*Figure 4.* Visual samples for LSUN-Bedroom.

in terms of structural information—a crucial measure for high-quality image restoration.

## 5.4. Visualizing Confirmation

To evaluate the cross-prior generalization capability of PGM, we execute additional tests on the LSUN-Bedroom datasets. Qualitative results on LSUN-Bedroom are displayed in Figure 4, visually confirming the model's capability to generate high-fidelity and natural-looking images.

## 6. Conclusion

In this paper, we have introduced PGM, a novel framework that bridges proximal operators and generative modeling to address fundamental challenges in Bayesian inverse problems. Built upon a theoretical equivalence between Gaussian convolution and Moreau-Yosida regularization, PGM enables a new pre-trained proximal diffusion sampling mechanism driven by the proposed Moreau score. We provide non-asymptotic convergence guarantees under convexity and demonstrate state-of-the-art empirical performance across multiple image restoration tasks. In summary, PGM not only advances the theoretical understanding of generative modeling but also offers a practical, efficient, and highly effective tool for tackling inverse problems in imaging and beyond.

## Acknowledgements

The work of Boyang Zhang and Ya-Feng Liu was supported in part by the National Key R&D Program of China under Grant 2024YFA1014203 and the National Natural Science Foundation of China (NSFC) under Grant 12371314. The work of Zhiguo Wang was supported in part by the NSFC under Grant 12571503 and the Open Foundation of the State Key Laboratory of Mathematical Sciences under Grant SLMS-2025-KFKT-TD-02.

## Impact Statement

This paper presents work whose goal is to advance the field of Machine Learning. There are many potential societal consequences of our work, none which we feel must be specifically highlighted here.

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

## Outline of Appendices

The appendices are organized as follows:

(i) **Appendix** A provides complement related works and detailed discussions on limitations and future research directions of this work.

## A. Limitations and Future Works

### A.1. Complement Related Works

**Diffusion models for inverse problems.** Recent years have seen growing interest in using diffusion models as powerful structural priors for solving inverse problems, motivated by their success in high fidelity image generation (Dhariwal & Nichol, 2021; Ho et al., 2022; Rombach et al., 2022; Li & Wang, 2025; Zhang et al., 2025b) and time-varying channel modeling (Croitoru et al., 2023; Wu et al., 2024b; Zhou et al., 2025; Tang et al., 2025; Fan et al., 2025).

To solve an inverse problem, the goal is to sample from the posterior distribution $p(\boldsymbol{x}_0|\boldsymbol{y})$. According to Bayes' theorem:

$$p(\boldsymbol{x}_0|\boldsymbol{y}) \propto p(\boldsymbol{y}|\boldsymbol{x}_0)p(\boldsymbol{x}_0).$$

To implement this within the diffusion framework, we must modify the reverse process to sample from the conditional path $p(\boldsymbol{x}_{0:T}|\boldsymbol{y})$. For the reverse SDE, this means replacing the unconditional score $\nabla_{\boldsymbol{x}} \log p_t(\boldsymbol{x}_t)$ with the conditional score $\nabla_{\boldsymbol{x}} \log p_t(\boldsymbol{x}_t|\boldsymbol{y})$.

In some early works, a conditional neural network architecture is explicitly trained to map the specific measurement $\boldsymbol{y}$ directly to the clean image $\boldsymbol{x}_0$. They do this by learning the conditional score $\nabla_{\boldsymbol{x}} \log p_t(\boldsymbol{x}_t|\boldsymbol{y})$ directly from massive datasets of paired training data $(\boldsymbol{x}_0^{(i)}, \boldsymbol{y}^{(i)})$. SR3 (Saharia et al., 2022b) and Palette (Saharia et al., 2022a) define the foundational approach for supervised conditional diffusion. These models treat the inverse problem strictly as a supervised learning task. Although highly effective for fixed tasks, these approaches require large task-specific paired datasets and retraining for each new forward operator, limiting their flexibility and generalizability.

In recent works, people are seeking methods that require absolutely no retraining or fine-tuning of the diffusion model parameters, called zero-shot posterior sampling. Instead, they mathematically modify the reverse sampling trajectory (either through hard geometric projections or gradient-based score approximations) to force the generated sample to agree with the measurement $\boldsymbol{y}$.

For linear inverse problems, subspace and null-space projection is applied to modify the sampling trajectory by forcing the intermediate sample to directly satisfy the measurement consistency. DDRM (Kawar et al., 2022) formulated one of the first mathematically exact, non-heuristic solvers specifically targeting arbitrary linear inverse problems. It relies heavily on the singular value decomposition (SVD) of the measurement matrix $A = U\Sigma V^\top$. DDNM (Wang et al., 2022) utilizes the Moore-Penrose pseudoinverse $A^\dagger$ to decompose the data strictly into a range-space and a null-space $\boldsymbol{x} = A^\dagger A\boldsymbol{x} + (I - A^\dagger A)\boldsymbol{x}$.

It enforces absolute data consistency by overwriting the range-space with the exact measurements, while retaining the generative prior in the null space. DiffStateGrad (Zirvi et al., 2025) projected the measurement gradient onto a subspace that is a low-rank approximation of an intermediate state of the diffusion process. As a module, it can be added to a wide range of diffusion-based inverse solvers to improve the preservation of the diffusion process on the prior manifold and filter out artifact-inducing components.

For nonlinear inverse problems, a series of approximations on the conditional score $\nabla_{\boldsymbol{x}} \log p_t(\boldsymbol{x}_t|\boldsymbol{y})$ is applied to obtain an explicit sampling process (Chung et al., 2022a; Song et al., 2023b; Boys et al., 2023; Wu et al., 2024a; Xu et al., 2025). Applying Bayes' rule at an intermediate timestep $t$, we can decompose this conditional score as (2). The unconditional score is provided off-the-shelf by the pre-trained diffusion model $\boldsymbol{s}_{\boldsymbol{\theta}}(\boldsymbol{x}_t, t)$. Much of diffusion-based zero-shot nonlinear inverse problem solvers is fundamentally dedicated to computing or approximating the second term $\nabla_{\boldsymbol{x}} \log p(\boldsymbol{y}|\boldsymbol{x}_t)$. Note that the physical measurement $\boldsymbol{y}$ depends on the clean, uncorrupted signal $\boldsymbol{x}_0$ via $\boldsymbol{y} = \mathcal{A}(\boldsymbol{x}_0) + \boldsymbol{\xi}$, not on the intermediate noisy state $\boldsymbol{x}_t$. To compute the likelihood exactly, one must marginalize over all possible clean images

$$\nabla_{\boldsymbol{x}} \log p(\boldsymbol{y}|\boldsymbol{x}_t) = \int_{\mathbb{R}^d} p(\boldsymbol{y}|\boldsymbol{x}_0) p(\boldsymbol{x}_0|\boldsymbol{x}_t) d\boldsymbol{x}_0.$$

Here, $p(\boldsymbol{y}|\boldsymbol{x}_0)$ is known. However, $p(\boldsymbol{x}_0|\boldsymbol{x}_t)$ is the true reverse transition distribution mapping a noisy image back to the clean manifold. This distribution is highly complex, multimodal, and analytically intractable. Thus, the integral cannot be computed directly.

A prominent class of approaches is posterior diffusion sampling, which focuses on estimating the time-dependent likelihood score $\nabla_{\boldsymbol{x}} \log p(\boldsymbol{y}|\boldsymbol{x}_t)$. Diffusion posterior sampling (DPS) (Chung et al., 2022a) addresses this challenge by combining diffusion sampling with manifold constrained gradients (Chung et al., 2022b). This approach relies on a Laplace approximation of the conditional likelihood:

$$p_t(\boldsymbol{y}|\boldsymbol{x}_t) \sim \mathcal{N}(A\hat{\boldsymbol{x}}_0(\boldsymbol{x}_t), \sigma_{\xi}^2 I),$$

where $\hat{\boldsymbol{x}}_0(\boldsymbol{x}_t) \approx \mathbb{E}[\boldsymbol{x}_0|\boldsymbol{x}_t]$ is a posterior mean estimator with a structured network. Instead of relying purely on the mean, TMPD (Boys et al., 2023) utilizes an extended Tweedie's formula to estimate the full covariance matrix $\text{Cov}[\boldsymbol{x}_0|\boldsymbol{x}_t]$, which involves calculating the complex Jacobian of the score function. The diffusion process is then projected such that it matches the exact statistical moments of the true posterior at every step. Subsequent work of Resample (Song et al., 2023a) extended this paradigm to latent diffusion models through a two-stage data consistency framework, incorporating data consistency by solving an optimization problem during the reverse sampling process. Despite their strong empirical performance, most existing methods require backpropagation through the posterior mean estimator at each sampling step, which leads to substantial computational overhead and memory consumption.

Other categories to avoid calculation of the time-dependent conditional score $\nabla_{\boldsymbol{x}} \log p(\boldsymbol{y}|\boldsymbol{x}_t)$ are plug-and-play (PnP) and variational inference (VI). PnP algorithms adapt decades of classical convex optimization literature by substituting classical math filters with diffusion models acting as regularizing denoisers. VI formalizes inverse problem solving from a purely statistical perspective by minimizing the evidence lower bound (ELBO) of the true posterior. Both focus on solving specific regularized optimization problems. DiffPIR (Zhu et al., 2023) bridges the gap between classical half-quadratic splitting (HQS) optimization and DDPM prior for nonlinear inverse problems. It splits the MAP objective into two alternating subproblems. The data-fidelity step solves the physics equation, while the prior step solves a regularized problem by replacing the classical prior entirely with the unconditional DDPM estimation. RED-Diff (Mardani et al., 2023) constructs an explicit regularized objective using a DDPM denoiser and performs joint gradient-based optimization. DAPS (Zhang et al., 2025a) mathematically decouples the physical diffusion SDE timestep $t$ from an algorithmic annealing parameter. The formulation becomes a sequence of relaxed inverse problems where the assumed measurement noise variance is gradually reduced to zero completely independently of the SDE state, dynamically easing the strictness of data-consistency. EquiReg (Tolooshams et al., 2025) further enforces physical symmetries, such as rotation or translation equivariance, during sampling.

Despite their success, most of these methods are based on traditional score matching. As a result, for nonsmooth priors in the real world, these methods lack convergence guarantees and may degrade in performance for practical applications.

**Proximal Langevin Algorithm.** Classical posterior sampling has long been dominated by Markov chain Monte Carlo (MCMC) methods. These approaches are most effective when the prior is conjugate to the likelihood or when the negative log-density is smooth with Lipschitz continuous gradients (Geyer, 1992). Such assumptions are violated by many modern imaging priors involving nonsmooth regularizers, including $\ell_1$ penalties or total variation norms.

To address this limitation, proximal Langevin algorithms have been developed (Pereyra, 2016; Salim et al., 2019; Pillai, 2024). These methods constitute an important step toward bridging convex optimization and stochastic simulation by replacing gradients with proximal mappings and Moreau–Yosida envelopes. This convex analytic machinery enables scalable Bayesian inference for high-dimensional log-concave models that are not differentiable, thereby supporting uncertainty quantification and sparse regression in settings previously accessible mainly through deterministic optimization.

However, existing proximal unadjusted Langevin algorithms still rely on explicit knowledge of the prior potential, which restricts their applicability in imaging sciences. In parallel, the field has witnessed a shift away from hand-crafted regularizers toward expressive learned priors, including variational autoencoders, generative adversarial networks, and diffusion models (Kingma & Welling, 2013; Goodfellow et al., 2014; Ho et al., 2020; Chung et al., 2022a). Such implicit priors are accessible only through samples or score networks, preventing the direct application of traditional Langevin or proximal MCMC schemes that require analytic access to the underlying potential function.

**Convergence of the diffusion model.** The empirical success of diffusion models has motivated rigorous efforts to understand their theoretical convergence properties (De Bortoli, 2022; Benton et al., 2023; Gu et al., 2023; Li et al., 2024a; Zhang et al., 2026). For example, (Lee et al., 2022) provided the first polynomial convergence guarantee for smooth distributions satisfying the log-Sobolev inequality. (Chen et al., 2022) accommodated a broad family of data distributions under the assumption that the score functions along the entire forward process trajectory are Lipschitz. A non-asymptotic analysis for the discretized probability flow ODE was derived by (Chen et al., 2023), establishing the first non-asymptotic convergence guarantees. Further, (Li et al., 2023) developed a suite of non-asymptotic theory to understand the discrete-time data generation process, demonstrating a faster convergence rate.

Despite their success, a fundamental limitation persists: an early-stopping mechanism is typically required to ensure smoothness, which introduces a bias dependent on the stopping time. This early-stopping error corresponds to the deviation between the reverse sampling time region $t \in [\epsilon, \infty)$ and the forward diffusion time region $t \in [0, \infty)$. It typically arises from partitioning the sampling interval into $[0, \epsilon]$ and $[\epsilon, T]$ to address nonsmoothness near $t = 0$. As $t$ approaches zero, the true score $\nabla \log \pi_t(\boldsymbol{x}_t)$ diverges, creating a singularity that must be avoided.

Currently, there are still limited convergence results for diffusion models in the context of solving inverse problems. In (Song et al., 2023a), a variance induction result is provided for the stochastic resampling process. Further in (Li & Wang, 2025), a total variation distance error is characterized for the middle approximation processes. However, these results rely heavily on strong assumptions, such as the Gaussian distribution or dirac posterior assumption.

## A.2. Connections with Existing Algorithms

**Connection with P-ULA (Pereyra, 2016).** Consider a VE-SDE forward process with $\mu(t) \equiv 1$ and $\sigma(\tau_{k+1})/\sigma(\tau_k) \equiv 1 - \rho_k$, the iteration (27) reduces to

$$\bar{\boldsymbol{x}}_{k+1} = (1 - \rho_k)\,\bar{\boldsymbol{x}}_k + \rho_k \boldsymbol{P}_k + \sqrt{\lambda(\tau_k)/\lambda}\boldsymbol{\xi}_k,$$

which recovers the P-ULA iteration in (11) up to a minor difference in the noise scaling. If we instead set $\sigma(\tau_{k+1})/\sigma(\tau_k) = \rho_k^2$ and consider the reverse-time ordinary differential equation associated with (16), then

$$\boldsymbol{x}_{k+1} = (1 - \rho_k)\boldsymbol{x}_k + \rho_k \boldsymbol{P}_k,$$

which can be interpreted as a generalized proximal point iteration (Rockafellar, 1976; Eckstein & Bertsekas, 1992).

**Connection with DPS (Chung et al., 2022a).** In the context of linear inverse problem (6) with $\mathcal{A}(\boldsymbol{x}) = A\boldsymbol{x}$, suppose that $p(\boldsymbol{x}_0|\boldsymbol{x}_t) \sim \mathcal{N}(\hat{\boldsymbol{x}}_0(\boldsymbol{x}_t), \lambda(t)I)$ for some posterior mean estimator $\hat{\boldsymbol{x}}_0(\boldsymbol{x}_t)$ and $b(t) \equiv 1$, then DPS (Chung et al., 2022a) takes the reverse SDE:

$$d\boldsymbol{x}_t = \frac{\boldsymbol{x}_t - \hat{\boldsymbol{x}}_0(\boldsymbol{x}_t) + \eta_t \lambda(t)\nabla_{\boldsymbol{x}_t} f_{\boldsymbol{y}}(\hat{\boldsymbol{x}}_0(\boldsymbol{x}_t))}{\lambda(t)}dt + d\bar{\boldsymbol{B}}_t,$$

with a predefined weighting $\eta_t$. In contrast, PGM employs

$$d\boldsymbol{x}_t = \frac{\boldsymbol{x}_t - \mathrm{Prox}_g^{\lambda(t)}\left(\boldsymbol{x}_t - \beta\lambda(t)\nabla f_{\boldsymbol{y}}(\boldsymbol{x}_t)\right)}{\lambda(t)}dt + d\bar{\boldsymbol{B}}_t.$$

Compared with DPS, PGM avoids backpropagation through the posterior mean estimator $\hat{\boldsymbol{x}}_0(\boldsymbol{x}_t)$, resulting in substantial savings in memory usage and sampling time, which is confirmed by the experiments in Section 5.

**Connection with DiffPIR (Zhu et al., 2023).** Both methods address Bayesian inverse problems with prior $p \propto \exp\{-g\}$ and likelihood $\exp\{-f\}$, and both are related to proximal-type optimization ideas. However, the key difference lies in how the optimization problem is approximated and solved. From an optimization viewpoint, the target problem is

$$\min_{\boldsymbol{x}} \ f(\boldsymbol{x}) + g(\boldsymbol{x}).$$

DiffPIR adopts a half-quadratic splitting strategy and considers the augmented problem

$$\min_{\boldsymbol{x},\boldsymbol{z}} \ f(\boldsymbol{x}) + g(\boldsymbol{z}) + \frac{1}{2\lambda/\mu}\|\boldsymbol{x} - \boldsymbol{z}\|^2,$$

which leads to the two subproblems

$$\boldsymbol{z}_k = \arg\min_{\boldsymbol{z}} \frac{1}{2\lambda/\mu}\|\boldsymbol{z} - \boldsymbol{x}_k\|^2 + g(\boldsymbol{z}),$$

$$\boldsymbol{x}_{k+1} = \arg\min_{\boldsymbol{x}} f(\boldsymbol{x}) + \mu\sigma_n^2\|\boldsymbol{x} - \boldsymbol{z}_k\|^2.$$

To solve the first subproblem, DiffPIR further uses a first-order approximation

$$g(\boldsymbol{z}) \approx g(\boldsymbol{x}_k) + \nabla g(\boldsymbol{x}_k)^\top(\boldsymbol{z} - \boldsymbol{x}_k),$$

where the Stein score $\boldsymbol{s}_{\boldsymbol{\theta}}(\boldsymbol{x})$ is used to approximate $\nabla \log p(\boldsymbol{x})$, equivalently $-\nabla g(\boldsymbol{x})$. This approximation relies on the smoothness of $g$, which does not hold in our nonsmooth setting. In contrast, PGM does not approximate the prior term $g$. Instead, we linearize the likelihood term

$$f(\boldsymbol{x}) \approx f(\boldsymbol{x}_k) + \nabla f(\boldsymbol{x}_k)^\top(\boldsymbol{x} - \boldsymbol{x}_k),$$

and apply a proximal-gradient step directly to the original problem:

$$\boldsymbol{x}_{k+1} = \arg\min_{\boldsymbol{x}} \nabla f(\boldsymbol{x}_k)^\top(\boldsymbol{x} - \boldsymbol{x}_k) + g(\boldsymbol{x}) + \frac{1}{2\mu\sigma_n^2}\|\boldsymbol{x} - \boldsymbol{x}_k\|^2.$$

This leads to the proximal splitting

$$\boldsymbol{x}_{k+1} = \mathrm{Prox}_g^{\lambda/\mu}\left(\boldsymbol{x}_k - \frac{1}{2\mu\sigma_n^2}\nabla f(\boldsymbol{x}_k)\right).$$

We then learn a proximal network $\boldsymbol{\phi}_{\boldsymbol{\theta}}(\boldsymbol{x}, \lambda)$ to approximate $\mathrm{Prox}_g^\lambda(\boldsymbol{x})$, which naturally accommodates nonsmooth priors.

Overall, while both methods are motivated by proximal optimization, DiffPIR approximates and solves a split formulation by smoothing the prior subproblem, whereas PGM applies proximal gradient directly to the original objective and approximates the proximal operator itself. This difference in the approximation and optimization strategy leads to two distinct frameworks, suitable for different problem settings.

## A.3. Limitations

Despite strong performance, PGM has some limitations:

- **Training complexity**: Learning proximal operators requires careful hyperparameter tuning.

- **Network design**: Network architectures could be more consistent with the proximal properties.

- **Nonconvex priors**: The theoretical guarantee is insufficient under nonconvex settings.

These limitations suggest directions for future work, including more efficient architectures and extensions to non-convex settings.

## A.4. Future Works

While our framework shows promising results, several interesting directions remain for future research:

- **Nonconvex priors**: Our current theory primarily addresses convex priors. Extending the framework to handle nonconvex priors through nonconvex proximal operators or other generalizations would significantly broaden its applicability.

- **Convergence rates**: While we established convergence guarantees, deriving explicit convergence rates under more general conditions and characterizing the dependence on the problem dimension would provide deeper theoretical understanding.

# B. More Details of Experiments

## B.1. Experimental Setup

**Parameters.** Our PGM implementation uses the following default configurations:

- U-Net architecture: 128 hidden channels, 128 time-embedding dimension, 4 encoder blocks, and 4 decoder blocks.

- Training: AdamW optimizer, learning rate $10^{-4}$, weight decay $10^{-5}$, batch size 8, and warm-up cosine scheduler.

- Regularization schedule: $\lambda(t) = \exp(10t - 8)$.

- Proximal matching schedule: $\zeta(t) = 10/t$.

- Inverse temperature: $\beta = 10.0$.

- Number of steps: $K = 100$.

Our code is available at https://github.com/boyangzhang2000/PGM.

**Network backbone.** All baseline methods use the same pre-trained backbone ADM (Dhariwal & Nichol, 2021) to ensure a fair comparison, which improves the U-Net architecture with time embedding, multi-head attention, and BigGAN residual blocks for upsampling/downsampling. PGM uses a similar U-Net backbone for the proximal network, which adopts $\lambda$ embedding and simpler CNN residual blocks.

**Inverse problem settings.** We apply PGM to inverse problems across four image restoration tasks: super-resolution, random mask inpainting, Gaussian deblurring, and nonlinear deblurring (Tran et al., 2021). Models are trained on the MNIST (LeCun et al., 2002), FFHQ (Kazemi & Sullivan, 2014), CelebA-HQ (Liu et al., 2015), LSUN-Bedroom (Yu et al., 2015), and ImageNet-100 (Zhao et al., 2020) datasets. Various quantitative metrics are used for evaluation including learned perceptual image patch similarity (LPIPS) distance, peak signal-to-noise-ratio (PSNR), and structural similarity index (SSIM).

The experimental setup for different inverse problems is summarized in Table 6:

*Table 6.* Inverse problem configurations

| Problem type | Key parameters | Values |
|:---:|:---:|:---:|
| **Inpainting** | Type | random |
| | Mask ratio | 0.7 |
| | Noise level | 0.01 / 0.05 |
| **Super-resolution** | Scale factor | 4× |
| | Upsampling | Bilinear |
| | Noise level | 0.01 / 0.05 |
| **Gaussian deblurring** | Kernel size | 61 |
| | Sigma ($\sigma$) | 3.0 |
| | Noise level | 0.01 / 0.05 |

**Baseline methods.** Our proposed PGM framework is compared with state-of-the-art generative models: DPS (Chung et al., 2022a), manifold constrained gradients (MCG) (Chung et al., 2022b), denoising diffusion destoration models (DDRM) (Kawar et al., 2022), DDNM (Wang et al., 2022), diffusion model posterior sampling (DMPS) (Meng & Kabashima, 2022), plug-and-play using ADMM (ADMM-PnP) (Ahmad et al., 2020), ΠGDM (Song et al., 2023b), posterior sampling with latent diffusion (PSLD) (Rout et al., 2023), Resample (Song et al., 2023a), Tweedie moment projected diffusion (TMPD) (Boys et al., 2023), RED-diff (Mardani et al., 2023), DiffPIR (Zhu et al., 2023), SITCOM (Alkhouri et al., 2024), decoupled annealing posterior sampling (DAPS) (Zhang et al., 2025a), diffusion state-guided projected gradient (DiffStateGrad) (Zirvi et al., 2025), and equivariance regularized (EquiReg) diffusion (Tolooshams et al., 2025).

## B.2. Training Costs

As a kind of zero-shot posterior sampling method, PGM still follows the pre-trained prior paradigm. Once the proximal operator is learned, it can be used as a pre-trained operator for any sampling processes or further fine-tuning. Though this training is performed only once on prior samples and can be reused across different inverse problems, there is still necessity to compare the training costs with pre-trained unconditional diffusion models. Some results on the runtime and NFEs of ImageNet-100 dataset are shown in Table 7, where all baseline methods and PGM are tested on 4 NVIDIA A800 GPUs.

*Table 7.* Training and sampling costs comparison.

| Method | Model type | Model size | Training time | Sampling time | NFEs |
|---|---|---|---|---|---|
| PGM(Ours) | ProxNet (Our) | 2055MB | 51h32min | **4s** | **100** |
| DAPS | | | | 33s | 1000 |
| DiffStateGrad-DAPS | Pre-trained [4] | 2108MB | N/A | 37s | 1000 |
| SITCOM | | | | 21s | 600 |

Compared with the baseline methods, the main advantage of PGM is that it achieves substantially lower sampling time and fewer NFEs while remaining competitive in the reconstruction quality. Although training the proximal operator requires additional computations, this is a one-time cost. Once the proximal operator is learned, it can be used as a pre-trained network for any optimization/sampling algorithms or further fine-tuning.

## B.3. Additional Results on MNIST Inpainting

We evaluate PGM on the MNIST inpainting task. We compare with VAE-based methods, e.g., VQ-VAE-2 (Razavi et al., 2019), S-IntroVAE (Daniel & Tamar, 2021), and $\beta$-VAET (Solera-Rico et al., 2024), as well as diffusion models DDPM (Ho et al., 2020) and DeepONet (Fang et al., 2025). Various quantitative metrics are used for evaluation, including peak signal-to-noise-ratio (PSNR), structural similarity index (SSIM), and mean square error (MSE).

*Table 8.* Quantitative evaluation of samples for MNIST.

| Method | MNIST inpainting | | |
|---|---|---|---|
| | PSNR ↑ | SSIM ↑ | MSE ↓ |
| VAE (Kingma & Welling, 2013) | 18.0573 | 0.6963 | 0.0477 |
| VQ-VAE-2 (Razavi et al., 2019) | 20.1591 | 0.7107 | 0.0211 |
| S-IntroVAE (Daniel & Tamar, 2021) | 21.6251 | 0.8703 | 0.0114 |
| DDPM (Ho et al., 2020) | 26.0319 | 0.9448 | 0.0090 |
| $\beta$-VAET (Solera-Rico et al., 2024) | 14.0290 | 0.4101 | 0.5788 |
| DeepONet (Fang et al., 2025) | 28.1062 | 0.9673 | 0.0085 |
| PGM (Ours) | **28.6867** | **0.9908** | **0.0048** |

Quantitative results in Table 8 show that PGM achieves the highest reconstruction fidelity across all evaluation metrics (including PSNR, SSIM, and MSE), outperforming all the compared baselines. Specifically, PGM reaches a remarkably high SSIM, implying that PGM has excellent structural and visual fidelity, and can perfectly reconstruct important visual structure and texture information in the original images.

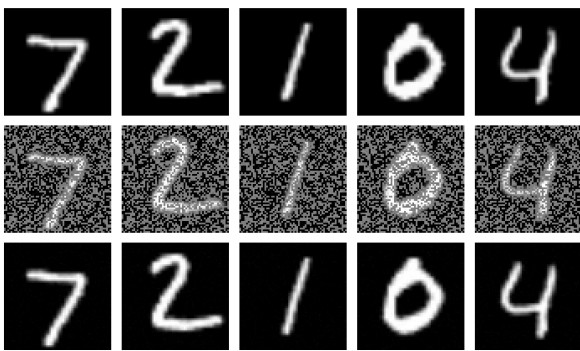

*Figure 5.* Samples for MNIST (first line: original images, second line: measurements, third line: reconstructed images).

### B.4. Additional Results on Human Face Reconstruction

Qualitative results on FFHQ and CelebA-HQ are displayed in Figures 6 and 7, respectively, visually confirming the model's capability to generate high-fidelity and natural-looking images.

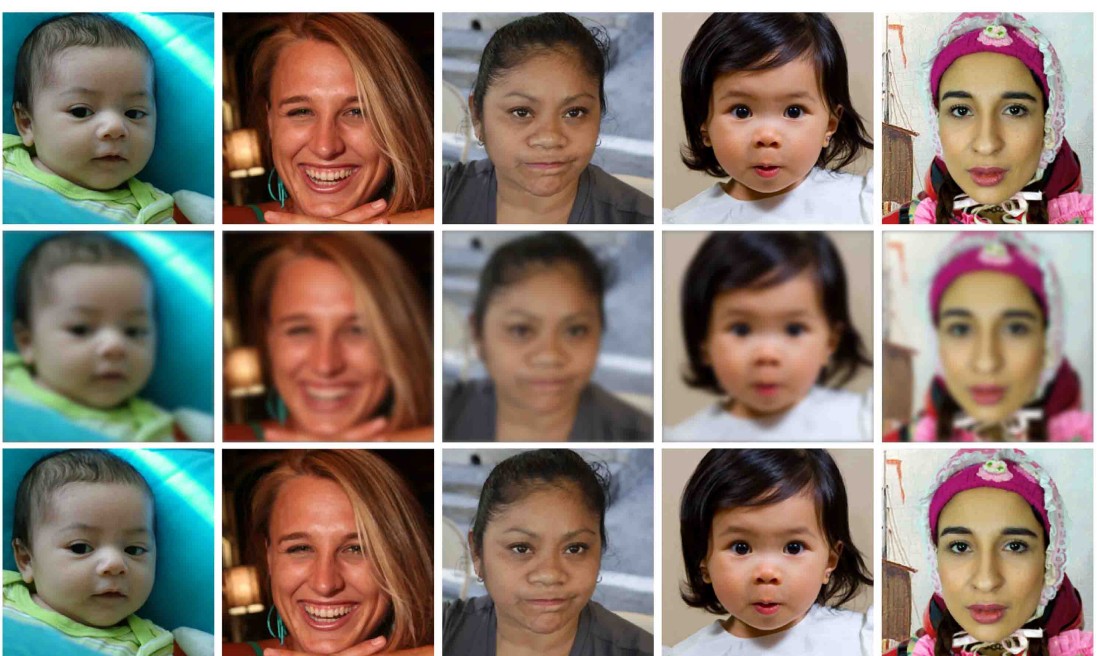

*Figure 6.* Samples for FFHQ (first line: original images, second line: measurements, third line: reconstructed images).

Further, we provide the Pareto front to compare the trade-off between reconstruction quality and inference time. In Figure 8, we show the least inference cost, measured by the number of neural function evaluations (NFEs), of various methods with a fixed target reconstruction quality (PSNR) for the FFHQ Gaussian deblurring problem.

The experimental results indicate that, for a fixed PSNR target, the number of NFEs required by PGM is less than that of other methods, and this difference becomes more pronounced as the PSNR increases. This suggests that, compared to existing methods, PGM exhibits a better performance trade-off.

### B.5. Additional Results on ImageNet-100

We further test performances on ImageNet-100 dataset and compare with recent works on diffusion-based inverse problems. Some results are shown in Table 9. PSNR and LPIPS results are presented as mean ± standard deviation over 100 fixed validation images. Some results are shown below, where performances of baseline methods are taken from (Alkhouri et al.,

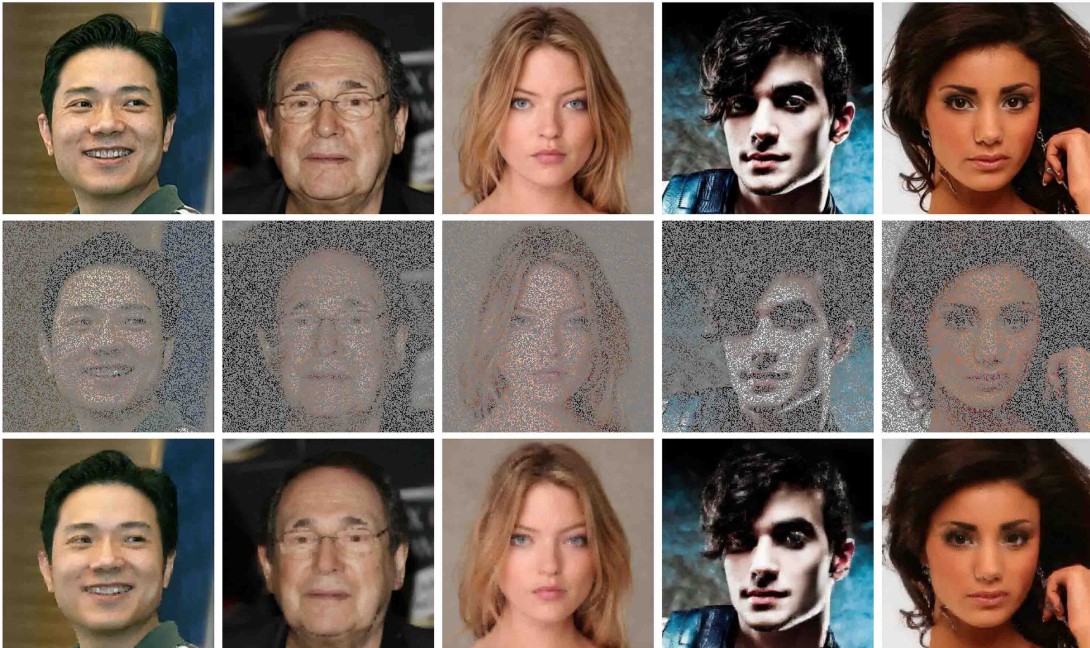

*Figure 7.* Samples for CelebA-HQ (first line: original images, second line: measurements, third line: reconstructed images).

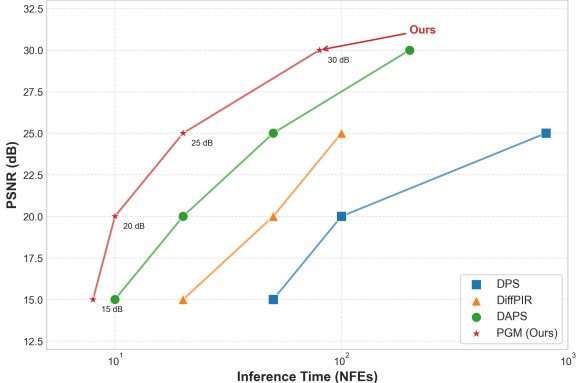

*Figure 8.* Trade-off between reconstruction quality and inference time.

2024; Zhang et al., 2025a; Zirvi et al., 2025).

Based on the quantitative results on ImageNet-100, our proposed PGM achieves competitive performance across three inverse problems. For super resolution and inpainting, PGM attains the highest PSNR while DiffStateGrad-DAPS obtains the best LPIPS. For Gaussian deblurring, PGM obtains the best LPIPS and competitive PSNR. Overall, PGM demonstrates strong reconstruction quality in terms of both distortion and perceptual similarity.

### B.6. Additional Results on Nonlinear Deblurring

In our PGM framework, the only information about $f$ required is its gradient $\nabla f$; therefore, the method naturally extends to nonlinear inverse problems. Since nonlinearity often leads to nonconvexity, the assumptions underlying Theorem 4.3 may no longer hold in such settings. As a result, the current convergence theory is not directly applicable. Nevertheless, the algorithm itself remains well-defined and can still be used in practice. To evaluate its behavior beyond the convex regime, we add additional experiments on nonlinear inverse problems. We use the default setting and leverage the neural network approximated forward model as described in (Tran et al., 2021) for nonlinear deblurring, i.e., $f_{\boldsymbol{y}}(\boldsymbol{x}) = \frac{1}{2}\|\boldsymbol{y} - \mathcal{G}_{\boldsymbol{\theta}}(\boldsymbol{x})\|^2$. All Gaussian noise is added to the measurement domain with $\sigma = 0.05$. Selected results are listed in Table 10.

*Table 9.* Quantitative evaluation of samples for ImageNet-100.

| ImageNet-100 | Super Resolution 4× | | Inpainting Random 70% | | Gaussian Deblurring | |
|---|---|---|---|---|---|---|
| Method | PSNR↑ | LPIPS↓ | PSNR↑ | LPIPS↓ | PSNR↑ | LPIPS↓ |
| PGM(Ours) | **26.60±2.90** | 0.264±0.060 | **31.14±3.28** | 0.116±0.045 | 26.65±2.93 | **0.206±0.050** |
| DPS | 23.86±0.34 | 0.357±0.069 | 24.26±0.42 | 0.326±0.034 | 21.86±0.45 | 0.362±0.034 |
| DDNM | 23.96±0.89 | 0.475±0.044 | 29.22±0.55 | 0.191±0.048 | **28.06±0.52** | 0.278±0.089 |
| DAPS | 25.67±0.73 | 0.256±0.067 | 28.44±0.45 | 0.135±0.052 | 26.12±0.78 | 0.245±0.022 |
| DiffStateGrad-DAPS | 26.40±3.44 | **0.229±0.057** | 29.78±4.17 | **0.107±0.037** | 25.87±3.56 | 0.243±0.075 |
| SITCOM | 26.35±1.21 | 0.232±0.038 | 29.60±0.78 | 0.127±0.039 | 27.40±0.45 | 0.236±0.039 |

*Table 10.* Quantitative evaluation of nonlinear deblurring.

| Nonlinear Deblurring | | FFHQ | | | ImageNet | | |
|---|---|---|---|---|---|---|---|
| Methods | Type | PSNR | SSIM | LPIPS | PSNR | SSIM | LPIPS |
| PGM(Ours) | | 28.47 | 0.756 | 0.231 | 26.39 | 0.727 | 0.280 |
| DPS | Pixel | 23.39 | 0.623 | 0.278 | 22.49 | 0.591 | 0.306 |
| RED-diff | | 30.86 | **0.795** | 0.160 | **30.07** | **0.754** | 0.211 |
| DAPS | | 28.29 | 0.783 | **0.155** | 27.73 | 0.724 | **0.169** |
| ReSample | Latent | 28.24 | 0.742 | 0.185 | 26.20 | 0.653 | 0.206 |
| LatentDAPS | | 28.11 | 0.713 | 0.235 | 25.34 | 0.615 | 0.314 |

The experimental results indicate that, although the performance of PGM is superior to that of the classical DPS, it is inferior compared to the latest methods suitable for handling nonlinear inverse problems. This is caused by the first-order approximation $f(\boldsymbol{x}) \approx f(\boldsymbol{x}_k) + \nabla f(\boldsymbol{x}_k)^\top (\boldsymbol{x} - \boldsymbol{x}_k)$ in operator splitting. As nonlinear inverse problems often suffer from severe nonconvexity, the performance of PGM may degrade. In future work, we will investigate how to extend PGM to general nonconvex problems and explore its convergence results in such settings.

## C. Properties of Moreau Approximation

### C.1. Moreau-Yosida Regularization

Given a proper closed convex function $f : \mathbb{R}^d \to \mathbb{R} \cup \{+\infty\}$, its Moreau-Yosida regularization (or Moreau envelope) with parameter $\lambda > 0$ is defined as:

$$f^\lambda(\boldsymbol{x}) = \min_{\boldsymbol{u} \in \mathbb{R}^d} \left\{ f(\boldsymbol{u}) + \frac{1}{2\lambda} \|\boldsymbol{u} - \boldsymbol{x}\|^2 \right\}. \tag{33}$$

And the proximal operator associated with $f$ is:

$$\text{Prox}_f^\lambda(\boldsymbol{x}) = \underset{\boldsymbol{u} \in \mathbb{R}^d}{\arg\min} \left\{ f(\boldsymbol{u}) + \frac{1}{2\lambda} \|\boldsymbol{u} - \boldsymbol{x}\|^2 \right\}. \tag{34}$$

The Moreau envelope possesses several fundamental properties (for the proof, please refer to (Rockafellar & Wets, 1998)).

**Lemma C.1** (Properties of Moreau envelope). Let $f$ be proper, closed, and convex. For any $\lambda > 0$, we have:

- (Convergence) $f^\lambda(\boldsymbol{x}) \to f(\boldsymbol{x})$ as $\lambda \to 0^+$;

- (Differentiability) $f^\lambda(\boldsymbol{x})$ is convex and continuously differentiable with gradient

$$\nabla f^\lambda(\boldsymbol{x}) = \frac{1}{\lambda}(\boldsymbol{x} - \text{Prox}_f^\lambda(\boldsymbol{x})); \tag{35}$$

- (Subdifferential) The point $\frac{1}{\lambda}(\boldsymbol{x} - \text{Prox}_f^\lambda(\boldsymbol{x}))$ belongs to the subdifferential of $f(\boldsymbol{u})$, i.e.,

$$\boldsymbol{u} = \text{Prox}_f^\lambda(\boldsymbol{x}) \iff \frac{1}{\lambda}(\boldsymbol{x} - \boldsymbol{u}) \in \partial f(\boldsymbol{u}); \tag{36}$$

- (L-smoothness) $f^\lambda(\boldsymbol{x})$ has $\frac{1}{\lambda}$-Lipschitz continuous gradient.

## C.2. Moreau Approximation

Given a probability density $\pi(\boldsymbol{x}) \propto \exp\{-f(\boldsymbol{x})\}$, its $\lambda$-Moreau approximation is defined as:

$$\pi^\lambda(\boldsymbol{x}) = \frac{1}{Z} \sup_{\boldsymbol{u} \in \mathbb{R}^d} \left\{ \pi(\boldsymbol{u}) \exp\left\{ -\|\boldsymbol{u} - \boldsymbol{x}\|^2/2\lambda \right\} \right\}, \tag{37}$$

where $Z$ is the normalizing constant ensuring that $\pi^\lambda$ integrates to one.

The connection between this density approximation and the Moreau envelope follows directly from the definition in (37):

$$\pi^\lambda(\boldsymbol{x}) \propto \sup_{\boldsymbol{u}} \exp\left\{ -f(\boldsymbol{u}) - \frac{1}{2\lambda}\|\boldsymbol{u} - \boldsymbol{x}\|^2 \right\} \propto \exp\left\{ -\inf_{\boldsymbol{u}} \left\{ f(\boldsymbol{u}) + \frac{1}{2\lambda}\|\boldsymbol{u} - \boldsymbol{x}\|^2 \right\} \right\} \propto \exp\{-f^\lambda(\boldsymbol{x})\}, \tag{38}$$

where the second proportionality follows from the monotonicity of the exponential function, and the last one uses the definition of the Moreau–Yosida regularization. This shows that applying the Moreau approximation to a density $\pi$ is equivalent to applying the Moreau–Yosida regularization to its negative log-density.

**Proposition C.2** (Properties of Moreau approximation). *Let $\pi(\boldsymbol{x}) \propto \exp\{-f(\boldsymbol{x})\}$, where $f : \mathbb{R}^d \to \mathbb{R} \cup \{+\infty\}$ is proper, closed, and convex. For any $\lambda > 0$, we have:*

- (Convergence) $\pi^\lambda(\boldsymbol{x}) \to \pi(\boldsymbol{x})$ point-wise as $\lambda \to 0^+$;

- (Differentiability) $\pi^\lambda(\boldsymbol{x})$ is convex and continuously differentiable with log-gradient

$$\nabla \log \pi^\lambda(\boldsymbol{x}) = \frac{1}{\lambda}(\mathrm{Prox}_f^\lambda(\boldsymbol{x}) - \boldsymbol{x}); \tag{39}$$

- (Subdifferential) The point $\frac{1}{\lambda}(\mathrm{Prox}_f^\lambda(\boldsymbol{x}) - \boldsymbol{x})$ belongs to the subdifferential of $\log \pi(\boldsymbol{u})$, i.e.,

$$\boldsymbol{u} = \mathrm{Prox}_f^\lambda(\boldsymbol{x}) \iff \frac{1}{\lambda}(\boldsymbol{u} - \boldsymbol{x}) \in \partial \log \pi(\boldsymbol{u}); \tag{40}$$

- (L-smoothness) $\pi^\lambda(\boldsymbol{x})$ has $\frac{1}{\lambda}$-Lipschitz continuous log-gradient;

*Proof.* The convergence follows from the fact that the Gaussian kernel $\exp\left\{ -\|\boldsymbol{u} - \boldsymbol{x}\|^2/(2\lambda) \right\}$ converges to a Dirac delta at $\boldsymbol{x}$ as $\lambda \to 0^+$. Consequently,

$$\lim_{\lambda \to 0^+} \pi^\lambda(\boldsymbol{x}) = \pi(\boldsymbol{x}), \quad \forall \boldsymbol{x}.$$

From (38), we have $\pi^\lambda(\boldsymbol{x}) \propto \exp\{-f^\lambda(\boldsymbol{x})\}$. By Lemma C.1, $f^\lambda$ is continuously differentiable with gradient

$$\nabla f^\lambda(\boldsymbol{x}) = \frac{1}{\lambda}\left( \boldsymbol{x} - \mathrm{Prox}_f^\lambda(\boldsymbol{x}) \right).$$

Therefore, $\pi^\lambda(\boldsymbol{x}) = \exp\{-f^\lambda(\boldsymbol{x})\}/Z_\lambda$ is continuously differentiable and its log-gradient

$$\nabla \log \pi^\lambda(\boldsymbol{x}) = -\nabla f^\lambda(\boldsymbol{x}) = \frac{1}{\lambda}\left( \mathrm{Prox}_f^\lambda(\boldsymbol{x}) - \boldsymbol{x} \right).$$

By the optimality condition of the proximal operator, we get

$$\boldsymbol{u} = \mathrm{Prox}_f^\lambda(\boldsymbol{x}) \iff \boldsymbol{0} \in \partial f(\boldsymbol{u}) + \frac{1}{\lambda}(\boldsymbol{u} - \boldsymbol{x}).$$

Since $\log \pi(\boldsymbol{u}) = -f(\boldsymbol{u}) + \mathrm{const}$, we have $\partial \log \pi(\boldsymbol{u}) = -\partial f(\boldsymbol{u})$. Thus,

$$\boldsymbol{u} = \mathrm{Prox}_f^\lambda(\boldsymbol{x}) \iff \frac{1}{\lambda}(\boldsymbol{u} - \boldsymbol{x}) \in -\partial f(\boldsymbol{u}) = \partial \log \pi(\boldsymbol{u}).$$

The gradient of $f^\lambda$ is Lipschitz continuous with constant $1/\lambda$, i.e.,

$$\|\nabla f^\lambda(\boldsymbol{x}) - \nabla f^\lambda(\boldsymbol{y})\| \leq \frac{1}{\lambda}\|\boldsymbol{x} - \boldsymbol{y}\|.$$

Since $\nabla \log \pi^\lambda = -\nabla f^\lambda$, the log-gradient of $\pi^\lambda$ is also $1/\lambda$-Lipschitz.

$\square$

## D. Moreau-Yosida-Gaussian Equivalence

### D.1. Stein Score and Moreau Score

Firstly, we provide a detailed discussion comparing the Stein score $\nabla_{\boldsymbol{x}} \log \pi_t(\boldsymbol{x}_t)$ used in diffusion models and Moreau score $\nabla_{\boldsymbol{x}} \log \pi_t^\lambda$.

In diffusion models, the classic *Stein score* is the gradient of the log-marginal density $\pi_t$ of the forward process. By Tweedie's formula (Efron, 2011), it can be expressed in terms of the posterior mean:

$$\nabla_{\boldsymbol{x}} \log \pi_t(\boldsymbol{x}_t) = \frac{\mu(t)\,\mathbb{E}[\boldsymbol{x}_0|\boldsymbol{x}_t] - \boldsymbol{x}_t}{\sigma^2(t)},$$

where $\mu(t)$ and $\sigma^2(t)$ are the scale and noise variance of the forward transition. In contrast, the proposed *Moreau score* replaces the conditionally expected clean image $\mathbb{E}[\boldsymbol{x}_0|\boldsymbol{x}_t]$ with the proximal operator of the negative log-prior $U$, yielding

$$\nabla_{\boldsymbol{x}} \log \pi_t^\lambda(\boldsymbol{x}_t) = \frac{\mu(t)\,\mathrm{Prox}_U^{\lambda(t)}\!\left(\frac{\boldsymbol{x}_t}{\mu(t)}\right) - \boldsymbol{x}_t}{\sigma^2(t)}.$$

However, the classic Stein score faces two fundamental challenges. First, due to the integral form of $\pi_t$, it does not admit a closed-form expression and therefore can only be learned approximately via denoising score matching in (5). Second, for nonsmooth priors, the Stein score may not be well-defined. In contrast, the Moreau score is given explicitly by a deterministic proximal operator, and it remains well-defined for any proper, closed, convex potential $U$.

From a Bayesian perspective, $\mathbb{E}[\boldsymbol{x}_0|\boldsymbol{x}_t]$ is the posterior mean, whereas the proximal output $\mathrm{Prox}_U^{\lambda(t)}\!\left(\frac{\boldsymbol{x}_t}{\mu(t)}\right)$ is the posterior mode under the forward Gaussian likelihood. Hence the Stein score and the Moreau score differ precisely by the deviation between the posterior mean and the posterior mode. When the potential $U$ is quadratic, the mean and mode of the posterior coincide, and the two scores become exactly equal (Lemma D.1). For general $m-$strongly convex $U$, Proposition D.4 bounds the score discrepancy uniformly. Consequently, the Moreau score provides a principled, analytically tractable approximation to the Stein score, with an error controlled by the curvature of the potential, while naturally handling nonsmooth priors that are inaccessible to the classical formulation.

### D.2. Proof of Lemma 3.1

In this subsection, we prove Lemma 3.1, which shows that convolving a quadratic-density distribution with a Gaussian kernel is equivalent—up to a normalization constant—to applying the Moreau–Yosida regularization to its potential function.

The main idea comes from the similarity between Gaussian convolution and Moreau-Yosida regularization, both of which involving quadratic Gaussian kernels. One key observation is that for quadratic potentials, the integration in convolution can be replaced equivalently by the supremum via Laplace approximation, which typically yields the Moreau-Yosida regularization.

**Lemma D.1** (Moreau–Yosida–Gaussian equivalence)**.** Let $\pi(\boldsymbol{x}) \propto \exp\{-U(\boldsymbol{x})\}$ be a probability density on $\mathbb{R}^d$, where $U(\boldsymbol{x}) = \boldsymbol{x}^\top A\boldsymbol{x}$ with $A \succ 0$. For any $\lambda > 0$, the following two densities that are proportional to each other:

(i) **Moreau approximation:**

$$\pi_t^\lambda(\boldsymbol{x}) \propto \sup_{\boldsymbol{u} \in \mathbb{R}^d} \pi(\boldsymbol{u}) \exp\{-\|\boldsymbol{u} - \boldsymbol{x}\|^2/(2\lambda)\} \propto \exp\{-U^\lambda(\boldsymbol{x})\}. \tag{41}$$

(ii) **Gaussian convolution:**

$$\pi_t(\boldsymbol{x}) \propto (\pi * \mathcal{N}(\mathbf{0}, \lambda I))(\boldsymbol{x}) \propto \int_{\mathbb{R}^d} \pi(\boldsymbol{u}) \exp\{-\|\boldsymbol{u} - \boldsymbol{x}\|^2/(2\lambda)\} \, d\boldsymbol{u}. \tag{42}$$

*Proof.* Since $U$ is quadratic, the function

$$\boldsymbol{u} \mapsto U(\boldsymbol{u}) + \frac{1}{2\lambda}\|\boldsymbol{u} - \boldsymbol{x}\|^2$$

is strictly convex and admits a unique minimizer

$$\boldsymbol{u}^* = \arg\min_{\boldsymbol{u} \in \mathbb{R}^d}\left\{U(\boldsymbol{u}) + \frac{1}{2\lambda}\|\boldsymbol{u} - \boldsymbol{x}\|^2\right\}.$$

The first-order optimality condition yields

$$\nabla U(\boldsymbol{u}^*) + \frac{1}{\lambda}(\boldsymbol{u}^* - \boldsymbol{x}) = \mathbf{0}.$$

Consider the convolution integral in (42). Using $\pi(\boldsymbol{u}) \propto \exp\{-U(\boldsymbol{u})\}$, we have

$$\pi_t(\boldsymbol{x}) \propto \int_{\mathbb{R}^d} \exp\left\{-\left[U(\boldsymbol{u}) + \frac{1}{2\lambda}\|\boldsymbol{u} - \boldsymbol{x}\|^2\right]\right\} d\boldsymbol{u}. \tag{43}$$

Because $U$ is quadratic, the exponent is itself a quadratic function of $\boldsymbol{u}$ and can be written exactly as

$$U(\boldsymbol{u}) + \frac{1}{2\lambda}\|\boldsymbol{u} - \boldsymbol{x}\|^2 = U(\boldsymbol{u}^*) + \frac{1}{2\lambda}\|\boldsymbol{u}^* - \boldsymbol{x}\|^2 + (\boldsymbol{u} - \boldsymbol{u}^*)^\top(A + \lambda^{-1}I)(\boldsymbol{u} - \boldsymbol{u}^*).$$

Substituting it into (43) gives

$$\pi_t(\boldsymbol{x}) \propto \exp\left\{-\left[U(\boldsymbol{u}^*) + \frac{1}{2\lambda}\|\boldsymbol{u}^* - \boldsymbol{x}\|^2\right]\right\} \int_{\mathbb{R}^d} \exp\{-(\boldsymbol{u} - \boldsymbol{u}^*)^\top(A + \lambda^{-1}I)(\boldsymbol{u} - \boldsymbol{u}^*)\} \, d\boldsymbol{u}.$$

The remaining integral is Gaussian and depends only on $A$ and $\lambda$, hence it contributes a multiplicative constant that does not depend on $\boldsymbol{x}$. Therefore,

$$\pi_t(\boldsymbol{x}) \propto \exp\left\{-\inf_{\boldsymbol{u} \in \mathbb{R}^d}\left[U(\boldsymbol{u}) + \frac{1}{2\lambda}\|\boldsymbol{u} - \boldsymbol{x}\|^2\right]\right\}.$$

By the definition of the Moreau–Yosida envelope,

$$U^\lambda(\boldsymbol{x}) = \inf_{\boldsymbol{u} \in \mathbb{R}^d}\left\{U(\boldsymbol{u}) + \frac{1}{2\lambda}\|\boldsymbol{u} - \boldsymbol{x}\|^2\right\},$$

which, together with (41), implies

$$\pi_t(\boldsymbol{x}) \propto \exp\{-U^\lambda(\boldsymbol{x})\} \propto \pi_t^\lambda(\boldsymbol{x}).$$

This proves the claim. $\qquad\square$

This lemma reveals a fundamental duality between variational and probabilistic smoothing techniques for Gaussian measures. The Moreau–Yosida regularization, originating from nonsmooth optimization, performs a pointwise supremum that yields a regularized potential $U^\lambda$. Gaussian convolution, a standard probabilistic operation, smooths the density by integrating with a Gaussian kernel. This result provides a bridge between optimization and diffusion processes, enabling a new perspective for nonsmooth sampling.

### D.3. Generalized Moreau Approximation

In this subsection, we also derive the Moreau-Yosida-Gaussian equivalence in the general context of diffusion models. Consider the forward diffusion process:

$$d\boldsymbol{x}_t^{\rightarrow} = \{a(t)\boldsymbol{x}_t^{\rightarrow}\}\, dt + b(t)d\boldsymbol{B}_t, \quad t \in [0, T], \tag{44}$$

where the marginal

$$\pi_t(\boldsymbol{x}_t) \propto \int_{\mathbb{R}^d} \pi(\boldsymbol{x}_0) \exp\left\{-\frac{\|\mu(t)\boldsymbol{x}_0 - \boldsymbol{x}_t\|^2}{2\sigma^2(t)}\right\} d\boldsymbol{x}_0, \tag{45}$$

with

$$\mu(t) = \exp\left\{\int_0^t a(s)ds\right\}, \quad \sigma^2(t) = \int_0^t b^2(s)\exp\left\{2\int_s^t a(r)dr\right\} ds.$$

The reverse process is also a diffusion process, running backwards in time and given by the reverse-time SDE:

$$d\boldsymbol{x}_t^{\leftarrow} = \{a(t)\boldsymbol{x}_t^{\leftarrow} - b^2(t)\nabla \log \pi_t(\boldsymbol{x}_t^{\leftarrow})\}\, dt + b(t)d\bar{\boldsymbol{B}}_t. \tag{46}$$

Define the time-varying regularization parameter

$$\lambda(t) = \sigma^2(t)/\mu^2(t) = \int_0^t b^2(s)/\mu^2(s)ds, \tag{47}$$

the Moreau approximation is given by

$$\pi_t^\lambda(\boldsymbol{x}_t) \propto \exp\left\{-U^{\lambda(t)}\left(\frac{\boldsymbol{x}_t}{\mu(t)}\right)\right\}. \tag{48}$$

Using (39), the Moreau score is obtained by

$$\nabla_{\boldsymbol{x}} \log \pi_t^\lambda(\boldsymbol{x}_t) = \frac{\mu(t)\text{Prox}_U^{\lambda(t)}\left(\frac{\boldsymbol{x}_t}{\mu(t)}\right) - \boldsymbol{x}_t}{\sigma^2(t)}. \tag{49}$$

Compared with the original Moreau score in (17), the generalized expression in (49) involves a rescaling of the proximal input by $1/\mu(t)$ and substitutes the denominator with $\sigma^2(t)$.

**Proposition D.2.** Let $\pi \propto \exp\{-U(\boldsymbol{x})\}$, where $U$ is a positive semidefinite quadratic function. Then the posterior score $\nabla_{\boldsymbol{x}} \log \pi_t(\boldsymbol{x}_t)$ based on (45) coincides with the Moreau score in (49), namely,

$$\nabla_{\boldsymbol{x}} \log \pi_t(\boldsymbol{x}_t) = \nabla_{\boldsymbol{x}} \log \pi_t^\lambda(\boldsymbol{x}_t).$$

*Proof.* Define the scaled energy

$$\tilde{U}_t(\boldsymbol{y}) = U\left(\frac{\boldsymbol{y}}{\mu(t)}\right).$$

Let $\boldsymbol{y} = \mu(t)\boldsymbol{x}_0$. Then $d\boldsymbol{y} = \mu(t)d\boldsymbol{x}_0$, and the target density becomes

$$\pi(\boldsymbol{x}_0) \propto \exp\{-U(\boldsymbol{x}_0)\} = \exp\left\{-U\left(\frac{\boldsymbol{y}}{\mu(t)}\right)\right\} = \exp\{-\tilde{U}_t(\boldsymbol{y})\}.$$

From the forward noising model (45), the marginal is given by

$$\pi_t(\boldsymbol{x}_t) \propto \int \exp\{-\tilde{U}_t(\boldsymbol{y})\} \frac{1}{(2\pi\sigma^2(t))^{d/2}} \exp\left\{-\frac{\|\boldsymbol{x}_t - \boldsymbol{y}\|^2}{2\sigma^2(t)}\right\} d\boldsymbol{y}. \tag{50}$$

Hence $\pi_t$ is obtained by convolving $\exp\{-\tilde{U}_t\}$ with the Gaussian kernel $\mathcal{N}(0, \sigma^2(t)I)$. By Proposition D.1, such a Gaussian convolution is equivalent to the Moreau–Yosida regularization of $\tilde{U}_t$ with parameter $\sigma^2(t)$, namely,

$$\pi_t(\boldsymbol{x}_t) \propto \exp\left\{-\tilde{U}_t^{\sigma^2(t)}(\boldsymbol{x}_t)\right\}.$$

We now relate this envelope to that of the original function $U$. For any $\mu > 0$ and $\lambda > 0$, if $h(\boldsymbol{y}) = U(\boldsymbol{y}/\mu)$, then the scaling property of the Moreau envelope yields

$$h^\lambda(\boldsymbol{x}) = U^{\lambda/\mu^2}\left(\frac{\boldsymbol{x}}{\mu}\right).$$

Applying this identity with $h = \tilde{U}_t$, $\mu = \mu(t)$, and $\lambda = \sigma^2(t)$ gives

$$\tilde{U}_t^{\sigma^2(t)}(\boldsymbol{x}_t) = U^{\sigma^2(t)/\mu(t)^2}\left(\frac{\boldsymbol{x}_t}{\mu(t)}\right) = U^{\lambda(t)}\left(\frac{\boldsymbol{x}_t}{\mu(t)}\right). \tag{51}$$

Substituting (51) into the previous expression (50) for $\pi_t$ yields

$$\pi_t(\boldsymbol{x}_t) \propto \exp\left\{-U^{\lambda(t)}\left(\frac{\boldsymbol{x}_t}{\mu(t)}\right)\right\} \propto \pi_t^\lambda(\boldsymbol{x}_t),$$

which implies equality of their score functions. $\qquad\square$

### D.4. Asymptotic Moreau-Yosida-Gaussian Equivalence

In this subsection, we provide some asymptotic error bounds between the true score and the Moreau score under non-quadratic settings.

With Tweedie's formula (Efron, 2011), we have

$$\nabla_{\boldsymbol{x}} \log \pi_t(\boldsymbol{x}_t) = \frac{\mu(t)\mathbb{E}\left[\boldsymbol{x}_0|\boldsymbol{x}_t, \boldsymbol{y}\right] - \boldsymbol{x}_t}{\sigma^2(t)}, \tag{52}$$

where $\mathbb{E}\left[\boldsymbol{x}_0|\boldsymbol{x}_t, \boldsymbol{y}\right]$ is the posterior mean. Recall that the Moreau score is given by

$$\nabla_{\boldsymbol{x}} \log \pi_t^\lambda(\boldsymbol{x}_t) = \frac{\mu(t)\mathrm{Prox}_U^{\sigma^2(t)/\mu^2(t)}\left(\frac{\boldsymbol{x}_t}{\mu(t)}\right) - \boldsymbol{x}_t}{\sigma^2(t)}. \tag{53}$$

Similar to (19), we can see that the proximal operator

$$\mathrm{Prox}_U^{\sigma^2(t)/\mu^2(t)}\left(\frac{\boldsymbol{x}_t}{\mu(t)}\right) = \arg\max_{\boldsymbol{x}_0 \in \mathbb{R}^d}\exp\left\{-U(\boldsymbol{x}_0) - \frac{\mu^2(t)\|\frac{\boldsymbol{x}_t}{\mu(t)} - \boldsymbol{x}_0\|^2}{2\sigma^2(t)}\right\} = \arg\max_{\boldsymbol{x}_0 \in \mathbb{R}^d} p(\boldsymbol{x}_0|\boldsymbol{x}_t, \boldsymbol{y})$$

is the posterior mode. This reveals the connection between the true score and the Moreau score, and the difference can be characterized by the deviation between the posterior mean $\mathbb{E}\left[\boldsymbol{x}_0|\boldsymbol{x}_t, \boldsymbol{y}\right]$ and mode $\mathrm{Prox}_U^{\sigma^2(t)/\mu^2(t)}(\frac{\boldsymbol{x}_t}{\mu(t)})$.

**Lemma D.3** (Deviation between mean and mode). Let $\pi(\boldsymbol{x}) \propto \exp\{-U(\boldsymbol{x})\}$ be a probability distribution on $\mathbb{R}^d$, with a proper, closed, convex function $U$. Denote the mean of $\pi(\boldsymbol{x})$ as

$$\boldsymbol{\mu} = \mathbb{E}_{\boldsymbol{x}\sim\pi}\left[\boldsymbol{x}\right] = \int_{\mathbb{R}^d}\boldsymbol{x}\pi(\boldsymbol{x})d\boldsymbol{x}, \tag{54}$$

and the mode

$$\boldsymbol{u}^* = \arg\max_{\boldsymbol{u}\in\mathbb{R}^d}\{\pi(\boldsymbol{u})\}. \tag{55}$$

Then the following hold.

1. If $\pi(\boldsymbol{x})$ is symmetric unimodal, e.g., $U$ is symmetric convex, then we have

$$\boldsymbol{\mu} = \boldsymbol{u}^*.$$

2. (Stein, 1981) If $U$ is $m$-strongly convex, then we have

$$\|\boldsymbol{\mu} - \boldsymbol{u}^*\| \leq \sqrt{\frac{2d}{m}}.$$

The following proposition generalizes Lemma 3.1 to non-quadratic potentials and bounds the error between the true score and the Moreau score, offering a general asymptotic analysis for the Moreau-Yosida-Gaussian equivalence and the proof of Proposition 3.2.

**Proposition D.4** (Asymptotic Moreau-Yosida-Gaussian equivalence). Consider the diffusion process in (44) with $x_0 \sim \pi(x) \propto \exp\{-U(x)\}$ and

$$\pi_t(x_t) = \int_{\mathbb{R}^d} \pi(x_0) p_{0t}(x_t | x_0) dx_0,$$

where $U = \beta f_y + g$ with $f_y$ being $m$-strongly convex and $g$ being convex. Let $\mu(t)$, $\sigma^2(t)$, and $\lambda(t)$ be as defined in Section 3. Then, for any $t > 0$,

$$\|\nabla \log \pi_t(x_t) - \nabla \log \pi_t^\lambda(x_t)\| \leq \frac{\mu(t)}{\sigma(t)} \sqrt{\frac{2d}{\beta m \sigma^2(t) + \mu^2(t)}}. \tag{56}$$

In particular, as $\beta \to \infty$, we have

$$\|\nabla_{x_t} \log \pi_t(x_t) - \nabla_{x_t} \log \pi_t^\lambda(x_t)\| \to 0.$$

*Proof.* By Tweedie's formula, we have

$$\nabla \log \pi_t(x_t) = (\mu(t) \mathbb{E}[x_0 | x_t, y] - x_t) / \sigma^2(t).$$

Moreover, recall the Moreau score

$$\nabla \log \pi_t^\lambda(x_t) = (\mu(t) \text{Prox}_U^{\lambda(t)}(x_t / \mu(t)) - x_t) / \sigma^2(t).$$

Thus,

$$\|\nabla \log \pi_t(x_t) - \nabla \log \pi_t^\lambda(x_t)\| = \frac{\mu(t)}{\sigma^2(t)} \|\mathbb{E}[x_0 | x_t, y] - \text{Prox}_U^{\lambda(t)}(x_t / \mu(t))\|. \tag{57}$$

Note that $\text{Prox}_U^{\lambda(t)}(x_t / \mu(t))$ is the mode of the posterior $p(x_0 | x_t, y)$, from Bayes' rule, we get

$$p(x_0 | x_t, y) \propto p(x_t | x_0, y) p(x_0 | y) \propto p(x_t | x_0) \pi(x_0) \propto \exp\left\{-U(x_0) - \frac{\|x_t - \mu(t) x_0\|^2}{2\sigma^2(t)}\right\}.$$

Denote $c(x) = U(x) + \frac{\|x_t - \mu(t)x\|^2}{2\sigma^2(t)}$. Then $c(x)$ is $(\beta m + \frac{\mu^2(t)}{\sigma^2(t)})$-strongly convex. Denote

$$\mu = \mathbb{E}[x_0 | x_t, y], \quad u^* = \arg\max_{x_0 \in \mathbb{R}^d} p(x_0 | x_t, y).$$

Then from Lemma D.3, we have

$$\|\mu - u^*\| \leq \sqrt{\frac{2d}{\beta m + \frac{\mu^2(t)}{\sigma^2(t)}}} = \sigma(t) \sqrt{\frac{2d}{\beta m \sigma^2(t) + \mu^2(t)}}.$$

Substituting the above into the score error (57), we have

$$\|\nabla_{x_t} \log \pi_t(x_t) - \nabla_{x_t} \log \pi_t^\lambda(x_t)\| = \frac{\mu(t)}{\sigma^2(t)} \|\mu - u^*\| \leq \frac{\mu(t)}{\sigma(t)} \sqrt{\frac{2d}{\beta m \sigma^2(t) + \mu^2(t)}},$$

or equivalently

$$\|\nabla_{x_t} \log \pi_t(x_t) - \nabla_{x_t} \log \pi_t^\lambda(x_t)\| \leq \frac{1}{\mu(t)} \sqrt{\frac{2d}{\beta m \lambda^2(t) + \lambda(t)}}.$$

Further, as $\beta \to \infty$, we have the error bound tends to 0 with rate $\mathcal{O}(\beta^{-1/2})$.

$\square$

This proposition demonstrates that the Moreau–Yosida approximation serves as a provably accurate surrogate for the true score in diffusion processes when the potential $U$ contains a strongly convex likelihood term. The error bound quantifies how the discrepancy depends on the diffusion parameters $\mu(t)$, $\sigma(t)$, the strong convexity constant $m$, and the likelihood weight $\beta$. Crucially, as $\beta \to \infty$, the bound tends to zero, revealing an asymptotic equivalence between the traditional score and the Moreau score.

# E. Moreau Score Matching

## E.1. Learning of the Proximal Operator

If we have realizations $\left\{\left(\boldsymbol{x}^{(i)}, \mathrm{Prox}_g^{\lambda(t)}\left(\boldsymbol{x}^{(i)}\right)\right)\right\}$, then we can use a network to learn the proximal mapping through data. Unfortunately, paired ground-truth do not exist in common settings, making supervised training infeasible. Instead, we seek to train a proximal network using only i.i.d. samples from the unknown data distribution $\pi(\boldsymbol{x}) \propto \exp\{-g(\boldsymbol{x})\}$ in an unsupervised way (Fang et al., 2023; 2026).

The key insight comes from the observation that the proximal operator is the maximum a posteriori (MAP) denoiser for additive Gaussian noise. For $\boldsymbol{x}_0 \sim p_0(\boldsymbol{x}) \propto \exp\{-g(\boldsymbol{x}_0)\}$ and $\boldsymbol{x}_t = \boldsymbol{x}_0 + \sqrt{\lambda}\boldsymbol{\xi}_t$ with $\boldsymbol{\xi}_t \sim \mathcal{N}(\mathbf{0}, I)$, we have $\mathrm{Prox}_g^\lambda(\boldsymbol{x}_t) = \arg\max_{\boldsymbol{x}_0} p(\boldsymbol{x}_0|\boldsymbol{x}_t)$. Consider a loss function of the form $\mathbb{E}_{\boldsymbol{x}_0, \boldsymbol{x}_t}[\mathrm{dist}(\boldsymbol{x}_0, \phi(\boldsymbol{x}_t))]$. If we use squared $\ell_2$ distance, minimizing this loss will lead to the minimum mean square error (MMSE) estimator given by the mean of the posterior $p(\boldsymbol{x}_0|\boldsymbol{x}_t)$. However, our goal is to learn the mode of the posterior.

The core idea is to design a distance metric $\mathrm{dist}_\zeta$ that, in the limit $\zeta \to 0$, it recovers the posterior mode (which equals the proximal operator) as the minimizer of the expected loss. A key observation is that as $\zeta \to 0$, the Gaussian kernel $\mathcal{N}(\cdot; \mathbf{0}, \zeta^2 I)$ behaves like a Dirac delta function, which can be designed to forces $\phi$ to output the posterior mode for each $\boldsymbol{x}_t$. In the following proposition, we will extend the proximal matching technique in (Fang et al., 2023) to general diffusion processes, thereby obtaining a training loss to learn the proximal operator.

**Proposition E.1.** Suppose that $U(\boldsymbol{x}) = \beta f_{\boldsymbol{y}}(\boldsymbol{x}) + g(\boldsymbol{x})$, where $f : \mathbb{R}^d \to \mathbb{R}$ is $L$-smooth and $m$-strongly convex, $g : \mathbb{R}^d \to \mathbb{R}$ is convex and $\mathrm{dom}(g) = \mathcal{X}$ is compact, and $\beta > 0$. Denote $\boldsymbol{x}_0 \sim p_0(\boldsymbol{x}) \propto \exp\{-g(\boldsymbol{x}_0)\}$ and $\boldsymbol{x}_t = \boldsymbol{x}_0 + \sqrt{\lambda(t)}\boldsymbol{\xi}_t$ where $\boldsymbol{\xi}_t \sim \mathcal{N}(\mathbf{0}, I)$. Let $\mathrm{dist}_\zeta(\boldsymbol{x}, \boldsymbol{x}') : \mathbb{R}^d \times \mathbb{R}^d \to \mathbb{R}$ be defined as

$$\mathrm{dist}_\zeta(\boldsymbol{x}, \boldsymbol{x}') = 1 - \mathcal{N}(\boldsymbol{x} - \boldsymbol{x}'; \mathbf{0}, \zeta^2 I), \quad \zeta > 0, \tag{58}$$

where $\mathcal{N}(\boldsymbol{x} - \boldsymbol{x}'; \mathbf{0}, \zeta^2 I) = \frac{1}{(2\pi\zeta^2)^{d/2}} \exp\left\{-\frac{\|\boldsymbol{x} - \boldsymbol{x}'\|_2^2}{2\zeta^2}\right\}$. Consider the optimization problem

$$\boldsymbol{\theta}^* = \arg\min_{\boldsymbol{\theta}} \lim_{\zeta \to 0} \mathbb{E}_{t \sim \mathcal{U}[0,T]} \mathbb{E}_{\boldsymbol{x}_0 \sim p_0} \mathbb{E}_{\boldsymbol{x}_t|\boldsymbol{x}_0 \sim \mathcal{N}(\mathbf{0}, \lambda(t)I)} \left[\mathrm{dist}_\zeta\left(\phi_{\boldsymbol{\theta}}(\boldsymbol{x}_t, \lambda(t)), \boldsymbol{x}_0\right)\right]. \tag{59}$$

Then, almost surely (i.e., for almost every $\boldsymbol{x}_t$, $t$),

$$\phi_{\boldsymbol{\theta}^*}(\boldsymbol{x}_t, \lambda(t)) = \arg\max_{\boldsymbol{x}_0 \in \mathbb{R}^d} p(\boldsymbol{x}_0|\boldsymbol{x}_t) = \mathrm{Prox}_g^{\lambda(t)}(\boldsymbol{x}_t). \tag{60}$$

*Proof.* From the definition of $\mathrm{dist}_\zeta(\boldsymbol{x}, \boldsymbol{x}')$, we have

$$\lim_{\zeta \to 0} \mathbb{E}_{t, \boldsymbol{x}_0, \boldsymbol{x}_t}[\mathrm{dist}_\zeta(\phi_{\boldsymbol{\theta}}(\boldsymbol{x}_t, \lambda(t)), \boldsymbol{x}_0)] = 1 - \lim_{\zeta \to 0} \mathbb{E}_{t, \boldsymbol{x}_0, \boldsymbol{x}_t}\left[\mathcal{N}(\phi_{\boldsymbol{\theta}}(\boldsymbol{x}_t, \lambda(t)) - \boldsymbol{x}_0; \mathbf{0}, \zeta^2 I)\right].$$

Using the law of the total expectation, we have

$$\mathbb{E}_{t, \boldsymbol{x}_0, \boldsymbol{x}_t}\left[\mathcal{N}(\phi_{\boldsymbol{\theta}}(\boldsymbol{x}_t, \lambda(t)) - \boldsymbol{x}_0; \mathbf{0}, \zeta^2 I)\right] = \mathbb{E}_t \mathbb{E}_{\boldsymbol{x}_t|t} \mathbb{E}_{\boldsymbol{x}_0|\boldsymbol{x}_t, t}\left[\mathcal{N}(\phi_{\boldsymbol{\theta}}(\boldsymbol{x}_t, \lambda(t)) - \boldsymbol{x}_0; \mathbf{0}, \zeta^2 I)\right].$$

We now analyze the inner conditional expectation. For fixed $\boldsymbol{x}_t, t$, define the function

$$F_\zeta(\boldsymbol{c}) = \mathbb{E}_{\boldsymbol{x}_0|\boldsymbol{x}_t, t}\left[\mathcal{N}(\boldsymbol{c} - \boldsymbol{x}_0; \mathbf{0}, \zeta^2 I)\right] = \int_{\mathbb{R}^d} \mathcal{N}(\boldsymbol{c} - \boldsymbol{x}_0; \mathbf{0}, \zeta^2 I) p(\boldsymbol{x}_0|\boldsymbol{x}_t, t) \, d\boldsymbol{x}_0.$$

Since $p(\boldsymbol{x}_0|\boldsymbol{x}_t, t) \propto p_0(\boldsymbol{x}_0) \exp\{-\|\boldsymbol{x}_t - \boldsymbol{x}_0\|^2/(2\lambda(t))\}$ and $p_0$ is bounded, the product is integrable. Moreover, as $\zeta \to 0$, the Gaussian kernel $\mathcal{N}(\boldsymbol{c} - \boldsymbol{x}_0; \mathbf{0}, \zeta^2 I)$ tends to Dirac $\delta_{\boldsymbol{c}}(\boldsymbol{x}_0)$. Then for almost every $\boldsymbol{x}_t$ and for each $\boldsymbol{c}$, there holds

$$\lim_{\zeta \to 0} F_\zeta(\boldsymbol{c}) = p(\boldsymbol{c}|\boldsymbol{x}_t, t).$$

In particular, for $\boldsymbol{c} = \phi_{\boldsymbol{\theta}}(\boldsymbol{x}_t, \lambda(t))$, we have

$$\lim_{\zeta \to 0} \mathbb{E}_{\boldsymbol{x}_0|\boldsymbol{x}_t, t}\left[\mathcal{N}(\phi_{\boldsymbol{\theta}}(\boldsymbol{x}_t, \lambda(t)) - \boldsymbol{x}_0; \mathbf{0}, \zeta^2 I)\right] = p(\phi_{\boldsymbol{\theta}}(\boldsymbol{x}_t, \lambda(t))|\boldsymbol{x}_t, t).$$

Since $p_0$ is supported on a compact set $\mathcal{X}$, it follows that $\mathcal{N}(\boldsymbol{x} - \boldsymbol{x}'; \mathbf{0}, \zeta^2 I)$ is bounded by its maximum at $\mathbf{0}$, and $p(\boldsymbol{x}_0|\boldsymbol{x}_t, t)$ is bounded. Hence, the integrand is uniformly bounded by an integrable function. By the dominated convergence theorem, we have

$$\lim_{\zeta \to 0} \mathbb{E}_{\boldsymbol{x}_t, t} \mathbb{E}_{\boldsymbol{x}_0|\boldsymbol{x}_t, t} \left[ \mathcal{N}(\boldsymbol{\phi}_{\boldsymbol{\theta}}(\boldsymbol{x}_t, \lambda(t)) - \boldsymbol{x}_0; \mathbf{0}, \zeta^2 I) \right] = \mathbb{E}_{\boldsymbol{x}_t, t} \left[ \lim_{\zeta \to 0} \mathbb{E}_{\boldsymbol{x}_0|\boldsymbol{x}_t, t} \left[ \mathcal{N}(\boldsymbol{\phi}_{\boldsymbol{\theta}}(\boldsymbol{x}_t, \lambda(t)) - \boldsymbol{x}_0; \mathbf{0}, \zeta^2 I) \right] \right].$$

Combining the above results, we obtain

$$\lim_{\zeta \to 0} \mathbb{E}_{t, \boldsymbol{x}_0, \boldsymbol{x}_t} \left[ \text{dist}_\zeta(\boldsymbol{\phi}_{\boldsymbol{\theta}}(\boldsymbol{x}_t, \lambda(t)), \boldsymbol{x}_0) \right] = 1 - \mathbb{E}_{t, \boldsymbol{x}_t} \left[ p(\boldsymbol{\phi}_{\boldsymbol{\theta}}(\boldsymbol{x}_t, \lambda(t))|\boldsymbol{x}_t, t) \right].$$

Minimizing the left-hand side over measurable functions $\boldsymbol{\phi}_{\boldsymbol{\theta}}$ is equivalent to maximizing $\mathbb{E}_{t, \boldsymbol{x}_t} \left[ p(\boldsymbol{\phi}_{\boldsymbol{\theta}}(\boldsymbol{x}_t, \lambda(t))|\boldsymbol{x}_t, t) \right]$. Since the expectation is over $\boldsymbol{x}_t, t$ and $\boldsymbol{\phi}_{\boldsymbol{\theta}}(\boldsymbol{x}_t, \lambda(t))$ can be chosen pointwise, the optimum is achieved by choosing, for each $\boldsymbol{x}_t, t$, $\boldsymbol{\phi}_{\boldsymbol{\theta}}(\boldsymbol{x}_t, \lambda(t))$ to maximize $p(\boldsymbol{\phi}_{\boldsymbol{\theta}}(\boldsymbol{x}_t, \lambda(t))|\boldsymbol{x}_t, t)$. Therefore,

$$\boldsymbol{\phi}_{\boldsymbol{\theta}^*}(\boldsymbol{x}_t, \lambda(t)) = \arg \max_{\boldsymbol{x}_0 \in \mathbb{R}^d} p(\boldsymbol{x}_0|\boldsymbol{x}_t, t).$$

Recall that in Section 3, we give the equivalence between the posterior mode and proximal operator, i.e., $\arg \max_{\boldsymbol{x}_0 \in \mathbb{R}^d} p(\boldsymbol{x}_0|\boldsymbol{x}_t, t) = \text{Prox}_g^{\lambda(t)}(\boldsymbol{x}_t)$. Then we obtain that

$$\boldsymbol{\phi}_{\boldsymbol{\theta}^*}(\boldsymbol{x}_t, \lambda(t)) = \text{Prox}_g^{\lambda(t)}(\boldsymbol{x}_t).$$

This completes the proof. $\qquad \square$

This proposition provides a principled training objective for learning proximal operators in an unsupervised manner, offering a theoretical foundation for using neural networks to approximate proximal operators in iterative denoising algorithms.

### E.2. Score Estimation Error

In this subsection, we provide a detailed characterization for the score estimation error, which will be used in the following convergence analysis. The score estimation error is mainly composed of two components: proximal approximation error and deviation between posterior mean and mode.

Note that in practice, obtaining realizations from the composite distribution is intractable. Here we assume that only realizations from $\exp\{-g(\boldsymbol{x})\}$ can be obtained. As a consequence, we use the proximal splitting:

$$\text{Prox}_U^{\lambda(t)}(\boldsymbol{x}) \approx \text{Prox}_g^{\lambda(t)}(\boldsymbol{x} - \lambda(t)\beta \nabla_{\boldsymbol{x}} f(\boldsymbol{x})). \tag{61}$$

Next, we give the detailed error bound for this first-order approximation.

**Proposition E.2** (Proximal splitting). Suppose Assumption 3.3 holds. For any $\boldsymbol{x} \in \mathcal{X}$, define

$$\boldsymbol{u}^*(\boldsymbol{x}) = \text{Prox}_{\beta f + g}^{\lambda}(\boldsymbol{x}) = \arg \min_{\boldsymbol{u} \in \mathbb{R}^d} \left\{ \beta f(\boldsymbol{u}) + g(\boldsymbol{u}) + \frac{1}{2\lambda} \|\boldsymbol{u} - \boldsymbol{x}\|^2 \right\},$$
$$\hat{\boldsymbol{u}}^*(\boldsymbol{x}) = \text{Prox}_g^{\lambda}(\boldsymbol{x} - \lambda\beta\nabla f(\boldsymbol{x})) = \arg \min_{\boldsymbol{u} \in \mathbb{R}^d} \left\{ g(\boldsymbol{u}) + \frac{1}{2\lambda} \|\boldsymbol{u} - (\boldsymbol{x} - \lambda\beta\nabla f(\boldsymbol{x}))\|^2 \right\}. \tag{62}$$

Then, for $\beta \geq 2$ and sufficiently small $\lambda$, we have

$$\|\boldsymbol{u}^*(\boldsymbol{x}) - \hat{\boldsymbol{u}}^*(\boldsymbol{x})\| \leq \frac{4}{\beta L} \left( \|\nabla f(\boldsymbol{x})\| + 1 \right). \tag{63}$$

In particular, as $\beta \to \infty$, the error bound tends to 0 at a rate of $\mathcal{O}(\beta^{-1})$.

*Proof.* By the first-order optimality condition for $\boldsymbol{u}^*$, there exists $\boldsymbol{\eta} \in \partial g(\boldsymbol{u}^*)$ such that

$$\beta \nabla f(\boldsymbol{u}^*) + \boldsymbol{\eta} + \frac{1}{\lambda}(\boldsymbol{u}^* - \boldsymbol{x}) = \mathbf{0},$$

which implies

$$\boldsymbol{u}^* - \boldsymbol{x} = -\lambda\beta\nabla f(\boldsymbol{u}^*) - \lambda\boldsymbol{\eta}. \tag{64}$$

Equivalently, $\boldsymbol{x} - \lambda\beta\nabla f(\boldsymbol{u}^*) - \boldsymbol{u}^* \in \lambda\partial g(\boldsymbol{u}^*)$, and by the characterization of proximal maps in (36), we get

$$\boldsymbol{u}^* = \mathrm{Prox}_g^\lambda(\boldsymbol{x} - \lambda\beta\nabla f(\boldsymbol{u}^*)).$$

On the other hand, by definition,

$$\hat{\boldsymbol{u}}^* = \mathrm{Prox}_g^\lambda(\boldsymbol{x} - \lambda\beta\nabla f(\boldsymbol{x})).$$

Since the proximal operator $\mathrm{Prox}_g^\lambda$ is nonexpansive, we have

$$\|\boldsymbol{u}^* - \hat{\boldsymbol{u}}^*\| \le \|(\boldsymbol{x} - \lambda\beta\nabla f(\boldsymbol{u}^*)) - (\boldsymbol{x} - \lambda\beta\nabla f(\boldsymbol{x}))\| = \lambda\beta\|\nabla f(\boldsymbol{u}^*) - \nabla f(\boldsymbol{x})\|. \tag{65}$$

Using the $L$-smoothness of $f$, we obtain

$$\|\boldsymbol{u}^* - \hat{\boldsymbol{u}}^*\| \le \lambda\beta L\|\boldsymbol{u}^* - \boldsymbol{x}\|. \tag{66}$$

We now bound $\|\boldsymbol{u}^* - \boldsymbol{x}\|$. Taking norms in (64) yields

$$\|\boldsymbol{u}^* - \boldsymbol{x}\| \le \lambda\beta\|\nabla f(\boldsymbol{u}^*)\| + \lambda\|\boldsymbol{\eta}\|. \tag{67}$$

Since $g : \mathbb{R}^d \to \mathbb{R}$ is convex with a compact domain $\mathrm{dom}\, g = \mathcal{X}$, there exists a bounded constant $L_g$ such that $g$ is $L_g$-Lipschitz on $\mathcal{X}$. For sufficiently large $\beta \ge \max\{2, L_g\}$, we have $\|\boldsymbol{\eta}\| \le \beta$. Moreover, by the $L$-smoothness of $f$, we obtain

$$\|\nabla f(\boldsymbol{u}^*)\| \le \|\nabla f(\boldsymbol{x})\| + L\|\boldsymbol{u}^* - \boldsymbol{x}\|. \tag{68}$$

Combining (67) and (68), we get

$$\|\boldsymbol{u}^* - \boldsymbol{x}\| \le \lambda\beta\left(\|\nabla f(\boldsymbol{x})\| + L\|\boldsymbol{u}^* - \boldsymbol{x}\|\right) + \lambda\beta.$$

Rearranging gives

$$(1 - \lambda\beta L)\|\boldsymbol{u}^* - \boldsymbol{x}\| \le \lambda\beta\left(\|\nabla f(\boldsymbol{x})\| + 1\right).$$

For $\lambda\beta L < 1$, we get

$$\|\boldsymbol{u}^* - \boldsymbol{x}\| \le \frac{\lambda\beta}{1 - \lambda\beta L}\left(\|\nabla f(\boldsymbol{x})\| + 1\right). \tag{69}$$

Substituting (69) into (66) gives

$$\|\boldsymbol{u}^* - \hat{\boldsymbol{u}}^*\| \le \frac{\lambda^2\beta^2 L}{1 - \lambda\beta L}\left(\|\nabla f(\boldsymbol{x})\| + 1\right).$$

Choosing $\lambda \le (2\beta^{3/2}L)^{-1}$ ensures $1 - \lambda\beta L \ge 1/2$, from which it follows that

$$\|\boldsymbol{u}^* - \hat{\boldsymbol{u}}^*\| \le \frac{4}{\beta L}\left(\|\nabla f(\boldsymbol{x})\| + 1\right).$$

The stated $\mathcal{O}(\beta^{-1})$ rate is immediate. $\qquad\square$

Then, we can combine the above results and obtain a total score estimation error bound, which corresponds to the results in Proposition 4.2. The proof employs a two-step error decomposition: first, between the true score and the Moreau score $\nabla\log\pi_t^\lambda(\boldsymbol{x}_t)$; second, between the Moreau score and the network approximation $\boldsymbol{v}_t$. The first term is bounded using Proposition D.4, which exploits the strong convexity of the posterior to control the distance between the posterior mean and mode. The second term is bounded by the proximal splitting approximation in Proposition E.2, which leverages the composite structure $U = \beta f + g$ to approximate the full proximal operator.

**Theorem E.3** (Score estimation error). Under Assumption 3.3, consider a given distribution $\pi(\boldsymbol{x}) \propto \exp\{-U(\boldsymbol{x})\} \propto \exp\{-(\beta f_{\boldsymbol{y}} + g)(\boldsymbol{x})\}$ with $\boldsymbol{x}$ and $\boldsymbol{y}$ satisfying the inverse model (6). Let $\boldsymbol{v}_t$ be the approximated score using a proximal network $\boldsymbol{\phi}_{\boldsymbol{\theta}^*}(\boldsymbol{x}_t, \lambda(t)) = \mathrm{Prox}_g^{\lambda(t)}(\boldsymbol{x}_t)$, i.e.,

$$\boldsymbol{v}_t = \frac{\mu(t)\boldsymbol{\phi}_{\boldsymbol{\theta}^*}(\frac{\boldsymbol{x}_t}{\mu(t)} - \lambda(t)\beta\nabla_{\boldsymbol{x}}f(\frac{\boldsymbol{x}_t}{\mu(t)}), \lambda(t)) - \boldsymbol{x}_t}{\mu^2(t)\lambda(t)}. \tag{70}$$

Then, for any $t \in (0, T]$, we have

$$\|\nabla \log \pi_t(\boldsymbol{x}_t) - \boldsymbol{v}_t\| \leq \frac{1}{\mu(t)} \left( \sqrt{\frac{2d}{\beta m \lambda^2(t) + \lambda(t)}} + \frac{4}{\beta L} (\|\nabla f(\boldsymbol{x})\| + 1) \right). \tag{71}$$

Further, there exists $M = \mathcal{O}(\beta^{-1/2})$ such that

$$\|\nabla \log \pi_t(\boldsymbol{x}_t) - \boldsymbol{v}_t\| \leq \frac{M(1 + \|\boldsymbol{x}_t\|)}{\sigma^2(t)}. \tag{72}$$

*Proof.* Decompose the total score estimation error as

$$\|\nabla \log \pi_t(\boldsymbol{x}_t) - \boldsymbol{v}_t\| \leq \underbrace{\|\nabla \log \pi_t(\boldsymbol{x}_t) - \nabla \log \pi_t^\lambda(\boldsymbol{x}_t)\|}_{I_1} + \underbrace{\|\nabla \log \pi_t^\lambda(\boldsymbol{x}_t) - \boldsymbol{v}_t\|}_{I_2}. \tag{73}$$

From Proposition D.4, we have

$$I_1 \leq \frac{\mu(t)}{\sigma(t)} \sqrt{\frac{2d}{\beta m \sigma^2(t) + \mu^2(t)}}. \tag{74}$$

For the error $I_2$, note that $\nabla \log \pi_t^\lambda(\boldsymbol{x}_t) = (\mu(t)\mathrm{Prox}_U^{\lambda(t)}(\boldsymbol{x}_t/\mu(t)) - \boldsymbol{x}_t)/\sigma^2(t)$. Using Proposition E.2 with the proximal splitting approximation gives

$$I_2 \leq \frac{\mu(t)}{\sigma^2(t)} \cdot \frac{4}{\beta L} (\|\nabla f_{\boldsymbol{y}}(\boldsymbol{x}_t/\mu(t))\| + 1). \tag{75}$$

Substituting (74) and (75) into (73), we obtain the score estimation error

$$\|\nabla_{\boldsymbol{x}_t} \log \pi_t(\boldsymbol{x}_t) - \boldsymbol{v}_t\| \leq \frac{1}{\mu(t)} \left( \sqrt{\frac{2d}{\beta m \lambda^2(t) + \lambda(t)}} + \frac{4}{\beta L \lambda(t)} (\|\nabla f_{\boldsymbol{y}}(\boldsymbol{x}_t)\| + 1) \right).$$

Further, we reformulate the above score estimation error as

$$\|\nabla_{\boldsymbol{x}_t} \log \pi_t(\boldsymbol{x}_t) - \boldsymbol{v}_t\| \leq \frac{1}{\sqrt{\beta}} \frac{1}{\mu(t)\lambda(t)} \left( \sqrt{\frac{2d}{m + \frac{1}{\beta\lambda(t)}}} + \frac{4}{\sqrt{\beta}L} (\|\nabla f_{\boldsymbol{y}}(\boldsymbol{x}_t)\| + 1) \right).$$

From the $L$-smoothness of $f_{\boldsymbol{y}}$, we have

$$\|\nabla f_{\boldsymbol{y}}(\boldsymbol{x}_t)\| \leq L\|\boldsymbol{x}_t\| + \|\nabla f_{\boldsymbol{y}}(\boldsymbol{0})\|.$$

Suppose that $\beta \geq 2$, then we have

$$\sqrt{\frac{2d}{m + \frac{1}{\beta\lambda(t)}}} + \frac{4}{\sqrt{\beta}L} (\|\nabla f_{\boldsymbol{y}}(\boldsymbol{x}_t)\| + 1) \leq \sqrt{\frac{2d}{m}} + 2\sqrt{2} \left( \|\boldsymbol{x}_t\| + \frac{\|\nabla f_{\boldsymbol{y}}(\boldsymbol{0})\|}{L} \right).$$

Denote $C_f = \max\left\{ 2\sqrt{2}, \sqrt{\frac{2d}{m}} + \frac{2\sqrt{2}\|\nabla f_{\boldsymbol{y}}(\boldsymbol{0})\|}{L} \right\}$ and $M = C_f/\sqrt{\beta}$. Then we have

$$\|\nabla_{\boldsymbol{x}_t} \log \pi_t(\boldsymbol{x}_t) - \boldsymbol{v}_t\| \leq \frac{C_f}{\sqrt{\beta}} \frac{\mu(t)}{\mu^2(t)\lambda(t)} (1 + \|\boldsymbol{x}_t\|) \leq \frac{M(1 + \|\boldsymbol{x}_t\|)}{\sigma^2(t)},$$

where $M = \mathcal{O}(\beta^{-1/2})$.

$\square$

Theorem E.3 provides a rigorous non-asymptotic error bound for the Moreau score approximation. It demonstrates that the approximation $\boldsymbol{v}_t$—which combines a gradient step on $f$ with a learned proximal operator for $g$—can approximate the true score with an error that decays as $\mathcal{O}(\beta^{-1/2})$. This justifies the use of Moreau score for proximal diffusion sampling in Bayesian inverse problems.

## F. Discretization Schedule

To implement the reverse-time SDE (RP) in practice, we require a numerical discretization scheme. In this section, we provide details of the exponential interpolation method used for proximal diffusion sampling.

### F.1. Euler-Maruyama Discretization

Consider the following reverse process

$$d\boldsymbol{x}_t^{\leftarrow} = \left\{ a(t)\boldsymbol{x}_t^{\leftarrow} - b^2(t)\nabla \log \pi_t(\boldsymbol{x}_t^{\leftarrow}) \right\} dt + b(t)d\bar{\boldsymbol{B}}_t, \tag{76}$$

where $d\bar{\boldsymbol{B}}_t$ is a reverse-time Wiener process and $\nabla \log \pi_t(\boldsymbol{x}_t^{\leftarrow})$ is the score function. To generate samples from the reverse dynamics, we use numerical integration for discretization.

Consider a sequence of stepsizes $\{\gamma_k\}_{k=0,1,\dots,K}$, and denote $t_0 = 0, t_{k+1} = \sum_{j=0}^{k} \gamma_j$. Then the interpolation process of Euler-Maruyama discretization on the interval $t \in [t_k, t_{k+1}]$ is defined by

$$d\bar{\boldsymbol{x}}_t^{EM} = \left\{ a(t_k)\bar{\boldsymbol{x}}_{t_k}^{EM} - b^2(t_k)\nabla \log \pi_{t_k}(\bar{\boldsymbol{x}}_{t_k}^{EM}) \right\} dt + b(t_k)d\bar{\boldsymbol{B}}_t.$$

The corresponding iteration form is given by

$$\bar{\boldsymbol{x}}_{t_{k+1}}^{EM} = \bar{\boldsymbol{x}}_{t_k}^{EM} + \gamma_k \left\{ a(t_k)\bar{\boldsymbol{x}}_{t_k}^{EM} - b^2(t_k)\nabla \log \pi_{t_k}(\bar{\boldsymbol{x}}_{t_k}^{EM}) \right\} + b(t_k)\sqrt{\gamma_k}\boldsymbol{z}, \quad \boldsymbol{z} \sim \mathcal{N}(\boldsymbol{0}, \boldsymbol{I}).$$

After using the Moreau score in (49) to replace the traditional score in (76), the reverse process becomes

$$\begin{aligned} d\boldsymbol{x}_t^{\leftarrow} &= \left\{ a(t)\boldsymbol{x}_t^{\leftarrow} - b^2(t)\nabla \log \pi_t(\boldsymbol{x}_t^{\leftarrow}) \right\} dt + b(t)d\bar{\boldsymbol{B}}_t \\ &= \left\{ \left[ a(t) + \frac{b^2(t)}{\sigma^2(t)} \right] \boldsymbol{x}_t^{\leftarrow} - \frac{b^2(t)\mu(t)}{\sigma^2(t)} \text{Prox}_U^{\sigma^2(t)/\mu^2(t)} \left( \frac{\boldsymbol{x}_t^{\leftarrow}}{\mu(t)} \right) \right\} dt + b(t)d\bar{\boldsymbol{B}}_t. \end{aligned} \tag{77}$$

This leads to the following interpolation process

$$d\bar{\boldsymbol{y}}_t^{EM} = \left\{ - \left[ a(T - t_k) + \frac{b^2(T - t_k)}{\sigma^2(T - t_k)} \right] \bar{\boldsymbol{y}}_{t_k}^{EM} + \frac{b^2(T - t_k)\mu(T - t_k)}{\sigma^2(T - t_k)} \text{Prox}_U^{\lambda(T - t_k)} \left( \frac{\bar{\boldsymbol{y}}_{t_k}^{EM}}{\mu(T - t_k)} \right) \right\} dt + b(T - t_k)d\boldsymbol{B}_t.$$

And the corresponding iteration form is given by:

$$\begin{aligned} \bar{\boldsymbol{y}}_{t_{k+1}}^{EM} &= \bar{\boldsymbol{y}}_{t_k}^{EM} - \gamma_k \left\{ a(t_k)\bar{\boldsymbol{y}}_{t_k}^{EM} - \frac{b^2(t_k)\mu(t_k)}{\sigma^2(t_k)} \left[ \text{Prox}_U^{\sigma^2(t_k)/\mu^2(t_k)} \left( \frac{\bar{\boldsymbol{y}}_{t_k}^{EM}}{\mu(t_k)} \right) - \frac{\bar{\boldsymbol{y}}_{t_k}^{EM}}{\mu(t_k)} \right] \right\} + \sqrt{\gamma_k}b(t_k)\boldsymbol{z}, \\ &= \left( 1 - \delta_k a(t_k) - \gamma_k \frac{b^2(t_k)}{\sigma^2(t_k)} \right) \bar{\boldsymbol{y}}_{t_k}^{EM} + \gamma_k \frac{b^2(t_k)\mu(t_k)}{\sigma^2(t_k)} \text{Prox}_U^{\sigma^2(t_k)/\mu^2(t_k)} \left( \frac{\bar{\boldsymbol{y}}_{t_k}^{EM}}{\mu(t_k)} \right) + \sqrt{\gamma_k}b(t_k)\boldsymbol{z}. \end{aligned}$$

Similarly, for the probability flow ODE, the iteration can be formulated as

$$\bar{\boldsymbol{y}}_{t_{k+1}}^{EM} = \left( 1 - \delta_k a(t_k) - \gamma_k \frac{b^2(t_k)}{2\sigma^2(t_k)} \right) \bar{\boldsymbol{y}}_{t_k}^{EM} + \gamma_k \frac{b^2(t_k)\mu(t_k)}{2\sigma^2(t_k)} \text{Prox}_U^{\sigma^2(t_k)/\mu^2(t_k)} \left( \frac{\bar{\boldsymbol{y}}_{t_k}^{EM}}{\mu(t_k)} \right).$$

### F.2. Exponential Interpolation

As Euler-Maruyama discretization approximates the drift and diffusion coefficients as constant in the SDEs, it suffers from a large discretization error and has poor numerical stability. To obtain a better convergence guarantee, here we consider the exponential interpolation (Zhang & Chen, 2022).

A key observation is that the only nonlinear term with respect to $\boldsymbol{x}_t^{\leftarrow}$ in the reverse process (77) is the proximal operator. Fixing it at time $t = t_k$ will transform the reverse process into a linear SDE with time-dependent coefficients, which admits explicit solutions.

**Proposition F.1** (Exponential interpolation). Consider the reverse SDE in (77), and define a sequence of stepsizes $\{\gamma_k\}_{k=0,1,...,K}$ and $t_0 = 0, t_{k+1} = \sum_{j=0}^{k} \gamma_j$. For $\mu(t)$, $\sigma^2(t)$, and $\lambda(t) = \sigma^2(t)/\mu^2(t)$ defined as in Section 3, the interpolation process on the interval $t \in [t_k, t_{k+1}]$ is given by

$$d\bar{\boldsymbol{x}}_t^{EI} = \left\{ -\left[ a(T-t) + \frac{b^2(T-t)}{\sigma^2(T-t)} \right] \bar{\boldsymbol{x}}_t^{EI} + \frac{b^2(T-t)\mu(T-t)}{\sigma^2(T-t)} \text{Prox}_U^{\lambda(T-t_k)} \left( \frac{\bar{\boldsymbol{x}}_{t_k}^{EI}}{\mu(T-t_k)} \right) \right\} dt + b(T-t)d\boldsymbol{B}_t. \quad (78)$$

Moreover, the corresponding iteration form is given by

$$\bar{\boldsymbol{x}}_{t_{k+1}}^{EI} = \alpha_{1,k} \bar{\boldsymbol{x}}_{t_k}^{EI} + \alpha_{2,k} \text{Prox}_U^{\lambda(T-t_k)} \left( \frac{\bar{\boldsymbol{x}}_{t_k}^{EI}}{\mu(T-t_k)} \right) + \alpha_{3,k} \boldsymbol{\xi}_k, \quad (79)$$

where $\boldsymbol{\xi}_k \sim \mathcal{N}(\boldsymbol{0}, I)$ and

$$\alpha_{1,k} = \frac{\lambda(T-t_{k+1})\mu(T-t_{k+1})}{\lambda(T-t_k)\mu(T-t_k)}, \quad \alpha_{2,k} = \mu(T-t_{k+1})\left(1 - \frac{\lambda(T-t_{k+1})}{\lambda(T-t_k)}\right),$$

$$\alpha_{3,k} = \mu(T-t_{k+1})\sqrt{\lambda(T-t_{k+1})}\sqrt{1 - \frac{\lambda(T-t_{k+1})}{\lambda(T-t_k)}}. \quad (80)$$

*Proof.* For the interpolation process (78) on the interval $t \in [t_k, t_{k+1}]$, denote $\boldsymbol{P}_k = \text{Prox}_U^{\lambda(T-t_k)} \left( \frac{\bar{\boldsymbol{x}}_{t_k}^{EI}}{\mu(T-t_k)} \right)$ and

$$\alpha(t) = -\left[ a(T-t) + \frac{b^2(T-t)}{\sigma^2(T-t)} \right], \quad \beta(t) = \frac{b^2(T-t)\mu(T-t)}{\sigma^2(T-t)} \boldsymbol{P}_k, \quad \gamma(t) = b(T-t).$$

Then the interpolation process can be reformulated as

$$d\bar{\boldsymbol{x}}_t^{EI} = \left\{ \alpha(t)\bar{\boldsymbol{x}}_t^{EI} + \beta(t) \right\} dt + \gamma(t)d\boldsymbol{B}_t, t \in [t_k, t_{k+1}]. \quad (81)$$

This is a linear SDE and the exact solution at $\tau_{k+1}$ can be expressed explicitly. Denote $\Phi(t,s) = \exp\left\{ \int_s^t \alpha(u)du \right\}$. Then the solution to (81) can be written as

$$\bar{\boldsymbol{x}}_{t_{k+1}}^{EI} = \underbrace{\Phi(t_{k+1}, t_k)\bar{\boldsymbol{x}}_{t_k}^{EI}}_{H} + \underbrace{\int_{t_k}^{t_{k+1}} \Phi(t_{k+1}, s)\beta(s)ds}_{I} + \underbrace{\int_{t_k}^{t_{k+1}} \Phi(t_{k+1}, s)\gamma(s)d\boldsymbol{B}_s}_{J}. \quad (82)$$

To show the proposition, it suffices to calculate $H$, $I$, and $J$ in (82). In the following, we calculate them one by one. Firstly, we calculate $H$. For simplicity, we denote the reverse time $\tau_k = T - t_k$, and then

$$\alpha(t) = -\left[ a(\tau) + \frac{b^2(\tau)}{\sigma^2(\tau)} \right], \quad \beta(t) = \frac{b^2(\tau)\mu(\tau)}{\sigma^2(\tau)} \boldsymbol{P}_k, \quad \gamma(t) = b(\tau).$$

Note that $dt = -d\tau$, and as $u$ increases from $t_k$ to $t_{k+1}$, $\tau$ decreases from $\tau_k$ to $\tau_{k+1}$. Then we have

$$\Phi(t_{k+1}, t_k) = \exp\left\{ \int_{t_k}^{t_{k+1}} \alpha(u)du \right\} = \exp\left\{ \int_{\tau_{k+1}}^{\tau_k} \alpha(T-\tau)d\tau \right\}. \quad (83)$$

As $\alpha(T-\tau) = -\left[ a(\tau) + \frac{b^2(\tau)}{\sigma^2(\tau)} \right]$, by integration, the exponential term in (83) can be written as

$$\int_{t_k}^{t_{k+1}} \alpha(u)du = \int_{\tau_{k+1}}^{\tau_k} -\left[ a(\tau) + \frac{b^2(\tau)}{\sigma^2(\tau)} \right] d\tau = -\int_{\tau_{k+1}}^{\tau_k} a(\tau)d\tau - \int_{\tau_{k+1}}^{\tau_k} \frac{b^2(\tau)}{\sigma^2(\tau)}d\tau. \quad (84)$$

Recall that $\mu(\tau) = \exp\left\{ \int_0^\tau a(s)ds \right\}$. Then we have $\frac{d}{d\tau}\log\mu(\tau) = a(\tau)$, and then

$$\int_{\tau_{k+1}}^{\tau_k} a(\tau)d\tau = \log\mu(\tau_k) - \log\mu(\tau_{k+1}) = \log\frac{\mu(\tau_k)}{\mu(\tau_{k+1})}. \quad (85)$$

Further, note that $\sigma^2(\tau) = \mu^2(\tau) \int_0^\tau (b(s)/\mu(s))^2 ds$. Denote $\lambda(\tau) = \int_0^\tau (b(s)/\mu(s))^2 ds = \sigma^2(\tau)/\mu^2(\tau)$. Then we have

$$\frac{b^2(\tau)}{\sigma^2(\tau)} = \frac{b^2(\tau)}{\mu^2(\tau)\lambda(\tau)} = \frac{1}{\lambda(\tau)} \frac{b^2(\tau)}{\mu^2(\tau)}.$$

Also, we have $\lambda'(\tau) = (b(\tau)/\mu(\tau))^2$, and then

$$\frac{b^2(\tau)}{\sigma^2(\tau)} = \frac{1}{\lambda(\tau)} \frac{b^2(\tau)}{\mu^2(\tau)} = \frac{\lambda'(\tau)}{\lambda(\tau)} = \frac{d}{d\tau} \log \lambda(\tau).$$

So we have

$$\int_{\tau_{k+1}}^{\tau_k} \frac{b^2(\tau)}{\sigma^2(\tau)} d\tau = \log \lambda(\tau_k) - \log \lambda(\tau_{k+1}) = \log \frac{\lambda(\tau_k)}{\lambda(\tau_{k+1})}. \tag{86}$$

Substituting (84), (85), and (86) into (83) yields

$$\Phi(t_{k+1}, t_k) = \exp\left\{ \int_{t_k}^{t_{k+1}} \alpha(u)du \right\} = \frac{\lambda(\tau_{k+1})}{\lambda(\tau_k)} \frac{\mu(\tau_{k+1})}{\mu(\tau_k)} = \frac{\sigma^2(\tau_{k+1})}{\sigma^2(\tau_k)} \frac{\mu(\tau_k)}{\mu(\tau_{k+1})}. \tag{87}$$

Next, we calculate the second term $I = \int_{t_k}^{t_{k+1}} \Phi(t_{k+1}, s)\beta(s)ds$ in (82). Similarly, for $s \in [t_k, t_{k+1}]$, we denote $\eta = T - s = \rho$. Then we have

$$\Phi(t_{k+1}, s) = \exp\left\{ \int_s^{t_{k+1}} \alpha(u)du \right\} = \exp\left\{ \int_{\tau_{k+1}}^{\rho} \alpha(T - \eta)d\eta \right\}.$$

Note that

$$\int_{\tau_{k+1}}^{\rho} \alpha(T - \eta)d\eta = \int_{\tau_{k+1}}^{\rho} -\left[ a(\eta) + \frac{b^2(\eta)}{\sigma^2(\eta)} \right] d\eta = -\int_{\tau_{k+1}}^{\rho} a(\eta)d\eta - \int_{\tau_{k+1}}^{\rho} \frac{b^2(\eta)}{\sigma^2(\eta)} d\eta.$$

Similarly, we have

$$\Phi(t_{k+1}, s) = \frac{\lambda(\tau_{k+1})}{\lambda(\rho)} \frac{\mu(\tau_{k+1})}{\mu(\rho)} = \frac{\sigma^2(\tau_{k+1})}{\sigma^2(\rho)} \frac{\mu(\rho)}{\mu(\tau_{k+1})}.$$

Substituting the above equation into $I$, we have

$$I = \int_{t_k}^{t_{k+1}} \Phi(t_{k+1}, s)\beta(s)ds = \int_{\tau_{k+1}}^{\tau_k} \frac{\sigma^2(\tau_{k+1})}{\sigma^2(\rho)} \frac{\mu(\rho)}{\mu(\tau_{k+1})} \frac{b^2(\rho)\mu(\rho)}{\sigma^2(\rho)} P_k d\rho = \frac{\sigma^2(\tau_{k+1})}{\mu(\tau_{k+1})} P_k \int_{\tau_{k+1}}^{\tau_k} \frac{b^2(\rho)\mu^2(\rho)}{\sigma^4(\rho)} d\rho. \tag{88}$$

Note that

$$\frac{\mu^2(\rho)b^2(\rho)}{\sigma^4(\rho)} = \frac{b^2(\rho)}{\mu^2(\rho)\lambda^2(\rho)} = \frac{\lambda'(\rho)}{\lambda^2(\rho)} = -\frac{d}{d\rho}\left( \frac{1}{\lambda(\rho)} \right),$$

then by integration, we get

$$\int_{\tau_{k+1}}^{\tau_k} \frac{\mu^2(\rho)b^2(\rho)}{\sigma^4(\rho)} d\rho = \int_{\tau_{k+1}}^{\tau_k} \frac{\lambda'(\rho)}{\lambda^2(\rho)} d\rho = \left[ -\frac{1}{\lambda(\rho)} \right]_{\tau_{k+1}}^{\tau_k} = \frac{1}{\lambda(\tau_{k+1})} - \frac{1}{\lambda(\tau_k)}. \tag{89}$$

Substituting the above into (88), we get

$$I = \frac{\sigma^2(\tau_{k+1})}{\mu(\tau_{k+1})} P_k \int_{\tau_{k+1}}^{\tau_k} \frac{b^2(\rho)\mu^2(\rho)}{\sigma^4(\rho)} d\rho = \frac{\sigma^2(\tau_{k+1})}{\mu(\tau_{k+1})} P_k \left( \frac{1}{\lambda(\tau_{k+1})} - \frac{1}{\lambda(\tau_k)} \right). \tag{90}$$

Finally, we calculate the stochastic Itô integral of the third term $J$ in (82). The variance of $J$ can be expressed as

$$\mathrm{Var}(J) = \int_{t_k}^{t_{k+1}} \left[ \Phi(t_{k+1}, s)\gamma(s) \right]^2 ds.$$

Similarly, we have

$$[\Phi(t_{k+1}, T - \rho)b(\rho)]^2 = \frac{\mu^2(\rho)\sigma^4(\tau_{k+1})}{\mu^2(\tau_{k+1})\sigma^4(\rho)}b^2(\rho) = \frac{\sigma^4(\tau_{k+1})}{\mu^2(\tau_{k+1})} \cdot \frac{\mu^2(\rho)b^2(\rho)}{\sigma^4(\rho)},$$

and further

$$\int_{t_k}^{t_{k+1}} [\Phi(t_{k+1}, s)\gamma(s)]^2 \, ds = \int_{\tau_{k+1}}^{\tau_k} [\Phi(t_{k+1}, T - \rho)b(\rho)]^2 \, d\rho = \frac{\sigma^4(\tau_{k+1})}{\mu^2(\tau_{k+1})} \int_{\tau_{k+1}}^{\tau_k} \frac{\mu^2(\rho)b^2(\rho)}{\sigma^4(\rho)} d\rho$$

$$= \frac{\sigma^4(\tau_{k+1})}{\mu^2(\tau_{k+1})} \left( \frac{1}{\lambda(\tau_{k+1})} - \frac{1}{\lambda(\tau_k)} \right).$$

For $\boldsymbol{\xi}_k \sim \mathcal{N}(\mathbf{0}, I)$, we have

$$J = \frac{\sigma^2(\tau_{k+1})}{\mu(\tau_{k+1})} \sqrt{\frac{1}{\lambda(\tau_{k+1})} - \frac{1}{\lambda(\tau_k)}} \boldsymbol{\xi}_k. \tag{91}$$

Combining (87), (90), and (91), we have

$$\bar{\boldsymbol{x}}_{t_{k+1}}^{EI} = \Phi(t_{k+1}, t_k)\bar{\boldsymbol{x}}_{t_k}^{EI} + \int_{t_k}^{t_{k+1}} \Phi(t_{k+1}, s)\beta(s)ds + \int_{t_k}^{t_{k+1}} \Phi(t_{k+1}, s)\gamma(s)d\boldsymbol{B}_s$$

$$= \frac{\sigma^2(\tau_{k+1})}{\sigma^2(\tau_k)} \frac{\mu(\tau_k)}{\mu(\tau_{k+1})}\bar{\boldsymbol{x}}_{t_k}^{EI} + \frac{\sigma^2(\tau_{k+1})}{\mu(\tau_{k+1})}P_k \left( \frac{1}{\lambda(\tau_{k+1})} - \frac{1}{\lambda(\tau_k)} \right) + \frac{\sigma^2(\tau_{k+1})}{\mu(\tau_{k+1})} \sqrt{\frac{1}{\lambda(\tau_{k+1})} - \frac{1}{\lambda(\tau_k)}} \boldsymbol{\xi}_k. \tag{92}$$

Denote the coefficients

$$\alpha_{1,k} = \frac{\lambda(T - t_{k+1})\mu(T - t_{k+1})}{\lambda(T - t_k)\mu(T - t_k)}, \quad \alpha_{2,k} = \mu(T - t_{k+1}) \left( 1 - \frac{\lambda(T - t_{k+1})}{\lambda(T - t_k)} \right),$$

$$\alpha_{3,k} = \mu(T - t_{k+1})\sqrt{\lambda(T - t_{k+1})} \sqrt{1 - \frac{\lambda(T - t_{k+1})}{\lambda(T - t_k)}}.$$

Then the corresponding iteration form is given by

$$\bar{\boldsymbol{x}}_{t_{k+1}}^{EI} = \alpha_{1,k}\bar{\boldsymbol{x}}_{t_k}^{EI} + \alpha_{2,k}\text{Prox}_U^{\lambda(T - t_k)} \left( \frac{\bar{\boldsymbol{x}}_{t_k}^{EI}}{\mu(T - t_k)} \right) + \alpha_{3,k}\boldsymbol{\xi}_k, \quad \boldsymbol{\xi}_k \sim \mathcal{N}(\mathbf{0}, \boldsymbol{I}).$$

$\square$

The exponential interpolation scheme provides an exact discretization of the reverse SDE with proximal operator. The resulting iteration in (79) has a clear interpretation: it is a convex combination of the previous state $\bar{\boldsymbol{x}}_{t_k}^{EI}$ and the proximal update plus appropriate noise. The coefficients $\alpha_{1,k}, \alpha_{2,k}, \alpha_{3,k}$ depend only on the scheduling $\mu(t)$ and $\lambda(t)$, making the scheme easy to implement.

## G. Convergence Results

In this section, we establish the main convergence result of the work, providing a comprehensive error analysis for the proposed proximal diffusion sampling algorithm. Firstly, in subsection G.1, we show several quantitative controls on the forward density $\pi_t$. Next, we provide uniform second-moment bounds for the reverse diffusion processes with approximated Moreau score in subsection G.2. Then, we analyze the tangent process associated with the reverse SDE in subsection G.3, which quantifies how small perturbations propagate along the reverse-time dynamics. Finally, in subsection G.4, we present the proof of Theorem 4.3, which establishes the non-asymptotic convergence of the proximal diffusion sampler.

## G.1. Properties of the Forward Process

In this subsection, we establish several technical bounds for the forward diffusion process that will be used to control the growth of moments and to analyze the errors in the sampling phase.

Recall that the forward SDE is given by

$$d\boldsymbol{x}_t^{\rightarrow} = a(t)\boldsymbol{x}_t^{\rightarrow}dt + b(t)d\boldsymbol{B}_t, \qquad t \in [0, T],$$

with deterministic coefficients

$$\mu(t) = \exp\Big\{\int_0^t a(s)ds\Big\}, \qquad \sigma^2(t) = \int_0^t b^2(s)\exp\Big\{2\int_s^t a(r)dr\Big\}ds.$$

We work under Assumptions 3.3 and 4.1. Denote $\mathrm{diam}(\mathcal{X})$ the diameter of the manifold defined by $\mathrm{diam}(\mathcal{X}) = \sup\{\|x-y\| : x, y \in \mathcal{X}\}$. We emphasize that the compact assumption on $\mathcal{X}$ encompasses not only all distributions which admit a continuous density on a lower dimensional manifold but also all empirical densities.

Under Assumption 4.1, by directly restricting the log-gradient of $\mu(t)$ and $\lambda(t)$, we can obtain simple expressions for the ratio in the discrete iteration form, i.e.,

$$\exp\{\gamma_k m_\lambda\} \le \frac{\lambda(T - t_k)}{\lambda(T - t_{k+1})} \le \exp\{\gamma_k M_\lambda\}, \quad \exp\{\gamma_k m_\mu\} \le \frac{\mu(T - t_{k+1})}{\mu(T - t_k)} \le \exp\{\gamma_k M_\mu\}. \tag{93}$$

The gradient can also be expressed as

$$|\partial_t \mu(t)| \le |\partial_t \log \mu(t)| \le M_\mu, \quad \partial_t \lambda(t) \le \partial_t \log \lambda(t) \le \bar{\lambda} M_\lambda. \tag{94}$$

Further, we can conclude that both $\mu(t)$ and $\lambda(t)$ can be bounded, i.e.,

$$\mu(t) \in [\exp\{-M_\mu T\}, 1], \quad \lambda(t) \in [\bar{\lambda}\exp\{-M_\lambda T\}, \bar{\lambda}], \quad \forall\, t \in [0, T]. \tag{95}$$

In order to show the stability and growth of the processes at hand, we need to control quantities related to the gradient and Hessian of the log-density. For the target posterior $\pi(\boldsymbol{x}_0) \propto \exp\{-U(\boldsymbol{x}_0)\}$, denote $(\pi_t)_{t \in [0,T]}$ as the density w.r.t. the Lebesgue measure of the distribution of $\boldsymbol{x}_t^{\rightarrow}$. Similarly, we suppose $\pi^N$ to be an empirical version of $\pi$, i.e.,

$$\pi^N = \frac{1}{N}\sum_{k=1}^N \boldsymbol{x}^{(k)}, \quad \boldsymbol{x}^{(k)} \sim \pi.$$

And we denote $(\pi_t^N)_{t \in [0,T]}$ as the density w.r.t. the Lebesgue measure of the empirical version of distribution. Next we will give the regularity of the forward density. We adopt the backbone of the proof in (De Bortoli, 2022) and extend it to more general settings.

**Lemma G.1** (Regularity of the forward density). Under Assumption 3.3, for every $t \in [0, T]$ and $\boldsymbol{x} \in \mathbb{R}^d$ the following inequalities hold.

1. (Control on the gradient)

$$\langle \nabla \log \pi_t(\boldsymbol{x}_t), \boldsymbol{x}_t \rangle \le -\frac{\|\boldsymbol{x}_t\|^2}{\sigma^2(t)} + \frac{\mu(t)\mathrm{diam}(\mathcal{X})\|\boldsymbol{x}_t\|}{\sigma^2(t)},$$

$$\|\nabla \log \pi_t(\boldsymbol{x}_t)\|^2 \le \frac{2\|\boldsymbol{x}_t\|^2}{\sigma^4(t)} + \frac{2\mu^2(t)\mathrm{diam}(\mathcal{X})^2}{\sigma^4(t)}. \tag{96}$$

2. (Control on the Hessian) For any $\boldsymbol{y} \in \mathbb{R}^d$,

$$\langle \boldsymbol{y}, \nabla^2 \log \pi_t(\boldsymbol{x}_t)\boldsymbol{y} \rangle \le -\frac{2\sigma^2(t) - \mu^2(t)\mathrm{diam}(\mathcal{X})^2}{2\sigma^4(t)}\|\boldsymbol{y}\|^2,$$

$$\|\nabla^2 \log \pi_t(\boldsymbol{x}_t)\| \le \frac{1 + \mathrm{diam}(\mathcal{X})^2}{\sigma^4(t)}. \tag{97}$$

3. (Control on the time derivative)

$$\|\partial_t \nabla \log \pi_t(\boldsymbol{x})\| \le M_{f,t} \frac{(\text{diam}(\mathcal{X})^2 + \bar{\lambda})(\text{diam}(\mathcal{X}) + \|x\|)}{\sigma^6(t)}, \tag{98}$$

where $M_{f,t} = 2(M_\lambda + 4M_\mu)\bar{\lambda}$.

*Proof.* We prove each part in sequence.

**Part (i): Control on the gradient.**

We first show the dissipativity condition on the gradient, i.e., the inequalities in (96). For any $t \in [0,T]$ and $\boldsymbol{x}_t \in \mathbb{R}^d$ we have

$$\pi_t^N(\boldsymbol{x}_t) = \frac{1}{N} \sum_{k=1}^N \frac{\exp\left[-\|\boldsymbol{x}_t - \mu(t)\boldsymbol{x}^{(k)}\|^2/(2\sigma^2(t))\right]}{(2\pi\sigma^2(t))^{d/2}}.$$

And the gradient of $\log \pi_t^N$ is given by

$$\nabla \log \pi_t^N(\boldsymbol{x}_t) = \frac{-\frac{1}{N}\sum_{k=1}^N (\boldsymbol{x}_t - \mu(t)\boldsymbol{x}^{(k)})\exp\left[-\|\boldsymbol{x}_t - \mu(t)\boldsymbol{x}^{(k)}\|^2/(2\sigma^2(t))\right]/(2\pi\sigma^2(t))^{d/2}}{\sigma^2(t)\,\pi_t^N(\boldsymbol{x}_t)}. \tag{99}$$

Define the weights

$$w_k := \frac{\exp\left[-\|\boldsymbol{x}_t - \mu(t)X^k\|^2/(2\sigma^2(t))\right]}{\sum_{j=1}^N \exp\left[-\|\boldsymbol{x}_t - \mu(t)\boldsymbol{x}^{(j)}\|^2/(2\sigma^2(t))\right]}, \qquad k = 1, 2, \ldots, N.$$

Then we have $w_k \ge 0$ and $\sum_{k=1}^N w_k = 1$, and

$$\nabla \log \pi_t^N(\boldsymbol{x}_t) = -\frac{1}{\sigma^2(t)}\left(\boldsymbol{x}_t - \mu(t)\sum_{k=1}^N w_k \boldsymbol{x}^{(k)}\right). \tag{100}$$

Taking the inner product of (100) with $\boldsymbol{x}_t$ gives

$$\langle \nabla \log \pi_t^N(\boldsymbol{x}_t), \boldsymbol{x}_t \rangle = -\frac{1}{\sigma^2(t)}\left(\|\boldsymbol{x}_t\|^2 - \mu(t)\left\langle \boldsymbol{x}_t, \sum_{k=1}^N w_k \boldsymbol{x}^{(k)}\right\rangle\right) \le -\frac{\|\boldsymbol{x}_t\|^2}{\sigma^2(t)} + \frac{\mu(t)}{\sigma^2(t)}\left|\left\langle \boldsymbol{x}_t, \sum_{k=1}^N w_k \boldsymbol{x}^{(k)}\right\rangle\right|. \tag{101}$$

By the Cauchy-Schwarz inequality, we have

$$\left|\left\langle \boldsymbol{x}_t, \sum_{k=1}^N w_k \boldsymbol{x}^{(k)}\right\rangle\right| \le \|\boldsymbol{x}_t\|\left\|\sum_{k=1}^N w_k \boldsymbol{x}^{(k)}\right\| \le \|\boldsymbol{x}_t\|\sum_{k=1}^N w_k\|\boldsymbol{x}^{(k)}\|. \tag{102}$$

Since $\mathcal{X}$ is bounded with diameter $\text{diam}(\mathcal{X})$, we have $\|\boldsymbol{x}^{(k)}\| \le \text{diam}(\mathcal{X})$ for every $k$ (after possibly translating $\mathcal{X}$ so that it lies inside a ball of radius $\text{diam}(\mathcal{X})$ centered at the origin). Consequently,

$$\sum_{k=1}^N w_k\|\boldsymbol{x}^{(k)}\| \le \text{diam}(\mathcal{X})\sum_{k=1}^N w_k = \text{diam}(\mathcal{X}). \tag{103}$$

Substituting (102) and (103) into (101), we have

$$\langle \nabla \log \pi_t^N(\boldsymbol{x}_t), \boldsymbol{x}_t \rangle \le -\frac{\|\boldsymbol{x}_t\|^2}{\sigma^2(t)} + \frac{\mu(t)\text{diam}(\mathcal{X})\|\boldsymbol{x}_t\|}{\sigma^2(t)}. \tag{104}$$

Because the right-hand side does not depend on $N$, the same estimate holds for the limiting density $\pi_t$ by letting $N \to +\infty$.

Next, we show the second inequality in (96). From the above equation (100), we obtain

$$\|\nabla \log \pi_t^N(\boldsymbol{x}_t)\| = \frac{1}{\sigma^2(t)}\left\|\boldsymbol{x}_t - \mu(t)\sum_{k=1}^N w_k \boldsymbol{x}^{(k)}\right\|. \tag{105}$$

Using the inequality $\|a - b\|^2 \leq 2\|a\|^2 + 2\|b\|^2$, we have

$$\left\|\boldsymbol{x}_t - \mu(t)\sum_{k=1}^{N} w_k \boldsymbol{x}^{(k)}\right\|^2 \leq 2\|\boldsymbol{x}_t\|^2 + 2\mu^2(t)\left\|\sum_{k=1}^{N} w_k \boldsymbol{x}^{(k)}\right\|^2. \tag{106}$$

Again by the convexity of the norm and the bound $\|\boldsymbol{x}^{(k)}\| \leq \operatorname{diam}(\mathcal{X})$, we get

$$\left\|\sum_{k=1}^{N} w_k \boldsymbol{x}^{(k)}\right\| \leq \sum_{k=1}^{N} w_k \|\boldsymbol{x}^{(k)}\| \leq \operatorname{diam}(\mathcal{X}). \tag{107}$$

Hence,

$$\left\|\boldsymbol{x}_t - \mu(t)\sum_{k=1}^{N} w_k \boldsymbol{x}^{(k)}\right\|^2 \leq 2\|\boldsymbol{x}_t\|^2 + 2\mu^2(t)\operatorname{diam}(\mathcal{X})^2, \tag{108}$$

and consequently

$$\|\nabla \log \pi_t^N(\boldsymbol{x}_t)\|^2 \leq \frac{2\|\boldsymbol{x}_t\|^2}{\sigma^4(t)} + \frac{2\mu^2(t)\operatorname{diam}(\mathcal{X})^2}{\sigma^4(t)}. \tag{109}$$

Again, letting $N \to +\infty$ yields the same bound for $\pi_t$.

**Part (ii): Control on the Hessian.**

We now prove the control on the Hessian $\nabla^2 \log \pi_t$. Denote $\bar{\pi}_t^N = \pi_t^N (2\pi\sigma^2(t))^{d/2}$. Then we have

$$\bar{\pi}_t^N(\boldsymbol{x}) = (1/N)\sum_{k=1}^{N} \exp[-\|\boldsymbol{x} - \mu(t)\boldsymbol{x}^{(k)}\|^2/2\sigma^2(t)]. \tag{110}$$

Hence, taking the gradient on (110), we have

$$\nabla \log \bar{\pi}_t^N(\boldsymbol{x}) = (-1/N)\sum_{k=1}^{N}(\boldsymbol{x} - \mu(t)\boldsymbol{x}^{(k)})\exp[-\|\boldsymbol{x} - \mu(t)\boldsymbol{x}^{(k)}\|^2/2\sigma^2(t)]/(\sigma^2(t)\bar{\pi}_t^N(\boldsymbol{x})).$$

The Hessian can also be calculated as

$$\nabla^2 \log \bar{\pi}_t^N(\boldsymbol{x}) = -\frac{1}{\sigma^2(t)}I + (1/N)\sum_{k=1}^{N}(\boldsymbol{x} - \mu(t)\boldsymbol{x}^{(k)}) \otimes (\boldsymbol{x} - \mu(t)\boldsymbol{x}^{(k)})\exp[-\|\boldsymbol{x} - \mu(t)\boldsymbol{x}^{(k)}\|^2/2\sigma^2(t)]/(\sigma^4(t)\bar{\pi}_t^N(\boldsymbol{x}))$$

$$- (1/N^2)(\sum_{k=1}^{N}(\boldsymbol{x} - \mu(t)\boldsymbol{x}^{(k)})\exp[-\|\boldsymbol{x} - \mu(t)\boldsymbol{x}^{(k)}\|^2/2\sigma^2(t)])$$

$$\otimes (\sum_{k=1}^{N}(\boldsymbol{x} - \mu(t)\boldsymbol{x}^{(k)})\exp[-\|\boldsymbol{x} - \mu(t)\boldsymbol{x}^{(k)}\|^2/2\sigma^2(t)])/(\sigma^2(t)\bar{\pi}_t^N(\boldsymbol{x}))^2. \tag{111}$$

For any $k \in \{0, 1, \ldots, N-1\}$, denote $\boldsymbol{f}_t^k = -(\boldsymbol{x} - \mu(t)\boldsymbol{x}^{(k)})/\sigma^2(t)$ and $e_t^k = \exp[-\|\boldsymbol{f}_t^k\|^2]$. Substituting this into (111), we have

$$\nabla^2 \log \bar{\pi}_t^N(\boldsymbol{x}) = -\frac{1}{\sigma^2(t)}I + \sum_{k=1}^{N} \boldsymbol{f}_t^k \otimes \boldsymbol{f}_t^k e_t^k / \sum_{k=1}^{N} e_t^k - (\sum_{k=1}^{N} \boldsymbol{f}_t^k e_t^k / \sum_{k=1}^{N} e_t^k) \otimes (\sum_{k=1}^{N} \boldsymbol{f}_t^k e_t^k / \sum_{k=1}^{N} e_t^k)$$

$$= -\frac{1}{\sigma^2(t)}I + (1/2)\sum_{j,k=1}^{N}(\boldsymbol{f}_t^k - \boldsymbol{f}_t^j) \otimes (\boldsymbol{f}_t^k - \boldsymbol{f}_t^j)e_t^k e_t^j / \sum_{k,j=1}^{N} e_t^k e_t^j.$$

In addition, using that for any $\ell \in \{1, 2, \ldots, N\}, \boldsymbol{x}^{(\ell)} \in \mathcal{X}$ we have

$$\|\boldsymbol{f}_t^k - \boldsymbol{f}_t^j\| = \mu(t)\|\boldsymbol{x}^{(k)} - \boldsymbol{x}^{(j)}\|/\sigma^2(t) \leq \mu(t)\operatorname{diam}(\mathcal{X})/\sigma^2(t).$$

Therefore, we get that, for any $\boldsymbol{x}$,

$$\langle \boldsymbol{x}, \nabla^2 \log \bar{\pi}_t^N(\boldsymbol{x})\boldsymbol{x}\rangle \leq -(1 - \mu(t)^2 \mathrm{diam}(\mathcal{X})^2/(2\sigma^2(t)))/\sigma^2(t)\|\boldsymbol{x}\|^2.$$

Using the fact that $\mathcal{X}$ is compact and the strong law of large numbers, we have

$$\lim_{N \to +\infty} \nabla^2 \log \bar{\pi}_t^N(\boldsymbol{x}) = -\frac{1}{\sigma^2(t)}I$$
$$+ \int_{\mathbb{R}^d} (\boldsymbol{x} - \mu(t)\boldsymbol{x}_0) \otimes (\boldsymbol{x} - \mu(t)\bar{\boldsymbol{x}}_0) \exp[-\|\boldsymbol{x} - \mu(t)\boldsymbol{x}_0\|^2/(2\sigma^2(t))] \exp[-\|\boldsymbol{x} - \mu(t)\bar{\boldsymbol{x}}_0\|^2/(2\sigma^2(t))] \mathrm{d}\pi(\boldsymbol{x}_0)\mathrm{d}\pi(\bar{\boldsymbol{x}}_0)$$
$$/(\int_{\mathbb{R}^d} \exp[-\|\boldsymbol{x} - \mu(t)\bar{\boldsymbol{x}}_0\|^2/(2\sigma^2(t))]\mathrm{d}\pi(\boldsymbol{x}_0))^2.$$

Hence, we get that $\lim_{N \to +\infty} \nabla^2 \log \pi_t^N(\boldsymbol{x}) = \nabla^2 \log \pi_t$, and

$$\langle \boldsymbol{x}, \nabla^2 \log \pi_t(\boldsymbol{x})\boldsymbol{x}\rangle \leq -(1 - \mu(t)^2 \mathrm{diam}(\mathcal{X})^2/(2\sigma^2(t)))/\sigma^2(t)\|\boldsymbol{x}\|^2.$$

Similarly, we can show

$$\|\nabla^2 \log p_t(\boldsymbol{x}_t)\| \leq (1 + \mathrm{diam}(\mathcal{X})^2)/\sigma^4(t).$$

**Part (iii): Control on the time derivative.**

Finally, in order to control the local error of the time discretization, we also need to control the time derivative of the gradient in (98).

Note that for any $\boldsymbol{x} \in \mathbb{R}^d$, $\pi_t^N(\boldsymbol{x}) = \bar{\pi}_t^N(\boldsymbol{x})/(2\pi\sigma^2(t))^{d/2}$ with

$$\bar{\pi}_t^N(\boldsymbol{x}) = (1/N)\sum_{k=1}^N e_t^k(\boldsymbol{x}), \quad e_t^k(\boldsymbol{x}) = \exp[-\|\boldsymbol{x} - \mu(t)\boldsymbol{x}^{(k)}\|^2/(2\sigma^2(t))].$$

In what follows, we denote $f_t^k = \log e_t^k$ for any $k \in \{1, 2, \ldots, N\}$. For any $\boldsymbol{x} \in \mathbb{R}^d$, we have

$$\partial_t \log \bar{p}_t^N(\boldsymbol{x}) = \sum_{k=1}^N \partial_t f_t^k(\boldsymbol{x})e_t^k(\boldsymbol{x})/\sum_{k=1}^N e_t^k(\boldsymbol{x}).$$

Taking the gradient in the above, we further have

$$\begin{aligned}
\partial_t \nabla \log \bar{p}_t^N(\boldsymbol{x}) &= \sum_{k=1}^N \partial_t \partial_t^k(\boldsymbol{x})e_t^k(\boldsymbol{x})/\sum_{k=1}^N e_t^k(\boldsymbol{x}) + \sum_{k=1}^N \partial_t f_t^k(\boldsymbol{x})\nabla f_t^k(\boldsymbol{x})e_t^k(\boldsymbol{x})/\sum_{k=1}^N e_t^k(\boldsymbol{x}) \\
&\quad - \sum_{k,j=1}^N \partial_t f_t^k(\boldsymbol{x})\nabla f_t^j(\boldsymbol{x})e_t^k(\boldsymbol{x})e_t^j(\boldsymbol{x})/\sum_{k,j=1}^N e_t^k(\boldsymbol{x})e_t^j(\boldsymbol{x}) \\
&= \sum_{k=1}^N \partial_t \nabla f_t^k(\boldsymbol{x})e_t^k(\boldsymbol{x})/\sum_{k=1}^N e_t^k(\boldsymbol{x}) \\
&\quad + \frac{1}{2}\sum_{k,j=1}^N (\partial_t f_t^k(\boldsymbol{x}) - \partial_t f_t^j(\boldsymbol{x}))(\nabla f_t^k(\boldsymbol{x}) - \nabla f_t^j(\boldsymbol{x}))e_t^k(\boldsymbol{x})e_t^j(\boldsymbol{x})/\sum_{k,j=1}^N e_t^k(\boldsymbol{x})e_t^j(\boldsymbol{x}).
\end{aligned} \tag{112}$$

Note that for any $\boldsymbol{x} \in \mathbb{R}^d$,

$$\nabla f_t^k(\boldsymbol{x}) = -(\boldsymbol{x} - \mu(t)\boldsymbol{x}^{(k)})/\sigma^2(t).$$

Hence, using that $\mu(t) \leq 1$, we get

$$\|\nabla f_t^k(\boldsymbol{x}) - \nabla f_t^j(\boldsymbol{x})\| \leq \mu(t)\mathrm{diam}(\mathcal{X})/\sigma^2(t) \leq \mathrm{diam}(\mathcal{X})/\sigma^2(t). \tag{113}$$

In addition, we have that, for any $x \in \mathbb{R}^d$

$$\partial_t f_t^k(x) = \partial_t \sigma^2(t)/(2\sigma^4(t))\|x - \mu(t)x^{(k)}\|^2 + \partial_t \mu(t)/\sigma^2(t)\langle x^{(k)}, x - \mu(t)x^{(k)}\rangle.$$

Combining the above result with Assumption 4.1, and note that $\partial_t \sigma^2(t) = \mu^2(t)\partial_t \lambda(t) + 2\mu(t)\partial_t \mu(t)\lambda(t)$, we get

$$
\begin{aligned}
|\partial_t f_t^k(x) - \partial_t f_t^j(x)| &\leq \partial_t \sigma^2(t)/(2\sigma^4(t)) \left(4\|x\|\mathrm{diam}(\mathcal{X}) + 2\mathrm{diam}(\mathcal{X})^2\right) + \partial_t \mu(t)/\sigma^2(t) \left(2\mathrm{diam}(\mathcal{X})\|x\| + 2\mathrm{diam}(\mathcal{X})^2\right) \\
&\leq \mathrm{diam}(\mathcal{X})(M_\lambda + 2M_\mu)\bar{\lambda}/\sigma^4(t) \left(2\|x\| + \mathrm{diam}(\mathcal{X})\right) + 2\mathrm{diam}(\mathcal{X})M_\mu/\sigma^2(t) \left(\|x\| + 2\mathrm{diam}(\mathcal{X})\right) \\
&\leq (M_{f,t}/\sigma^4(t))\mathrm{diam}(\mathcal{X})(\mathrm{diam}(\mathcal{X}) + \|x\|),
\end{aligned}
$$

(114)

where $M_{f,t} = 2(M_\lambda + 4M_\mu)\bar{\lambda}$. Therefore, combining (114) and the fact that $\mu(t) \leq 1$, we get that, for any $x \in \mathbb{R}^d$,

$$\|\partial_t \nabla f_t^k(x)\| \leq (M_{f,t}/\sigma^4(t))(\mathrm{diam}(\mathcal{X}) + \|x\|). \tag{115}$$

Substituting (113) and (115) into (112), we get that, for any $x \in \mathbb{R}^d$,

$$
\begin{aligned}
\|\partial_t \nabla \log \bar{p}_t^N(x)\| &\leq (M_{f,t}/\sigma^4(t))(\mathrm{diam}(\mathcal{X}) + \|x\|) + (M_{f,t}/\sigma^6(t))\mathrm{diam}(\mathcal{X})^2(\mathrm{diam}(\mathcal{X}) + \|x\|) \\
&\leq (M_{f,t}/\sigma^6(t))(\bar{\lambda} + \mathrm{diam}(\mathcal{X})^2)(\mathrm{diam}(\mathcal{X}) + \|x\|).
\end{aligned}
$$

Using that $\lim_{N\to+\infty} \partial_t \nabla \log \pi_t^N(x_t) = \partial_t \nabla \log \pi_t$, we have

$$\|\partial_t \nabla \log \pi_t(x)\| \leq (M_{f,t}/\sigma^6(t))(\bar{\lambda} + \mathrm{diam}(\mathcal{X})^2)(\mathrm{diam}(\mathcal{X}) + \|x\|).$$

$\square$

This lemma provides crucial quantitative controls on the forward density $\pi_t$, which is essential for establishing stability of the reverse SDE. The time derivative bound controls how quickly the score changes with time, which is necessary for analyzing the local truncation error of time discretization.

### G.2. Properties of the Reverse Process

In order to control the error introduced by the discretization and the score estimation, we firstly obtain a uniform control of the moments of the reverse processes.

Consider the reverse diffusion process:

$$d\boldsymbol{x}_t^{\leftarrow} = \left\{a(t)\boldsymbol{x}_t^{\leftarrow} - b^2(t)\nabla_{\boldsymbol{x}_t^{\leftarrow}} \log \pi_t(\boldsymbol{x}_t^{\leftarrow})\right\} dt + b(t)d\bar{\boldsymbol{B}}_t.$$

Replacing the score function as the Moreau score in (49) and using the proximal network $\boldsymbol{\phi}_{\boldsymbol{\theta}^*}$ to approximate the proximal term $\mathrm{Prox}_U^{\sigma^2(t)/\mu^2(t)}(\boldsymbol{x}_t/\mu(t))$, we obtain the following SDE:

$$d\hat{\boldsymbol{x}}_t^{\leftarrow} = \left\{\left[a(t) + \frac{b^2(t)}{\sigma^2(t)}\right]\hat{\boldsymbol{x}}_t^{\leftarrow} - \frac{b^2(t)\mu(t)}{\sigma^2(t)}\boldsymbol{P}_t\right\} dt + b(t)d\bar{\boldsymbol{B}}_t, \tag{116}$$

where $\lambda(t) = \sigma^2(t)/\mu^2(t)$ and

$$\boldsymbol{P}_t = \boldsymbol{\phi}_{\boldsymbol{\theta}^*}\left(\frac{\boldsymbol{x}_t}{\mu(t)} - \lambda(t)\beta\nabla f(\frac{\boldsymbol{x}_t}{\mu(t)}), \lambda(t)\right).$$

Denote the approximated score

$$\boldsymbol{v}_t = \frac{\mu(t)\boldsymbol{\phi}_{\boldsymbol{\theta}^*}(\frac{\boldsymbol{x}_t}{\mu(t)} - \lambda(t)\beta\nabla_{\boldsymbol{x}} f(\frac{\boldsymbol{x}_t}{\mu(t)}), \lambda(t)) - \boldsymbol{x}_t}{\mu^2(t)\lambda(t)}. \tag{117}$$

Then we have

$$d\hat{\boldsymbol{x}}_t^{\leftarrow} = \left\{a(t)\hat{\boldsymbol{x}}_t^{\leftarrow} - b^2(t)\boldsymbol{v}_t\right\} dt + b(t)d\bar{\boldsymbol{B}}_t. \tag{118}$$

From Proposition F.1, the interpolation process on the interval $t \in [t_k, t_{k+1}]$ is given by

$$d\bar{\boldsymbol{x}}_t^{EI} = \left\{ -\left[ a(T-t) + \frac{b^2(T-t)}{\sigma^2(T-t)} \right] \bar{\boldsymbol{x}}_t^{EI} + \frac{b^2(T-t)\mu(T-t)}{\sigma^2(T-t)} \boldsymbol{P}_k \right\} dt + b(T-t)d\boldsymbol{B}_t,$$

where

$$\boldsymbol{P}_k = \boldsymbol{\phi}_{\boldsymbol{\theta}^*} \left( \frac{\bar{\boldsymbol{x}}_{t_k}^{EI}}{\mu(T-t_k)} - \lambda(T-t_k)\beta\nabla f(\frac{\bar{\boldsymbol{x}}_{t_k}^{EI}}{\mu(T-t_k)}), \lambda(T-t_k) \right).$$

And the corresponding iteration form is given by

$$\bar{\boldsymbol{x}}_{t_{k+1}}^{EI} = \alpha_{1,k}\bar{\boldsymbol{x}}_{t_k}^{EI} + \alpha_{2,k}\boldsymbol{P}_k + \alpha_{3,k}\boldsymbol{\xi}_k, \quad \boldsymbol{\xi}_k \sim \mathcal{N}(\boldsymbol{0}, I).$$

For simplicity, we denote $\bar{\boldsymbol{x}}_{t_k}^{EI}$ as $\boldsymbol{y}_k$, and then we have

$$\boldsymbol{y}_{k+1} = \alpha_{1,k}\boldsymbol{y}_k + \alpha_{2,k}\boldsymbol{P}_k + \alpha_{3,k}\boldsymbol{\xi}_k, \quad \boldsymbol{\xi}_k \sim \mathcal{N}(\boldsymbol{0}, I). \tag{119}$$

Denote the distribution of $\boldsymbol{y}_k$ as $\pi_\infty R_k$, where we choose $\boldsymbol{y}_0 \sim \pi_\infty$ and $R_k$ the transition kernel associated with $\boldsymbol{y}_k | \boldsymbol{y}_0$. The following lemma provides uniform bounds for the second moments of $\hat{\boldsymbol{x}}_t^\leftarrow$, $\boldsymbol{y}_k$ and for the increment of the interpolant, depending on the score estimation error introduced in Theorem E.3.

**Lemma G.2** (Moment bounds for the reverse process). Under Assumption 3.3 and 4.1, let $\hat{\boldsymbol{x}}_t^\leftarrow$ satisfy (118) and let $\boldsymbol{y}_k$ be defined by (119) with $\boldsymbol{y}_0 \sim \pi_\infty$. Then the following hold

1. (Control of the approximated process) For sufficiently small $M \leq 1/6$, and for any $t \in [0, T]$,

$$\mathbb{E}[\|\hat{\boldsymbol{x}}_t^\leftarrow\|^2] \leq 12M_\lambda(M + \mathrm{diam}(\mathcal{X}))^2 + 2\bar{\lambda}d. \tag{120}$$

2. (Control of the sampling process) For any $k = 0, 1, \ldots, K$, we get

$$\mathbb{E}[\|\boldsymbol{y}_k\|^2] \leq R_k := \bar{\lambda}d + B\left(1/A + \delta\right), \tag{121}$$

where

$$A = \frac{1}{2\ln 2}\left(1 + \frac{\gamma_k}{8\ln 2}\right) + (2M + \eta)(1 - \frac{\gamma_k}{4\ln 2})\exp\{M_\mu\delta\}M_\lambda + \exp\{2M_\mu\delta\}M_\lambda^2\gamma_k\left(2M^2 + 1 + (2M + \eta)\bar{\lambda}\right)$$
$$B = \exp\{M_\mu\delta\}M_\lambda\frac{(M + \mathrm{diam}(\mathcal{X}))^2}{\eta}\left(1 - \frac{\gamma_k}{4\ln 2}\right) + \exp\{M_\mu\delta\}M_\lambda\gamma_k(2\eta + \bar{\lambda}) + \bar{\lambda}M_\lambda d.$$

3. (Control on the interpolation process) For any $k \in 0, \ldots, K-1$ and $t \in [t_k, t_{k+1}]$, we have

$$\mathbb{E}[\|\bar{\boldsymbol{x}}_t^{EI} - \bar{\boldsymbol{x}}_{t_k}^{EI}\|^2] \leq (A' + B')\gamma_k, \tag{122}$$

where

$$A' = \left\{ \frac{\gamma_k}{(4\ln 2)^2} - \frac{2M + \eta}{4\ln 2}\exp\{M_\mu\delta\}M_\lambda + \exp\{2M_\mu\delta\}M_\lambda^2\gamma_k(2M^2 + 1 + (2M + \eta)\bar{\lambda}) \right\}R_k$$
$$B' = -\frac{\gamma_k}{4\ln 2}\exp\{M_\mu\delta\}M_\lambda\frac{(M + \mathrm{diam}(\mathcal{X}))^2}{\eta} + \exp\{2M_\mu\delta\}M_\lambda^2\gamma_k\frac{(2\eta + \bar{\lambda})(M + \mathrm{diam}(\mathcal{X}))^2}{\eta} + \bar{\lambda}M_\lambda d.$$

*Proof.* We prove the three estimates separately.

**1. Control of the approximated process $\hat{\boldsymbol{x}}_t^\leftarrow$.**

Let $\hat{\boldsymbol{x}}_t^\leftarrow$ satisfy the following reverse-time SDE:

$$d\hat{\boldsymbol{x}}_t^\leftarrow = \left\{ -a(T-t)\hat{\boldsymbol{x}}_t^\leftarrow + b^2(T-t)\boldsymbol{v}_{T-t} \right\} dt + b(T-t)d\boldsymbol{B}_t,$$

where the approximated score

$$\boldsymbol{v}_t = \frac{\mu(t)\boldsymbol{\phi}_{\boldsymbol{\theta}^*}\left(\frac{\boldsymbol{x}_t}{\mu(t)} - \lambda(t)\beta\nabla_{\boldsymbol{x}}f\left(\frac{\boldsymbol{x}_t}{\mu(t)}\right), \lambda(t)\right) - \boldsymbol{x}_t}{\mu^2(t)\lambda(t)}.$$

By the results in (96), we have that for any $t \in [0,T]$, $\mathbb{E}[\|\hat{\boldsymbol{x}}_t^{\leftarrow}\|^2] < +\infty$. Hence, apply Itô's lemma (Itô, 1944) to $f(\boldsymbol{x}) = \frac{1}{2}\|\boldsymbol{x}\|^2$, we have

$$
\begin{aligned}
d\left(\frac{1}{2}\|\hat{\boldsymbol{x}}_t^{\leftarrow}\|^2\right) &= \nabla f(\hat{\boldsymbol{x}}_t^{\leftarrow}) \cdot d\hat{\boldsymbol{x}}_t^{\leftarrow} + \frac{1}{2}\operatorname{Tr}\left(b^2(T-t)\nabla^2 f(\hat{\boldsymbol{x}}_t^{\leftarrow})\right)dt = \hat{\boldsymbol{x}}_t^{\leftarrow} \cdot d\hat{\boldsymbol{x}}_t^{\leftarrow} + \frac{1}{2}\operatorname{Tr}\left(b^2(T-t)I\right)dt \\
&= \hat{\boldsymbol{x}}_t^{\leftarrow} \cdot \left\{-a(T-t)\hat{\boldsymbol{x}}_t^{\leftarrow} + b^2(T-t)\boldsymbol{v}_{T-t}\right\}dt + \hat{\boldsymbol{x}}_t^{\leftarrow} \cdot b(T-t)d\boldsymbol{B}_t + \frac{d}{2}b^2(T-t)dt \\
&= \left\{-a(T-t)\|\hat{\boldsymbol{x}}_t^{\leftarrow}\|^2 + b^2(T-t)\langle\boldsymbol{v}_{T-t}, \hat{\boldsymbol{x}}_t^{\leftarrow}\rangle + \frac{d}{2}b^2(T-t)\right\}dt + b(T-t)\langle\hat{\boldsymbol{x}}_t^{\leftarrow}, d\boldsymbol{B}_t\rangle.
\end{aligned}
$$
(123)

Taking expectations on (123), we define $u_t = \frac{1}{2}\mathbb{E}[\|\hat{\boldsymbol{x}}_t^{\leftarrow}\|^2]$. Then $u_0 = \frac{d\bar{\lambda}}{2}$ and

$$\frac{du_t}{dt} = \left\{-a(T-t)\mathbb{E}[\|\hat{\boldsymbol{x}}_t^{\leftarrow}\|^2] + b^2(T-t)\mathbb{E}[\langle\boldsymbol{v}_{T-t}, \hat{\boldsymbol{x}}_t^{\leftarrow}\rangle] + \frac{d}{2}b^2(T-t)\right\}.$$
(124)

Now we bound the inner product term in (124). Denote $\tau = T - t$, we have

$$\langle\boldsymbol{v}_\tau, \hat{\boldsymbol{x}}_t^{\leftarrow}\rangle = \langle\boldsymbol{v}_\tau - \nabla\log p_\tau(\hat{\boldsymbol{x}}_t^{\leftarrow}), \hat{\boldsymbol{x}}_t^{\leftarrow}\rangle + \langle\nabla\log p_\tau(\hat{\boldsymbol{x}}_t^{\leftarrow}), \hat{\boldsymbol{x}}_t^{\leftarrow}\rangle.$$

From the result in Appendix E.2 and Theorem E.3, we have the score approximation error

$$\|\nabla_{\boldsymbol{x}_t}\log p_t(\boldsymbol{x}_t) - \boldsymbol{v}_t\| \le \frac{M(1 + \|\boldsymbol{x}_t\|)}{\sigma^2(t)}.$$

Then, with Cauchy–Schwarz and results in (96), we have

$$
\begin{aligned}
\langle\boldsymbol{v}_\tau - \nabla\log p_\tau(\hat{\boldsymbol{x}}_t^{\leftarrow}), \hat{\boldsymbol{x}}_t^{\leftarrow}\rangle &\le \|\boldsymbol{v}_\tau - \nabla\log p_\tau(\hat{\boldsymbol{x}}_t^{\leftarrow})\| \cdot \|\hat{\boldsymbol{x}}_t^{\leftarrow}\| \\
&\le \frac{M(1 + \|\hat{\boldsymbol{x}}_t^{\leftarrow}\|)}{\sigma^2(\tau)} \cdot \|\hat{\boldsymbol{x}}_t^{\leftarrow}\| \le \frac{M\|\hat{\boldsymbol{x}}_t^{\leftarrow}\|}{\sigma^2(\tau)} + \frac{M\|\hat{\boldsymbol{x}}_t^{\leftarrow}\|^2}{\sigma^2(\tau)},
\end{aligned}
$$
(125)

and

$$\langle\nabla\log p_\tau(\hat{\boldsymbol{x}}_t^{\leftarrow}), \hat{\boldsymbol{x}}_t^{\leftarrow}\rangle \le -\frac{\|\hat{\boldsymbol{x}}_t^{\leftarrow}\|^2}{\sigma^2(\tau)} + \frac{\mu(\tau)\operatorname{diam}(\mathcal{X})\|\hat{\boldsymbol{x}}_t^{\leftarrow}\|}{\sigma^2(\tau)}.$$
(126)

Summing (125) and (126), we have

$$2\langle\boldsymbol{v}_\tau, \hat{\boldsymbol{x}}_t^{\leftarrow}\rangle \le 2(M-1)\frac{\|\hat{\boldsymbol{x}}_t^{\leftarrow}\|^2}{\sigma^2(\tau)} + 2\left(M + \mu(\tau)\operatorname{diam}(\mathcal{X})\right)\frac{\|\hat{\boldsymbol{x}}_t^{\leftarrow}\|}{\sigma^2(\tau)}.$$
(127)

Apply the inequality $2ab \le a^2\eta + b^2/\eta$ with $a = \|\hat{\boldsymbol{x}}_t^{\leftarrow}\|/\sigma(\tau)$, $b = (M + \mu(\tau)\operatorname{diam}(\mathcal{X}))/\sigma(\tau)$, for any $\eta > 0$, we get

$$2\left(M + \mu(\tau)\operatorname{diam}(\mathcal{X})\right)\frac{\|\hat{\boldsymbol{x}}_t^{\leftarrow}\|}{\sigma^2(\tau)} \le \eta\frac{\|\hat{\boldsymbol{x}}_t^{\leftarrow}\|^2}{\sigma^2(\tau)} + \frac{1}{\eta}\frac{(M + \mu(\tau)\operatorname{diam}(\mathcal{X}))^2}{\sigma^2(\tau)}.$$
(128)

Substituting (128) into (127), we have

$$2\langle\boldsymbol{v}_\tau, \hat{\boldsymbol{x}}_t^{\leftarrow}\rangle \le (2M - 2 + \eta)\frac{\|\hat{\boldsymbol{x}}_t^{\leftarrow}\|^2}{\sigma^2(\tau)} + \frac{1}{\eta}\frac{(M + \mu(\tau)\operatorname{diam}(\mathcal{X}))^2}{\sigma^2(\tau)}.$$
(129)

Plug into $du/dt$ in (124), we have

$$
\begin{aligned}
\frac{du_t}{dt} &\le \left\{-2a(\tau)u_t + (2M - 2 + \eta)\frac{b^2(\tau)}{\sigma^2(\tau)}u_t + \frac{(M + \mu(\tau)\operatorname{diam}(\mathcal{X}))^2}{2\eta}\frac{b^2(\tau)}{\sigma^2(\tau)} + \frac{d}{2}b^2(\tau)\right\} \\
&= \left((2M - 2 + \eta)\frac{b^2(\tau)}{\sigma^2(\tau)} - 2a(\tau)\right)u_t + \left(\frac{(M + \mu(\tau)\operatorname{diam}(\mathcal{X}))^2}{2\eta}\frac{b^2(\tau)}{\sigma^2(\tau)} + \frac{d}{2}b^2(\tau)\right).
\end{aligned}
$$
(130)

For simplicity, define

$$A_t = \left( (2M - 2 + \eta) \frac{b^2(\tau)}{\sigma^2(\tau)} - 2a(\tau) \right), B_t = \left( \frac{(M + \mu(\tau)\text{diam}(\mathcal{X}))^2}{2\eta} \frac{b^2(\tau)}{\sigma^2(\tau)} + \frac{d}{2}b^2(\tau) \right).$$

Then

$$\frac{du_t}{dt} \leq A_t u_t + B_t.$$
(131)

Note that

$$\frac{a(\tau)\sigma^2(\tau)}{b^2(\tau)} = \frac{\partial_t \mu(\tau)}{\mu(\tau)} \frac{\lambda(\tau)}{\partial_t \lambda(\tau)} = \frac{\partial_t \log \mu(\tau)}{\partial_t \log \lambda(\tau)} \geq \frac{-M_\mu}{m_\lambda},$$

then we have

$$A_t = \left( (2M - 2 + \eta) \frac{b^2(\tau)}{\sigma^2(\tau)} - 2a(\tau) \right) \leq (2M - 2 + \eta)m_\lambda + 2M_\mu.$$
(132)

We choose $2M + \eta \leq 1 - 2M_\mu/m_\lambda$, then we have $A_t \leq -m_\lambda$. Applying Grönwall's lemma (Fontaine et al., 2021), we have

$$u_t \leq u_0 + \frac{M_\lambda}{m_\lambda} \frac{(M + \text{diam}(\mathcal{X}))^2 + \eta \bar{\lambda} d}{2\eta}.$$

Further, for sufficiently small $M$ such that $M \leq 1/6$, we choose $\eta = 1/6$, then $A_t \leq -1/2$ and

$$u_t \leq 6M_\lambda(M + \text{diam}(\mathcal{X}))^2 + \bar{\lambda} d.$$
(133)

Substituting $u_t = \frac{1}{2}\mathbb{E}[\|\hat{\boldsymbol{x}}_t^{\leftarrow}\|^2]$ into (133) gives

$$\mathbb{E}[\|\hat{\boldsymbol{x}}_t^{\leftarrow}\|^2] \leq 12M_\lambda(M + \text{diam}(\mathcal{X}))^2 + 2\bar{\lambda} d.$$

## 2. Control of the sampling process $y_k$.

Consider the iteration form

$$\boldsymbol{y}_{k+1} = \alpha_{1,k}\boldsymbol{y}_k + \alpha_{2,k}\boldsymbol{P}_k + \alpha_{3,k}\boldsymbol{\xi}_k, \quad k = 1, 2, \dots, K.$$

From the results in (129), we get

$$\langle \boldsymbol{v}_\tau, \boldsymbol{x}_t \rangle \leq (2M - 2 + \eta) \frac{\|\boldsymbol{x}_t\|^2}{2\sigma^2(\tau)} + \frac{1}{\eta} \frac{(M + \mu(\tau)\text{diam}(\mathcal{X}))^2}{2\sigma^2(\tau)}.$$

Recall the definition of $\boldsymbol{v}_t$ in (117), we have

$$\langle \boldsymbol{v}_\tau, \boldsymbol{x}_t \rangle = \frac{\mu(\tau)}{\sigma^2(\tau)} \langle \boldsymbol{P}_\tau, \boldsymbol{x}_t \rangle - \frac{1}{\sigma^2(\tau)} \langle \boldsymbol{x}_t, \boldsymbol{x}_t \rangle.$$
(134)

Substituting (134) into (129), we have

$$\langle \boldsymbol{P}_\tau, \boldsymbol{x}_t \rangle = \frac{\sigma^2(\tau)}{\mu(\tau)} \langle \boldsymbol{v}_\tau, \boldsymbol{x}_t \rangle + \frac{1}{\mu(\tau)} \langle \boldsymbol{x}_t, \boldsymbol{x}_t \rangle \leq \frac{2M + \eta}{2\mu(\tau)} \|\boldsymbol{x}_t\|^2 + \frac{1}{\eta} \frac{(M + \mu(\tau)\text{diam}(\mathcal{X}))^2}{2\mu(\tau)}.$$
(135)

Next, for the norm of $\boldsymbol{v}_\tau$, using the result in Theorem E.3 and Lemma G.1, we get

$$\begin{aligned}
\|\boldsymbol{v}_\tau\|^2 &\leq \|\boldsymbol{v}_\tau - \nabla \log p_\tau(\boldsymbol{x}_t)\|^2 + \|\nabla \log p_\tau(\boldsymbol{x}_t)\|^2 \leq \frac{M^2(1 + \|\boldsymbol{x}_t\|)^2}{\sigma^4(\tau)} + \frac{2\|\boldsymbol{x}_t\|^2}{\sigma^4(\tau)} + \frac{2\mu^2(\tau)\text{diam}(\mathcal{X})^2}{\sigma^4(\tau)} \\
&\leq \frac{(2M^2 + 2)}{\sigma^4(\tau)} \|\boldsymbol{x}_t\|^2 + \frac{2M^2 + 2\mu^2(\tau)\text{diam}(\mathcal{X})^2}{\sigma^4(\tau)}.
\end{aligned}$$
(136)

And substituting $\boldsymbol{v}_t$ as $\boldsymbol{P}_t$, we have

$$\|\boldsymbol{v}_\tau\|^2 = \|\frac{\mu(\tau)\boldsymbol{P}_\tau - \boldsymbol{x}_t}{\sigma^2(\tau)}\|^2 = \frac{\mu^2(\tau)}{\sigma^4(\tau)} \|\boldsymbol{P}_\tau\|^2 - \frac{2\mu(\tau)}{\sigma^2(\tau)} \langle \boldsymbol{P}_\tau, \boldsymbol{x}_t \rangle + \frac{\|\boldsymbol{x}_t\|^2}{\sigma^4(\tau)}.$$
(137)

Then we have

$$
\begin{aligned}
\|\boldsymbol{P}_\tau\|^2 =& \frac{\sigma^4(\tau)}{\mu^2(\tau)}\|\boldsymbol{v}_\tau\|^2 + \frac{2\sigma^2(\tau)}{\mu(\tau)}\langle \boldsymbol{P}_\tau, \boldsymbol{x}_t\rangle - \frac{\|\boldsymbol{x}_t\|^2}{\mu^2(\tau)} \\
\leq& \frac{(2M^2+2)}{\mu^2(\tau)}\|\boldsymbol{x}_t\|^2 + \frac{2M^2}{\mu^2(\tau)} + \frac{2\mathrm{diam}(\mathcal{X})^2}{\mu^2(\tau)} + \frac{(2M+\eta)\sigma^2(\tau)}{\mu^2(\tau)}\|\boldsymbol{x}_t\|^2 + \frac{\sigma^2(\tau)(M+\mu(\tau)\mathrm{diam}(\mathcal{X}))^2}{\eta\mu^2(\tau)} - \frac{\|\boldsymbol{x}_t\|^2}{\mu^2(\tau)} \\
=& \frac{(2M^2+1)+(2M+\eta)\sigma^2(\tau)}{\mu^2(\tau)}\|\boldsymbol{x}_t\|^2 + \frac{(2M^2+2\mathrm{diam}(\mathcal{X})^2)}{\mu^2(\tau)} + \frac{\sigma^2(\tau)(M+\mu(\tau)\mathrm{diam}(\mathcal{X}))^2}{\eta\mu^2(\tau)}.
\end{aligned}
\tag{138}
$$

Now, we compute the expectation of $\|\boldsymbol{y}_{k+1}\|^2$. Since $\boldsymbol{\xi}_k$ is independent of $\boldsymbol{y}_k$ and has mean zero, we have

$$
\begin{aligned}
\mathbb{E}[\|\boldsymbol{y}_{k+1}\|^2] =& \mathbb{E}\Big[\|\alpha_{1,k}\boldsymbol{y}_k + \alpha_{2,k}\boldsymbol{P}_k\|^2\Big] + \alpha_{3,k}^2 d \\
=& \alpha_{1,k}^2\mathbb{E}[\|\boldsymbol{y}_k\|^2] + 2\alpha_{1,k}\alpha_{2,k}\mathbb{E}[\langle \boldsymbol{P}_k, \boldsymbol{y}_k\rangle] + \alpha_{2,k}^2\mathbb{E}[\|\boldsymbol{P}_k\|^2] + \alpha_{3,k}^2 d.
\end{aligned}
\tag{139}
$$

Substituting (135) and (138) into (139), denote $\mu_k = \mu(T-t_k), \sigma_k = \sigma(T-t_k)$, we have the upper bound

$$
\begin{aligned}
\mathbb{E}[\|\boldsymbol{y}_{k+1}\|^2] \leq& \alpha_{1,k}^2\mathbb{E}[\|\boldsymbol{y}_k\|^2] + 2\alpha_{1,k}\alpha_{2,k}\left(\frac{2M+\eta}{2\mu_k}\mathbb{E}[\|\boldsymbol{y}_k\|^2] + \frac{1}{\eta}\frac{(M+\mu_k\mathrm{diam}(\mathcal{X}))^2}{2\mu_k}\right) + \\
& \alpha_{2,k}^2\left(\frac{2M^2+1+(2M+\eta)\sigma_k^2}{\mu_k^2}\mathbb{E}[\|\boldsymbol{y}_k\|^2] + \frac{2M^2+2\mathrm{diam}(\mathcal{X})^2}{\mu_k^2} + \frac{\sigma_k^2(M+\mu_k\mathrm{diam}(\mathcal{X}))^2}{\eta\mu_k^2}\right) + \alpha_{3,k}^2 d \\
\leq& \left\{\alpha_{1,k}^2 + \alpha_{1,k}\alpha_{2,k}\frac{2M+\eta}{\mu_k} + \alpha_{2,k}^2\frac{2M^2+1+(2M+\eta)\sigma_k^2}{\mu_k^2}\right\}\mathbb{E}[\|\boldsymbol{y}_k\|^2] + \\
& \alpha_{1,k}\alpha_{2,k}\frac{(M+\mathrm{diam}(\mathcal{X}))^2}{\eta\mu_k} + \alpha_{2,k}^2\left(\frac{2M^2+2\mathrm{diam}(\mathcal{X})^2}{\mu_k^2} + \frac{\sigma_k^2(M+\mathrm{diam}(\mathcal{X}))^2}{\eta\mu_k^2}\right) + \alpha_{3,k}^2 d.
\end{aligned}
\tag{140}
$$

Under Assumption 4.1, we have

$$
\begin{aligned}
\alpha_{1,k} =& \frac{\lambda_{k+1}\mu_{k+1}}{\lambda_k\mu_k} \leq \exp\left\{(M_\mu - M_\lambda)\gamma_k\right\} \\
\alpha_{2,k} =& \mu_k\frac{\mu_{k+1}}{\mu_k}\left(1 - \frac{\lambda_{k+1}}{\lambda_k}\right) \leq \mu_k\exp\left\{M_\mu\gamma_k\right\}(1-\exp\left\{-M_\lambda\gamma_k\right\}) \\
\alpha_{3,k} =& \mu_{k+1}\sqrt{\lambda_{k+1}}\sqrt{1 - \frac{\lambda_{k+1}}{\lambda_k}} \leq \sqrt{\lambda_{k+1}}\sqrt{1-\exp\left\{-M_\lambda\gamma_k\right\}}.
\end{aligned}
\tag{141}
$$

For $\gamma_k \leq \delta \leq 2\ln 2$, we have

$$
\alpha_{1,k} \leq 1 - \frac{1}{4\ln 2}\gamma_k, \quad \alpha_{2,k}/\mu_k \leq \exp\left\{M_\mu\delta\right\}M_\lambda\gamma_k, \quad \alpha_{3,k}^2 \leq \bar{\lambda}M_\lambda\gamma_k.
\tag{142}
$$

Substituting (142) into (140), we have

$$
\begin{aligned}
\mathbb{E}[\|\boldsymbol{y}_{k+1}\|^2] \leq& \{1 - \frac{1}{2\ln 2}\gamma_k + \frac{1}{(4\ln 2)^2}\gamma_k^2 + (2M+\eta)(1-\frac{1}{4\ln 2}\gamma_k)\exp\left\{M_\mu\delta\right\}M_\lambda\gamma_k + \\
& \exp\left\{2M_\mu\delta\right\}M_\lambda^2\gamma_k^2\left(2M^2+1+(2M+\eta)\bar{\lambda}\right)\}\mathbb{E}[\|\boldsymbol{y}_k\|^2] + \\
& (1 - \frac{1}{4\ln 2}\gamma_k)\exp\left\{M_\mu\delta\right\}M_\lambda\gamma_k\frac{(M+\mathrm{diam}(\mathcal{X}))^2}{\eta} + \\
& \exp\left\{2M_\mu\delta\right\}M_\lambda^2\gamma_k^2\left(2M^2+2\mathrm{diam}(\mathcal{X})^2 + \frac{\bar{\lambda}(M+\mathrm{diam}(\mathcal{X}))^2}{\eta}\right) + \bar{\lambda}M_\lambda d\gamma_k.
\end{aligned}
$$

Denote

$$
\begin{aligned}
A =& \frac{1}{2\ln 2}\left(1 + \frac{\gamma_k}{8\ln 2}\right) + (2M+\eta)(1-\frac{\gamma_k}{4\ln 2})\exp\left\{M_\mu\delta\right\}M_\lambda + \exp\left\{2M_\mu\delta\right\}M_\lambda^2\gamma_k\left(2M^2+1+(2M+\eta)\bar{\lambda}\right) \\
B =& \exp\left\{M_\mu\delta\right\}M_\lambda\frac{(M+\mathrm{diam}(\mathcal{X}))^2}{\eta}\left(1 - \frac{\gamma_k}{4\ln 2}\right) + \exp\left\{M_\mu\delta\right\}M_\lambda\gamma_k(2\eta+\bar{\lambda}) + \bar{\lambda}M_\lambda d.
\end{aligned}
$$

Then we have
$$\mathbb{E}[\|\boldsymbol{y}_{k+1}\|^2] \leq (1 - \gamma_k A)\,\mathbb{E}[\|\boldsymbol{y}_k\|^2] + \gamma_k B.$$

By recursion, we have that, for any $k = 0, 1, \ldots, K$,
$$\mathbb{E}[\|\boldsymbol{y}_{k+1}\|^2] \leq \bar{\lambda}d + B\,(1/A + \delta)\,.$$

Note that the same result holds for $(\bar{\boldsymbol{x}}_t^{EI})_{t \in [0,T]}$.

## 3. Control on the interpolation process.

Note that from the definition of $\bar{\boldsymbol{x}}_t^{EI}$, we have
$$\bar{\boldsymbol{x}}_t^{EI} = \alpha_{1,t_k}\bar{\boldsymbol{x}}_{t_k}^{EI} + \alpha_{2,t_k}\boldsymbol{P}_{t_k} + \alpha_{3,t_k}\boldsymbol{\xi}_k, \quad k = 1, 2, \ldots, K.$$

Therefore,
$$
\begin{aligned}
\mathbb{E}[\|\bar{\boldsymbol{x}}_t^{EI} - \bar{\boldsymbol{x}}_{t_k}^{EI}\|^2] =& \mathbb{E}\left[\|(\alpha_{1,t_k} - 1)\bar{\boldsymbol{x}}_{t_k}^{EI} + \alpha_{2,t_k}\boldsymbol{P}_{t_k}\|^2\right] + \alpha_{3,t_k}^2 d \\
=& (\alpha_{1,t_k} - 1)^2 \mathbb{E}[\|\bar{\boldsymbol{x}}_{t_k}^{EI}\|^2] + 2(\alpha_{1,t_k} - 1)\alpha_{2,t_k}\mathbb{E}[\langle\boldsymbol{P}_{t_k}, \bar{\boldsymbol{x}}_{t_k}^{EI}\rangle] + \alpha_{2,t_k}^2\mathbb{E}[\|\boldsymbol{P}_{t_k}\|^2] + \alpha_{3,t_k}^2 d.
\end{aligned}
\tag{143}
$$

From the former results in (135) and (138), we have
$$
\begin{aligned}
\mathbb{E}[\langle\boldsymbol{P}_{t_k}, \bar{\boldsymbol{x}}_{t_k}^{EI}\rangle] &\leq \frac{2M + \eta}{2\mu_k}\mathbb{E}[\|\bar{\boldsymbol{x}}_{t_k}^{EI}\|^2] + \frac{1}{\eta}\frac{(M + \text{diam}(\mathcal{X}))^2}{2\mu_k} \\
\mathbb{E}[\|\boldsymbol{P}_{t_k}\|^2] &\leq \frac{(2M^2 + 1) + (2M + \eta)\sigma_k^2}{\mu_k^2}\mathbb{E}[\|\bar{\boldsymbol{x}}_{t_k}^{EI}\|^2] + \frac{(2M^2 + 2\text{diam}(\mathcal{X})^2)}{\mu_k^2} + \frac{\bar{\lambda}(M + \text{diam}(\mathcal{X}))^2}{\eta\mu_k^2}.
\end{aligned}
\tag{144}
$$

Substituting (144) into (143), we get
$$
\begin{aligned}
\mathbb{E}[\|\bar{\boldsymbol{x}}_t^{EI} - \bar{\boldsymbol{x}}_{t_k}^{EI}\|^2] \leq& \left\{(\alpha_{1,t_k} - 1)^2 + (\alpha_{1,t_k} - 1)\alpha_{2,t_k}\frac{2M + \eta}{\mu_k} + \alpha_{2,t_k}^2\frac{(2M^2 + 1) + (2M + \eta)\sigma_k^2}{\mu_k^2}\right\}\mathbb{E}[\|\bar{\boldsymbol{x}}_{t_k}^{EI}\|^2] \\
&+ (\alpha_{1,t_k} - 1)\alpha_{2,t_k}\frac{(M + \text{diam}(\mathcal{X}))^2}{\eta\mu_k} + \alpha_{2,t_k}^2\left(\frac{(2M^2 + 2\text{diam}(\mathcal{X})^2)}{\mu_k^2} + \frac{\bar{\lambda}(M + \text{diam}(\mathcal{X}))^2}{\eta\mu_k^2}\right) + \alpha_{3,t_k}^2 d.
\end{aligned}
\tag{145}
$$

Similarly, denote $\tau = t - t_k \leq \gamma_k \leq \delta$, we have
$$
\begin{aligned}
\alpha_{1,t_k} =& \frac{\lambda(T - t)\mu(T - t)}{\lambda(T - t_k)\mu(T - t_k)} \leq \exp\left\{(M_\mu - M_\lambda)\tau\right\} \\
\alpha_{2,t_k} =& \mu(T - t)\left(1 - \frac{\lambda(T - t)}{\lambda(T - t_k)}\right) \leq \mu_k\exp\left\{M_\mu\tau\right\}(1 - \exp\left\{-M_\lambda\tau\right\}) \\
\alpha_{3,t_k} =& \mu(T - t)\sqrt{\lambda(T - t)}\sqrt{1 - \frac{\lambda(T - t)}{\lambda(T - t_k)}} \leq \sqrt{\lambda_{k+1}}\sqrt{1 - \exp\left\{-M_\lambda\tau\right\}}.
\end{aligned}
$$

Further, we get
$$\alpha_{1,t_k} \leq 1 - \frac{1}{4\ln 2}\gamma_k, \quad \alpha_{2,t_k}/\mu_k \leq \exp\left\{M_\mu\delta\right\}M_\lambda\gamma_k, \quad \alpha_{3,t_k}^2 \leq \bar{\lambda}M_\lambda\gamma_k.
\tag{146}$$

Therefore, substituting (146) and the result in (121) into (145), we have
$$
\begin{aligned}
\mathbb{E}[\|\bar{\boldsymbol{x}}_t^{EI} - \bar{\boldsymbol{x}}_{t_k}^{EI}\|^2] \leq& \left\{\frac{\gamma_k^2}{(4\ln 2)^2} - \frac{2M + \eta}{4\ln 2}\exp\left\{M_\mu\delta\right\}M_\lambda\gamma_k + \exp\left\{2M_\mu\delta\right\}M_\lambda^2\gamma_k^2(2M^2 + 1 + (2M + \eta)\bar{\lambda})\right\}R_k \\
&- \frac{\gamma_k}{4\ln 2}\exp\left\{M_\mu\delta\right\}M_\lambda\gamma_k\frac{(M + \text{diam}(\mathcal{X}))^2}{\eta} + \exp\left\{2M_\mu\delta\right\}M_\lambda^2\gamma_k^2\frac{(2\eta + \bar{\lambda})(M + \text{diam}(\mathcal{X}))^2}{\eta}.
\end{aligned}
$$

Denote
$$
\begin{aligned}
A' =& \left\{\frac{\gamma_k}{(4\ln 2)^2} - \frac{2M + \eta}{4\ln 2}\exp\left\{M_\mu\delta\right\}M_\lambda + \exp\left\{2M_\mu\delta\right\}M_\lambda^2\gamma_k(2M^2 + 1 + (2M + \eta)\bar{\lambda})\right\}R_k \\
B' =& -\frac{\gamma_k}{4\ln 2}\exp\left\{M_\mu\delta\right\}M_\lambda\frac{(M + \text{diam}(\mathcal{X}))^2}{\eta} + \exp\left\{2M_\mu\delta\right\}M_\lambda^2\gamma_k\frac{(2\eta + \bar{\lambda})(M + \text{diam}(\mathcal{X}))^2}{\eta} + \bar{\lambda}M_\lambda d.
\end{aligned}
$$

Then we have

$$\mathbb{E}[\|\bar{\boldsymbol{x}}_t^{EI} - \bar{\boldsymbol{x}}_{t_k}^{EI}\|^2] \leq (A' + B')\,\gamma_k.$$

$\square$

This lemma establishes uniform second-moment bounds for the reverse diffusion processes, which are crucial for proving stability and convergence of discrete-time samplers. The bounds guarantee that the reverse process does not explode in finite time, even when using an approximated score. Overall, these moment bounds are key technical tools for deriving non-asymptotic convergence for proximal diffusion sampling algorithms.

### G.3. Contraction of the Tangent Process

To control the error introduced by the discretization and score approximation, we analyze the tangent process associated with the reverse SDE. This process quantifies how small perturbations in the initial condition propagate along the reverse-time flow.

For the forward process $d\boldsymbol{x}_t^{\rightarrow} = \{a(t)\boldsymbol{x}_t^{\rightarrow}\}\,dt + b(t)d\boldsymbol{B}_t$, denote $\pi_\infty = \lim_{t\to\infty} \mathrm{Law}(\boldsymbol{x}_t^{\rightarrow})$. For any $\boldsymbol{x} \in \mathbb{R}^d$ and $s,t \in [0,T]$, $s \leq t$, denote the reverse process as

$$d\boldsymbol{y}_{s,t}^{\boldsymbol{x}} = \left\{-a(T-t)\boldsymbol{y}_{s,t}^{\boldsymbol{x}} + b^2(T-t)\nabla\log\pi_{T-t}(\boldsymbol{y}_{s,t}^{\boldsymbol{x}})\right\}dt + b(T-t)d\boldsymbol{B}_t, \quad \boldsymbol{y}_{s,s}^{\boldsymbol{x}} = \boldsymbol{x}. \tag{147}$$

The approximated reverse process is defined by

$$d\hat{\boldsymbol{y}}_{s,t}^{\boldsymbol{x}} = \left\{-a(T-t)\hat{\boldsymbol{y}}_{s,t}^{\boldsymbol{x}} + b^2(T-t)\boldsymbol{v}_{T-t}\right\}dt + b(T-t)d\boldsymbol{B}_t, \quad \hat{\boldsymbol{y}}_{s,s}^{\boldsymbol{x}} = \boldsymbol{x}, \tag{148}$$

where the approximated score

$$\boldsymbol{v}_{T-t} = \frac{\mu(T-t)\boldsymbol{\phi}_{\boldsymbol{\theta}^*}(\frac{\hat{\boldsymbol{y}}_{s,t}^{\boldsymbol{x}}}{\mu(T-t)} - \lambda(T-t)\beta\nabla f(\frac{\hat{\boldsymbol{y}}_{s,t}^{\boldsymbol{x}}}{\mu(T-t)}), \lambda(T-t)) - \hat{\boldsymbol{y}}_{s,t}^{\boldsymbol{x}}}{\sigma^2(T-t)}.$$

It can also be expressed as

$$d\hat{\boldsymbol{y}}_{s,t}^{\boldsymbol{x}} = \left\{-\left[a(T-t) + \frac{b^2(T-t)}{\sigma^2(T-t)}\right]\hat{\boldsymbol{y}}_{s,t}^{\boldsymbol{x}} + \frac{b^2(T-t)\mu(T-t)}{\sigma^2(T-t)}\boldsymbol{P}_{T-t}\right\}dt + b(T-t)d\boldsymbol{B}_t, \tag{149}$$

where $\lambda(t) = \sigma^2(t)/\mu^2(t)$ and

$$\boldsymbol{P}_{T-t} = \boldsymbol{\phi}_{\boldsymbol{\theta}^*}\left(\frac{\hat{\boldsymbol{y}}_{s,t}^{\boldsymbol{x}}}{\mu(T-t)} - \lambda(T-t)\beta\nabla f(\frac{\hat{\boldsymbol{y}}_{s,t}^{\boldsymbol{x}}}{\mu(T-t)}), \lambda(T-t)\right).$$

From Proposition F.1, the interpolation process on the interval $t \in [s_k, t_{k+1}]$ is defined by

$$d\bar{\boldsymbol{y}}_{s,t}^{\boldsymbol{x}} = \left\{-\left[a(T-t) + \frac{b^2(T-t)}{\sigma^2(T-t)}\right]\bar{\boldsymbol{y}}_{s,t}^{\boldsymbol{x}} + \frac{b^2(T-t)\mu(T-t)}{\sigma^2(T-t)}\boldsymbol{P}_k\right\}dt + b(T-t)d\boldsymbol{B}_t, \bar{\boldsymbol{y}}_{s,s}^{\boldsymbol{x}} = \boldsymbol{x}, \tag{150}$$

where $s_k = \max\{s, t_k\}$ and

$$\boldsymbol{P}_k = \boldsymbol{\phi}_{\boldsymbol{\theta}^*}\left(\frac{\bar{\boldsymbol{y}}_{s,t}^{\boldsymbol{x}}}{\mu(T-t_k)} - \lambda(T-t_k)\beta\nabla f(\frac{\bar{\boldsymbol{y}}_{s,t}^{\boldsymbol{x}}}{\mu(T-t_k)}), \lambda(T-t_k)\right).$$

We introduce the tangent process

$$d\nabla\boldsymbol{y}_{s,t}^{\boldsymbol{x}} = \left\{-a(T-t)I + b^2(T-t)\nabla\log\pi_{T-t}(\boldsymbol{y}_{s,t}^{\boldsymbol{x}})\right\}\nabla\boldsymbol{y}_{s,t}^{\boldsymbol{x}}dt, \tag{151}$$

where $\nabla\boldsymbol{y}_{s,s}^{\boldsymbol{x}} = I$. Then the bound on the approximation and discretization error relies on the following lemma.

**Lemma G.3** ((Del Moral & Singh, 2022)). Under Assumption 3.3, for any $\boldsymbol{x} \in \mathbb{R}^d$ and $s, t \in [0, T], s \leq t$, we have

$$\boldsymbol{y}_{s,t}^{\boldsymbol{x}} - \bar{\boldsymbol{y}}_{s,t}^{\boldsymbol{x}} = \int_s^t \left( \nabla \boldsymbol{y}_{u,t}^{\bar{\boldsymbol{y}}_{s,u}^{\boldsymbol{x}}} \right)^\top \Delta \boldsymbol{b}_u ((\bar{\boldsymbol{y}}_{s,v}^{\boldsymbol{x}})_{v \in [s,t]}) du, \tag{152}$$

where for any $u \in [s_k, t_{k+1}]$ and $(\boldsymbol{w}_v)_{v \in [s,t]}$, the drift

$$\begin{aligned}
\boldsymbol{b}_u(\boldsymbol{w}) &= -a(T-t)\boldsymbol{w} + b^2(T-t)\nabla \log \pi_{T-t}(\boldsymbol{w}), \\
\bar{\boldsymbol{b}}_u(\boldsymbol{w}) &= -\left[ a(T-t) + \frac{b^2(T-t)}{\sigma^2(T-t)} \right] w + \frac{b^2(T-t)\mu(T-t)}{\sigma^2(T-t)} \boldsymbol{P}_k(\boldsymbol{w}),
\end{aligned} \tag{153}$$

and $\Delta \boldsymbol{b}_u(\boldsymbol{w}) = \boldsymbol{b}_u(\boldsymbol{w}) - \bar{\boldsymbol{b}}_u(\boldsymbol{w})$.

Then our goal is to control $\nabla \boldsymbol{y}_{s,t}^{\boldsymbol{x}}$ and $\Delta \boldsymbol{b}_s((\bar{\boldsymbol{y}}_{s,t}^{\boldsymbol{x}})_{t \in [s,T]})$ for any $\boldsymbol{x} \in \mathbb{R}^d$ and $s, t \in [0, T], s \leq t$.

**Lemma G.4.** Under Assumption 3.3 and 4.1, for any $s, t \in [0, T]$ and $\boldsymbol{x}, \boldsymbol{x}' \in \mathbb{R}^d$, denote

$$\begin{aligned}
\bar{t} &= T - \lambda^{-1} \left( \frac{\operatorname{diam}^2(\mathcal{X}) M_\lambda}{m_\lambda - M_\mu} \right), \\
G_K &= \bar{t}/2 - (m_\lambda - M_\mu)(T - \bar{t}) \exp \left[ M_\lambda(T - \bar{t}) \right].
\end{aligned} \tag{154}$$

Then we have

$$\|\nabla \boldsymbol{y}_{0,t}^{\boldsymbol{x}}\|^2 \leq \begin{cases} \exp\left[-t/2\right], & t \leq \bar{t}, \\ \exp\left[-G_K\right], & t \geq \bar{t}, \end{cases} \tag{155}$$

and

$$\mathcal{W}_1(\delta_{\boldsymbol{x}} Q_t, \delta_{\boldsymbol{x}'} Q_t) \leq \begin{cases} \exp\left[-t/2\right] \|\boldsymbol{x} - \boldsymbol{x}'\|, & t \leq \bar{t}, \\ \exp\left[-G_K\right] \|\boldsymbol{x} - \boldsymbol{x}'\|, & t \geq \bar{t}. \end{cases} \tag{156}$$

*Proof.* Firstly, we consider a simple case where the target distribution $\pi = \delta_0$, and then $\operatorname{diam}(\mathcal{X}) = 0$. We will show that

$$\mathcal{W}_1(\delta_{\boldsymbol{x}} Q_t, \delta_{\boldsymbol{x}'} Q_t) \leq \mathbb{E}\left[ \|\boldsymbol{y}_{0,t}^{\boldsymbol{x}} - \boldsymbol{y}_{0,t}^{\boldsymbol{x}'}\| \right] \leq \exp\left\{ -(m_\lambda - M_\mu)t \right\} \|\boldsymbol{x} - \boldsymbol{x}'\|. \tag{157}$$

Especially, when $t = T$, the bound becomes

$$\mathcal{W}_1(\delta_{\boldsymbol{x}} Q_T, \delta_{\boldsymbol{x}'} Q_T) \leq 0, \tag{158}$$

implying finite-time convergence.

Consider the backward process $(\boldsymbol{y}_{0,t}^{\boldsymbol{x}})$ with the associated semi-group $(Q_t)_{t \in [0,T]}$ and its associated tangent process

$$d\nabla \boldsymbol{y}_{0,t}^{\boldsymbol{x}} = \left\{ -a(T-t)I + b^2(T-t)\nabla \log \pi_t(\boldsymbol{y}_{0,t}^{\boldsymbol{x}}) \right\} \nabla \boldsymbol{y}_{0,t}^{\boldsymbol{x}} dt, \quad \nabla \boldsymbol{y}_{0,0}^{\boldsymbol{x}} = I.$$

Then, for any $t \in [0, T]$, we have

$$\|\boldsymbol{y}_{0,t}^{\boldsymbol{x}} - \boldsymbol{y}_{0,t}^{\boldsymbol{x}'}\| \leq \int_0^1 \|\nabla \boldsymbol{y}_{0,t}^{\boldsymbol{z}_r}\| \, dr \cdot \|\boldsymbol{x} - \boldsymbol{x}'\|, \tag{159}$$

where $\boldsymbol{z}_r = r\boldsymbol{x} + (1-r)\boldsymbol{x}'$. Using the results in Lemma G.1, we have that, for any $\boldsymbol{z}$,

$$\langle \boldsymbol{z}, \nabla^2 \log \pi_t(\boldsymbol{y}_{0,t}^{\boldsymbol{x}})\boldsymbol{z} \rangle \leq -\frac{2\sigma^2(t) - \mu^2(t)\operatorname{diam}(\mathcal{X})^2}{2\sigma^4(t)} \|\boldsymbol{z}\|^2 = -\frac{\|\boldsymbol{z}\|^2}{\sigma^2(t)}.$$

Then we have

$$\begin{aligned}
\langle \boldsymbol{z}, \left( -a(T-t)I + b^2(T-t)\nabla \log \pi_{T-t}(\boldsymbol{y}_{s,t}^{\boldsymbol{x}}) \right) \boldsymbol{z} \rangle &\leq -a(T-t)\|\boldsymbol{z}\|^2 - \frac{b^2(T-t)}{\sigma^2(T-t)} \|\boldsymbol{z}\|^2 \\
&= -\left( a(T-t) + \frac{b^2(T-t)}{\sigma^2(T-t)} \right) \|\boldsymbol{z}\|^2.
\end{aligned}$$

And the evolution of $\|\nabla \boldsymbol{y}_{0,t}^{\boldsymbol{x}}\|^2$ can be bounded by

$$\frac{d}{dt}\|\nabla \boldsymbol{y}_{0,t}^{\boldsymbol{x}}\|^2 \le -2\left(a(T-t) + \frac{b^2(T-t)}{\sigma^2(T-t)}\right)\|\nabla \boldsymbol{y}_{0,t}^{\boldsymbol{x}}\|^2. \tag{160}$$

Integrating (160) from 0 to $t$ and using the initial condition $\nabla \boldsymbol{y}_{0,0}^{\boldsymbol{x}} = I$, we obtain

$$\|\nabla \boldsymbol{y}_{0,t}^{\boldsymbol{x}}\|^2 \le \exp\left[-2\int_0^t \left(a(T-s) + \frac{b^2(T-s)}{\sigma^2(T-s)}\right)ds\right]. \tag{161}$$

Denote $\tau = T - t$, and recall that $\mu(\tau) = \exp\left\{\int_0^\tau a(s)ds\right\}$. We have $\frac{d}{d\tau}\log \mu(\tau) = a(\tau)$, and then

$$\int_{T-t}^T a(\tau)d\tau = \log \mu(T) - \log \mu(T-t) = \log \frac{\mu(T)}{\mu(T-t)}.$$

Further, note that $\sigma^2(\tau) = \mu^2(\tau)\int_0^\tau (b(s)/\mu(s))^2 ds$. Denote $\lambda(\tau) = \int_0^\tau (b(s)/\mu(s))^2 ds = \sigma^2(\tau)/\mu^2(\tau)$. Then we have

$$\frac{b^2(\tau)}{\sigma^2(\tau)} = \frac{b^2(\tau)}{\mu^2(\tau)\lambda(\tau)} = \frac{1}{\lambda(\tau)}\frac{b^2(\tau)}{\mu^2(\tau)}.$$

Also, we have $\lambda'(\tau) = (b(\tau)/\mu(\tau))^2$, and then

$$\frac{b^2(\tau)}{\sigma^2(\tau)} = \frac{1}{\lambda(\tau)}\frac{b^2(\tau)}{\mu^2(\tau)} = \frac{\lambda'(\tau)}{\lambda(\tau)} = \frac{d}{d\tau}\log \lambda(\tau).$$

So we have

$$\int_{T-t}^T \frac{b^2(\tau)}{\sigma^2(\tau)}d\tau = \log \lambda(T) - \log \lambda(T-t) = \log \frac{\lambda(T)}{\lambda(T-t)}.$$

Then with Assumption 4.1, we get

$$\exp\left[-\int_0^t \left(a(T-s) + \frac{b^2(T-s)}{\sigma^2(T-s)}\right)ds\right] = \frac{\lambda(T-t)}{\lambda(T)}\frac{\mu(T-t)}{\mu(T)} \le \exp\left\{-(m_\lambda - M_\mu)t\right\}. \tag{162}$$

Substituting (162) into (161), we obtain

$$\|\nabla \boldsymbol{y}_{0,t}^{\boldsymbol{x}}\|^2 \le \exp\left[-2\int_0^t \left(a(T-s) + \frac{b^2(T-s)}{\sigma^2(T-s)}\right)ds\right] \le \exp\left\{-2(m_\lambda - M_\mu)t\right\}, \tag{163}$$

and when $t \to T$, $\|\nabla \boldsymbol{y}_{0,t}^{\boldsymbol{x}}\|^2 \to 0$.

Substituting this into the tangent error (159), we obtain

$$\|\boldsymbol{y}_{0,t}^{\boldsymbol{x}} - \boldsymbol{y}_{0,t}^{\boldsymbol{x}'}\| \le \int_0^1 \|\nabla \boldsymbol{y}_{0,t}^{\boldsymbol{z}_r}\|\, dr \cdot \|\boldsymbol{x} - \boldsymbol{x}'\| \le \exp\left\{-(m_\lambda - M_\mu)t\right\}\|\boldsymbol{x} - \boldsymbol{x}'\|.$$

By the definition of the 1-Wasserstein distance, we have

$$\mathcal{W}_1(\delta_{\boldsymbol{x}}Q_t, \delta_{\boldsymbol{x}'}Q_t) \le \mathbb{E}\left[\|\boldsymbol{y}_{0,t}^{\boldsymbol{x}} - \boldsymbol{y}_{0,t}^{\boldsymbol{x}'}\|\right] \le \exp\left\{-(m_\lambda - M_\mu)t\right\}\|\boldsymbol{x} - \boldsymbol{x}'\|.$$

Especially, when $t = T$, the bound becomes $\|\nabla \boldsymbol{y}_{0,t}^{\boldsymbol{x}}\|^2 \le 0$ and then

$$\mathcal{W}_1(\delta_{\boldsymbol{x}}Q_T, \delta_{\boldsymbol{x}'}Q_T) \le \mathbb{E}\left[\|\boldsymbol{y}_{0,t}^{\boldsymbol{x}} - \boldsymbol{y}_{0,t}^{\boldsymbol{x}'}\|\right] \le 0.$$

Hence, $\mathcal{W}_1(\delta_{\boldsymbol{x}}Q_T, \delta_{\boldsymbol{x}'}Q_T) = 0$ for all $\boldsymbol{x}, \boldsymbol{x}' \in \mathbb{R}^d$, implying that $\delta_{\boldsymbol{x}}Q_T = \delta_{\boldsymbol{x}'}Q_T$. In particular, taking $\boldsymbol{x}' \sim \delta_0$ and noting that $\pi = \delta_0$ is invariant under the forward-backward dynamics, we have

$$\delta_{\boldsymbol{x}}Q_T = \pi.$$

This holds for any $x \in \mathbb{R}^d$, implying finite-time convergence.

Next, we will control the growth of the tangent process for a general target $\pi$. For the tangent process

$$d\nabla y_{0,t}^{\boldsymbol{x}} = \left\{-a(T-t)I + b^2(T-t)\nabla \log \pi_t(y_{0,t}^{\boldsymbol{x}})\right\} \nabla y_{0,t}^{\boldsymbol{x}} dt, \quad \nabla y_{0,0}^{\boldsymbol{x}} = I,$$

using the results in Lemma G.1, we have that, for any $z \in \mathcal{X}$,

$$\langle z, \nabla^2 \log \pi_t(y_{0,t}^{\boldsymbol{x}})z\rangle \leq -\frac{2\sigma^2(t) - \mu^2(t)\mathrm{diam}(\mathcal{X})^2}{2\sigma^4(t)}\|z\|^2 = -\frac{\|z\|^2}{\sigma^2(t)} + \frac{\mu^2(t)\mathrm{diam}(\mathcal{X})^2\|z\|^2}{2\sigma^4(t)}.$$

Then we have

$$\langle z, \left(-a(T-t)I + b^2(T-t)\nabla \log \pi_{T-t}(y_{s,t}^{\boldsymbol{x}})\right) z\rangle$$
$$\leq -a(T-t)\|z\|^2 - \frac{b^2(T-t)}{\sigma^2(T-t)}\|z\|^2 + \frac{b^2(T-t)\mu^2(T-t)\mathrm{diam}(\mathcal{X})^2\|z\|^2}{2\sigma^4(T-t)}$$
$$= -\left(a(T-t) + \frac{b^2(T-t)}{\sigma^2(T-t)} - \frac{b^2(T-t)\mu^2(T-t)\mathrm{diam}(\mathcal{X})^2}{2\sigma^4(T-t)}\right)\|z\|^2.$$

And the evolution of $\|\nabla y_{0,t}^{\boldsymbol{x}}\|^2$ can be bounded by

$$\frac{d}{dt}\|\nabla y_{0,t}^{\boldsymbol{x}}\|^2 \leq -2\left(a(T-t) + \frac{b^2(T-t)}{\sigma^2(T-t)} - \frac{b^2(T-t)\mu^2(T-t)\mathrm{diam}(\mathcal{X})^2}{2\sigma^4(T-t)}\right)\|\nabla y_{0,t}^{\boldsymbol{x}}\|^2. \tag{164}$$

Integrating (164) from 0 to $t$ and using the initial condition $\nabla y_{0,0}^{\boldsymbol{x}} = I$, we obtain

$$\|\nabla y_{0,t}^{\boldsymbol{x}}\|^2 \leq \exp\left[-2\int_0^t \left(a(T-s) + \frac{b^2(T-s)}{\sigma^2(T-s)} - \frac{b^2(T-s)\mu^2(T-s)\mathrm{diam}(\mathcal{X})^2}{2\sigma^4(T-s)}\right)ds\right]. \tag{165}$$

Note that

$$(\ln \mu(t))' = a(t), \quad (\ln \lambda(t))' = \frac{b^2(t)}{\sigma^2(t)}, \quad \lambda'(t) = \frac{b^2(t)}{\mu^2(t)}, \quad \left(\frac{1}{\lambda(t)}\right)' = -\frac{\lambda'(t)}{\lambda^2(t)} = -\frac{b^2(t)\mu^2(t)}{\sigma^4(t)},$$

then we have

$$\int_0^t \left(a(T-s) + \frac{b^2(T-s)}{\sigma^2(T-s)} - \frac{b^2(T-s)\mu^2(T-s)\mathrm{diam}(\mathcal{X})^2}{2\sigma^4(T-s)}\right)ds$$
$$= \left\{\ln\left(\mu(s)\lambda(s)\right) + \frac{\mathrm{diam}^2(\mathcal{X})}{2}\frac{1}{\lambda(s)}\right\}\Bigg|_{s=T-t}^{T} = \ln\left(\frac{\mu(T)\lambda(T)}{\mu(T-t)\lambda(T-t)}\right) + \frac{\mathrm{diam}^2(\mathcal{X})}{2}\left(\frac{1}{\lambda(T)} - \frac{1}{\lambda(T-t)}\right). \tag{166}$$

Substituting 166 into (165), we have

$$\|\nabla y_{0,t}^{\boldsymbol{x}}\|^2 \leq \exp\left[-2\ln\left(\frac{\mu(T)\lambda(T)}{\mu(T-t)\lambda(T-t)}\right) - \mathrm{diam}^2(\mathcal{X})\left(\frac{1}{\lambda(T)} - \frac{1}{\lambda(T-t)}\right)\right]$$
$$\leq \left(\frac{\mu(T-t)\lambda(T-t)}{\mu(T)\lambda(T)}\right)^2 \exp\left[\mathrm{diam}^2(\mathcal{X})\left(\frac{1}{\lambda(T-t)} - \frac{1}{\lambda(T)}\right)\right]$$
$$\leq \exp\left[-2(m_\lambda - M_\mu)t\right]\exp\left[\frac{\mathrm{diam}^2(\mathcal{X})M_\lambda t}{\lambda(T-t)}\right] = \exp\left[-\bar{G}_K t\right],$$

where $\bar{G}_K = 2(m_\lambda - M_\mu) - \mathrm{diam}^2(\mathcal{X})M_\lambda/\lambda(T-t)$. Note that as $t \to T$, $\bar{G}_K$ may be negative. We denote

$$\bar{t} = T - \lambda^{-1}\left(\frac{\mathrm{diam}^2(\mathcal{X})M_\lambda}{m_\lambda - M_\mu}\right),$$

then we have $\bar{G}_K \geq (m_\lambda - M_\mu) \geq 1/2$ when $t \leq \bar{t}$ and then

$$\|\nabla y_{0,t}^{\boldsymbol{x}}\|^2 \leq \exp\left[-t/2\right]. \tag{167}$$

When $t \geq \bar{t}$, we have

$$
\begin{aligned}
\|\nabla \boldsymbol{y}_{0,t}^{\boldsymbol{x}}\|^2 &\leq \exp\left[-\bar{t}/2\right] \exp\left[-2\ln\left(\frac{\mu(T-\bar{t})\lambda(T-\bar{t})}{\mu(T-t)\lambda(T-t)}\right) - \operatorname{diam}^2(\mathcal{X})\left(\frac{1}{\lambda(T-\bar{t})} - \frac{1}{\lambda(T-t)}\right)\right] \\
&\leq \exp\left[-\bar{t}/2\right] \left(\frac{\mu(T-t)\lambda(T-t)}{\mu(T-\bar{t})\lambda(T-\bar{t})}\right)^2 \exp\left[\operatorname{diam}^2(\mathcal{X})\left(\frac{1}{\lambda(T-t)} - \frac{1}{\lambda(T-\bar{t})}\right)\right] \\
&\leq \exp\left[-\bar{t}/2\right] \exp\left[-2(m_\lambda - M_\mu)(t - \bar{t})\right] \exp\left[\frac{\operatorname{diam}^2(\mathcal{X})M_\lambda(t-\bar{t})}{\lambda(T-t)}\right] \\
&\leq \exp\left[-\bar{t}/2 + \frac{\operatorname{diam}^2(\mathcal{X})M_\lambda(T-\bar{t})}{\lambda(T-\bar{t})}\exp\left[M_\lambda(T-\bar{t})\right]\right].
\end{aligned}
\tag{168}
$$

Denote

$$
\bar{t} = T - \lambda^{-1}\left(\frac{\operatorname{diam}^2(\mathcal{X})M_\lambda}{m_\lambda - M_\mu}\right),
$$
$$
G_K = \bar{t}/2 - (m_\lambda - M_\mu)(T - \bar{t})\exp\left[M_\lambda(T-\bar{t})\right].
$$

Combine (168) and (167), we have

$$
\|\nabla \boldsymbol{y}_{0,t}^{\boldsymbol{x}}\|^2 \leq \begin{cases} \exp\left[-t/2\right], & t \leq \bar{t}, \\ \exp\left[-G_K\right], & t \geq \bar{t}. \end{cases}
\tag{169}
$$

Substituting (169) into the tangent error, we obtain

$$
\|\boldsymbol{y}_{0,t}^{\boldsymbol{x}} - \boldsymbol{y}_{0,t}^{\boldsymbol{x}'}\| \leq \int_0^1 \|\nabla \boldsymbol{y}_{0,t}^{\boldsymbol{z}_r}\| \, dr \cdot \|\boldsymbol{x} - \boldsymbol{x}'\| \leq \begin{cases} \exp\left[-t/2\right]\|\boldsymbol{x} - \boldsymbol{x}'\|, & t \leq \bar{t}, \\ \exp\left[-G_K\right]\|\boldsymbol{x} - \boldsymbol{x}'\|, & t \geq \bar{t}. \end{cases}
$$

By the definition of the 1-Wasserstein distance, we have that, for any $t \in [0, T]$

$$
\mathcal{W}_1(\delta_{\boldsymbol{x}}Q_t, \delta_{\boldsymbol{x}'}Q_t) \leq \mathbb{E}\left[\|\boldsymbol{y}_{0,t}^{\boldsymbol{x}} - \boldsymbol{y}_{0,t}^{\boldsymbol{x}'}\|\right] \leq \begin{cases} \exp\left[-t/2\right]\|\boldsymbol{x} - \boldsymbol{x}'\|, & t \leq \bar{t}, \\ \exp\left[-G_K\right]\|\boldsymbol{x} - \boldsymbol{x}'\|, & t \geq \bar{t}. \end{cases}
$$

$\square$

Lemma G.4 quantifies the norm of the tangent process and reveals the contractivity in the 1-Wasserstein distance, which is crucial for proving the convergence of the reverse sampling process.

Our next goal is to control $\Delta \boldsymbol{b}_s((\bar{\boldsymbol{y}}_{s,t}^{\boldsymbol{x}})_{t \in [s,T]})$ for any $\boldsymbol{x} \in \mathbb{R}^d$ and $s, t \in [0, T], s \leq t$. Recall that for $\boldsymbol{w} = (\boldsymbol{w}_v)_{v \in [t_k, t_{k+1}]}$, we have

$$
\begin{aligned}
\boldsymbol{b}_u(\boldsymbol{w}) &= -a(T-u)\boldsymbol{w}_u + b^2(T-u)\nabla \log \pi_{T-u}(\boldsymbol{w}_u), \\
\bar{\boldsymbol{b}}_u(\boldsymbol{w}) &= -\left[a(T-u) + \frac{b^2(T-u)}{\sigma^2(T-u)}\right]\boldsymbol{w}_u + \frac{b^2(T-u)\mu(T-u)}{\sigma^2(T-u)}\boldsymbol{P}_k(\boldsymbol{w}_{t_k}),
\end{aligned}
\tag{170}
$$

and $\Delta \boldsymbol{b}_u(\boldsymbol{w}) = \boldsymbol{b}_u(\boldsymbol{w}) - \bar{\boldsymbol{b}}_u(\boldsymbol{w})$. We introduce the intermediate drift functions that

$$
\begin{aligned}
\boldsymbol{b}_u^{(1)}(\boldsymbol{w}) &= -a(T-u)\boldsymbol{w}_u + b^2(T-u)\nabla \log \pi_{T-u}(\boldsymbol{w}_u) = \boldsymbol{b}_u(\boldsymbol{w}) \\
\boldsymbol{b}_u^{(2)}(\boldsymbol{w}) &= -a(T-t)\boldsymbol{w}_u + b^2(T-t)\boldsymbol{v}_{T-u}(\boldsymbol{w}_u) \\
&= -\left[a(T-u) + \frac{b^2(T-u)}{\sigma^2(T-u)}\right]\boldsymbol{w}_u + \frac{b^2(T-u)\mu(T-u)}{\sigma^2(T-u)}\boldsymbol{P}_{T-u}(\boldsymbol{w}_u) \\
\boldsymbol{b}_u^{(3)}(\boldsymbol{w}) &= -\left[a(T-u) + \frac{b^2(T-u)}{\sigma^2(T-u)}\right]\boldsymbol{w}_u + \frac{b^2(T-u)\mu(T-u)}{\sigma^2(T-u)}\boldsymbol{P}_{T-t_k}(\boldsymbol{w}_u) \\
\boldsymbol{b}_u^{(4)}(\boldsymbol{w}) &= -\left[a(T-u) + \frac{b^2(T-u)}{\sigma^2(T-u)}\right]\boldsymbol{w}_u + \frac{b^2(T-u)\mu(T-u)}{\sigma^2(T-u)}\boldsymbol{P}_{T-t_k}(\boldsymbol{w}_{t_k}) = \bar{\boldsymbol{b}}_u(\boldsymbol{w}),
\end{aligned}
\tag{171}
$$

where

$$\boldsymbol{P}_{T-u}(\boldsymbol{w}_u) = \phi_{\boldsymbol{\theta}^*}\left(\frac{\boldsymbol{w}_u}{\mu(T-u)} - \lambda(T-u)\beta\nabla f(\frac{\boldsymbol{w}_u}{\mu(T-u)}), \lambda(T-u)\right),$$

$$\boldsymbol{v}_{T-u}(\boldsymbol{w}_u) = \frac{\mu(T-u)\boldsymbol{P}_{T-u} - \boldsymbol{w}_u}{\mu^2(T-u)\lambda(T-u)}.$$

(172)

The discrepancy between the ideal reverse drift $\boldsymbol{b}_u$ and the discretized drift $\bar{\boldsymbol{b}}_u$ can be decomposed into three parts, corresponding to score approximation $B_1$, proximal-time discretization $B_2$, and proximal-spatial discretization $B_3$, i.e.,

$$\|\Delta\boldsymbol{b}_u(\boldsymbol{w})\| \le \underbrace{\|\boldsymbol{b}_u^{(1)}(\boldsymbol{w}) - \boldsymbol{b}_u^{(2)}(\boldsymbol{w})\|}_{B_1} + \underbrace{\|\boldsymbol{b}_u^{(2)}(\boldsymbol{w}) - \boldsymbol{b}_u^{(3)}(\boldsymbol{w})\|}_{B_2} + \underbrace{\|\boldsymbol{b}_u^{(3)}(\boldsymbol{w}) - \boldsymbol{b}_u^{(4)}(\boldsymbol{w})\|}_{B_3}.$$

(173)

The following lemma quantifies each part.

**Lemma G.5.** Under Assumption 3.3 and 4.1, for any $s, u \in [0, T]$ and $\boldsymbol{x} \in \mathbb{R}^d$, we have

$$\mathbb{E}\left[\|\Delta\boldsymbol{b}_u((\bar{\boldsymbol{y}}_{s,v}^{\boldsymbol{x}})_{v\in[s,T]})\|\right] \le W_{b,1}\gamma_k + W_{b,2}M + W_{b,3}\gamma_k^{1/2},$$

(174)

where

$$W_{b,1} = \exp\{M_\mu T\}\left[\left(1 + \beta\bar{\lambda}L + m_\lambda\right)\sqrt{R_k} + m_\lambda R_p\right] + \beta G_f\left(\bar{\lambda}M_\lambda + m_\lambda\right)$$

$$W_{b,2} = M_\lambda(\sqrt{R_k} + 1)$$

$$W_{b,3} = \exp\{M_\mu T\}M_\lambda(1 + \bar{\lambda}\beta L)\sqrt{A' + B'}.$$

(175)

*Proof.* **Term $B_1$: score approximation error.**

From the definition in (171), we have

$$\|\boldsymbol{b}_u^{(1)}(\boldsymbol{w}) - \boldsymbol{b}_u^{(2)}(\boldsymbol{w})\| = b^2(T-u)\|\nabla\log p_{T-u}(\boldsymbol{w}_u) - \boldsymbol{v}_{T-u}(\boldsymbol{w}_u)\|.$$

(176)

Note that from the result in Appendix E.2 and Theorem E.3, we have the score approximation error

$$\|\nabla\log p_{T-u}(\boldsymbol{w}_u) - \boldsymbol{v}_{T-u}(\boldsymbol{w}_u)\| \le \frac{M(1 + \|\boldsymbol{w}_u\|)}{\sigma^2(T-u)}.$$

(177)

Substituting (177) into the above (176), we have

$$\|\boldsymbol{b}_u^{(1)}(\boldsymbol{w}) - \boldsymbol{b}_u^{(2)}(\boldsymbol{w})\| \le \frac{Mb^2(T-u)}{\sigma^2(T-u)}(1 + \|\boldsymbol{w}_u\|).$$

(178)

**Term $B_2$: proximal-time discretization error.** Next, from the definition in (171), we have

$$\|\boldsymbol{b}_u^{(2)}(\boldsymbol{w}) - \boldsymbol{b}_u^{(3)}(\boldsymbol{w})\| = \frac{b^2(T-u)\mu(T-u)}{\sigma^2(T-u)}\|\boldsymbol{P}_{T-u}(\boldsymbol{w}_u) - \boldsymbol{P}_{T-t_k}(\boldsymbol{w}_u)\|.$$

(179)

Denote $\boldsymbol{A}(s, \boldsymbol{w}_u) = \frac{\boldsymbol{w}_u}{\mu(T-u)} - \lambda(T-u)\beta\nabla f(\frac{\boldsymbol{w}_u}{\mu(T-u)})$ where $s = T - u$ and $s_k = T - t_k$. Then we have

$$\|\boldsymbol{P}_{T-u}(\boldsymbol{w}_u) - \boldsymbol{P}_{T-t_k}(\boldsymbol{w}_u)\| = \|\text{Prox}_f^{\lambda(s)}(\boldsymbol{A}(s, \boldsymbol{w}_u)) - \text{Prox}_f^{\lambda(s_k)}(\boldsymbol{A}(s_k, \boldsymbol{w}_u))\|.$$

(180)

Let $\boldsymbol{p}_s = \text{Prox}_f^{\lambda(s)}(A(s, \boldsymbol{w}_u))$ and $\boldsymbol{p}_{s_k} = \text{Prox}_f^{\lambda(s_k)}(A(s_k, \boldsymbol{w}_u))$. By the optimality conditions of the proximal operator, we have

$$\boldsymbol{p}_s = \boldsymbol{A}(s, \boldsymbol{w}_u) - \lambda(s)\nabla f(\boldsymbol{p}_s),$$

$$\boldsymbol{p}_{s_k} = \boldsymbol{A}(s_k, \boldsymbol{w}_u) - \lambda(s_k)\nabla f(\boldsymbol{p}_{s_k}).$$

Taking difference, we have

$$\boldsymbol{p}_s - \boldsymbol{p}_{s_k} = \left(A(s, \boldsymbol{w}_u) - A(s_k, \boldsymbol{w}_u)\right) - \left(\lambda(s)\nabla f(\boldsymbol{p}_s) - \lambda(s_k)\nabla f(\boldsymbol{p}_{s_k})\right),$$

and then

$$\boldsymbol{p}_s - \boldsymbol{p}_{s_k} + \lambda(s)\big(\nabla f(\boldsymbol{p}_s) - \nabla f(\boldsymbol{p}_{s_k})\big) + \big(\lambda(s) - \lambda(s_k)\big)\nabla f(\boldsymbol{p}_{s_k}) = \boldsymbol{A}(s, \boldsymbol{w}_u) - \boldsymbol{A}(s_k, \boldsymbol{w}_u). \tag{181}$$

Taking the inner product of (181) with $\boldsymbol{p}_s - \boldsymbol{p}_{s_k}$, we have

$$
\begin{aligned}
&\langle \boldsymbol{p}_s - \boldsymbol{p}_{s_k}, \boldsymbol{p}_s - \boldsymbol{p}_{s_k}\rangle + \lambda(s)\langle\nabla f(\boldsymbol{p}_s) - \nabla f(\boldsymbol{p}_{s_k}), \boldsymbol{p}_s - \boldsymbol{p}_{s_k}\rangle + \big(\lambda(s) - \lambda(s_k)\big)\langle\nabla f(\boldsymbol{p}_{s_k}), \boldsymbol{p}_s - \boldsymbol{p}_{s_k}\rangle \\
&= \langle \boldsymbol{A}(s, \boldsymbol{w}_u) - \boldsymbol{A}(s_k, \boldsymbol{w}_u), \boldsymbol{p}_s - \boldsymbol{p}_{s_k}\rangle.
\end{aligned}
\tag{182}
$$

Since $f$ is $m$-strongly convex, we have

$$\langle\nabla f(\boldsymbol{p}_s) - \nabla f(\boldsymbol{p}_{s_k}), \boldsymbol{p}_s - \boldsymbol{p}_{s_k}\rangle \geq m\|\boldsymbol{p}_s - \boldsymbol{p}_{s_k}\|^2. \tag{183}$$

Applying (183) and the Cauchy–Schwarz inequality to (182) yields

$$\|\boldsymbol{p}_s - \boldsymbol{p}_{s_k}\| \leq (1 + \lambda(s)m)\|\boldsymbol{p}_s - \boldsymbol{p}_{s_k}\| \leq \|\boldsymbol{A}(s, \boldsymbol{w}_u) - \boldsymbol{A}(s_k, \boldsymbol{w}_u)\| + |\lambda(s) - \lambda(s_k)| \cdot \|\nabla f(\boldsymbol{p}_{s_k})\|. \tag{184}$$

Next we bound $\|\boldsymbol{A}(s, \boldsymbol{w}_u) - \boldsymbol{A}(s_k, \boldsymbol{w}_u)\|$. From the definition, we have

$$
\begin{aligned}
\boldsymbol{A}(s, \boldsymbol{w}_u) - \boldsymbol{A}(s_k, \boldsymbol{w}_u) &= \boldsymbol{w}_u\left(\frac{1}{\mu(s)} - \frac{1}{\mu(s_k)}\right) - \beta\left[\lambda(s)\nabla f\left(\frac{\boldsymbol{w}_u}{\mu(s)}\right) - \lambda(s_k)\nabla f\left(\frac{\boldsymbol{w}_u}{\mu(s_k)}\right)\right] \\
&= \boldsymbol{w}_u\left(\frac{1}{\mu(s)} - \frac{1}{\mu(s_k)}\right) - \beta\left[(\lambda(s) - \lambda(s_k))\nabla f\left(\frac{\boldsymbol{w}_u}{\mu(s)}\right) + \lambda(s_k)\left(\nabla f\left(\frac{\boldsymbol{w}_u}{\mu(s)}\right) - \nabla f\left(\frac{\boldsymbol{w}_u}{\mu(s_k)}\right)\right)\right].
\end{aligned}
$$

Taking norms and using the triangle inequality, we have

$$
\begin{aligned}
&\|\boldsymbol{A}(s, \boldsymbol{w}_u) - \boldsymbol{A}(s_k, \boldsymbol{w}_u)\| \leq \\
&\|\boldsymbol{w}_u\| \cdot \left|\frac{1}{\mu(s)} - \frac{1}{\mu(s_k)}\right| + \beta|\lambda(s) - \lambda(s_k)| \cdot \left\|\nabla f\left(\frac{\boldsymbol{w}_u}{\mu(s)}\right)\right\| + \beta\lambda(s_k)\left\|\nabla f\left(\frac{\boldsymbol{w}_u}{\mu(s)}\right) - \nabla f\left(\frac{\boldsymbol{w}_u}{\mu(s_k)}\right)\right\|.
\end{aligned}
\tag{185}
$$

Since $f$ is $L$-smooth, we have

$$\left\|\nabla f\left(\frac{\boldsymbol{w}_u}{\mu(s)}\right) - \nabla f\left(\frac{\boldsymbol{w}_u}{\mu(s_k)}\right)\right\| \leq L\left\|\frac{\boldsymbol{w}_u}{\mu(s)} - \frac{\boldsymbol{w}_u}{\mu(s_k)}\right\| \leq L\|\boldsymbol{w}_u\|\left(\frac{1}{\mu(s_k)} - \frac{1}{\mu(s)}\right). \tag{186}$$

Moreover, denote

$$G_f = \max_{\|\boldsymbol{x}\| \leq \|\boldsymbol{w}_u\|\exp\{M_\mu T\}} \|\nabla f(\boldsymbol{x})\|. \tag{187}$$

Substituting (186) and (187) into (185), then we have

$$\|\boldsymbol{A}(s, \boldsymbol{w}_u) - \boldsymbol{A}(s_k, \boldsymbol{w}_u)\| \leq \|\boldsymbol{w}_u\|\left(1 + \beta\bar{\lambda}L\right)\left(\frac{1}{\mu(s_k)} - \frac{1}{\mu(s)}\right) + \beta G_f|\lambda(s) - \lambda(s_k)|. \tag{188}$$

From the optimality condition, we have $\nabla f(\boldsymbol{p}_{s_k}) = \frac{\boldsymbol{A}(s_k, \boldsymbol{w}_u) - \boldsymbol{p}_{s_k}}{\lambda(s_k)}$, and then

$$\|\nabla f(\boldsymbol{p}_{s_k})\| \leq \frac{\|\boldsymbol{A}(s_k, \boldsymbol{w}_u)\| + \|\boldsymbol{p}_{s_k}\|}{\lambda(s_k)} \leq \frac{\|\boldsymbol{w}_u\| + \|\boldsymbol{p}_{s_k}\|}{\mu(s_k)\lambda(s_k)} + \frac{\beta G_f}{\lambda(s_k)}. \tag{189}$$

Since $f$ is $m$-strongly convex, there exists a lower bound, and the proximal minimization problem satisfies

$$f(\boldsymbol{p}_{s_k}) + \frac{1}{2\lambda(s_k)}\|\boldsymbol{p}_{s_k} - \boldsymbol{A}(s_k, \boldsymbol{w}_u)\|^2 \leq f(\boldsymbol{A}(s_k, \boldsymbol{w}_u)).$$

Since $\boldsymbol{A}(s_k, \boldsymbol{w}_u)$ is bounded, $f(\boldsymbol{A}(s_k, \boldsymbol{w}_u))$ is bounded. Therefore, $\|\boldsymbol{p}_{s_k}\|$ is bounded, i.e.,

$$\|\boldsymbol{p}_{s_k}\| \leq R_p. \tag{190}$$

Thus, substituting (190) into (189), we obtain

$$\|\nabla f(\boldsymbol{p}_{s_k})\| \leq \frac{\|\boldsymbol{w}_u\| + R_p}{\mu(s_k)\lambda(s_k)} + \frac{\beta G_f}{\lambda(s_k)}. \tag{191}$$

Combine (188) and (191) with (184) we have

$$\begin{aligned}
\|\boldsymbol{p}_s - \boldsymbol{p}_{s_k}\| \leq & \|\boldsymbol{w}_u\| \left(1 + \beta\bar{\lambda}L\right)\left(\frac{1}{\mu(s_k)} - \frac{1}{\mu(s)}\right) + \beta G_f \left(\lambda(s_k) - \lambda(s)\right) + \left(1 - \frac{\lambda(s)}{\lambda(s_k)}\right)\left(\frac{\|\boldsymbol{w}_u\| + R_p}{\mu(s_k)} + \beta G_f\right) \\
\leq & \frac{\|\boldsymbol{w}_u\|\left(1 + \beta\bar{\lambda}L\right)}{\mu(T - t_k)}\gamma_k + \beta G_f \bar{\lambda}M_\lambda \gamma_k + \left(\frac{\|\boldsymbol{w}_u\| + R_p}{\mu(T - t_k)} + \beta G_f\right)\left(1 - \exp\left\{-\gamma_k m_\lambda\right\}\right).
\end{aligned}$$

Denote

$$G_{2,3} = \left(\|\boldsymbol{w}_u\|\left(1 + m_\lambda + \beta\bar{\lambda}L\right) + m_\lambda R_p\right)\exp\left\{M_\mu T\right\} + \beta G_f\left(\bar{\lambda}M_\lambda + m_\lambda\right),$$

then we have

$$\|\boldsymbol{p}_s - \boldsymbol{p}_{s_k}\| \leq G_{2,3}\gamma_k. \tag{192}$$

Substituting (180) and (192) into (179), we obtain

$$\|\boldsymbol{b}_u^{(2)}(\boldsymbol{w}) - \boldsymbol{b}_u^{(3)}(\boldsymbol{w})\| \leq \frac{b^2(T - u)\mu(T - u)}{\sigma^2(T - u)}G_{2,3}\gamma_k.$$

**Term $B_3$: proximal-spatial discretization error.**

For the last term in (171), we have

$$\|\boldsymbol{b}_u^{(3)}(\boldsymbol{w}) - \boldsymbol{b}_u^{(4)}(\boldsymbol{w})\| = \frac{b^2(T - u)\mu(T - u)}{\sigma^2(T - u)}\|\boldsymbol{P}_{T - t_k}(\boldsymbol{w}_u) - \boldsymbol{P}_{T - t_k}(\boldsymbol{w}_{t_k})\|, \tag{193}$$

and

$$\|\boldsymbol{P}_{T - t_k}(\boldsymbol{w}_u) - \boldsymbol{P}_{T - t_k}(\boldsymbol{w}_{t_k})\| = \|\mathrm{Prox}_f^{\lambda(s_k)}(\boldsymbol{A}(s_k, \boldsymbol{w}_u)) - \mathrm{Prox}_f^{\lambda(s_k)}(\boldsymbol{A}(s_k, \boldsymbol{w}_{t_k}))\|.$$

Let $s_k = T - t_k$, we will show that $\boldsymbol{P}_{s_k}(\boldsymbol{w})$ is Lipschitz continuous with respect to $\boldsymbol{w}$.

Since $f$ is proper, closed, and convex, the proximal operator $\mathrm{Prox}_f^{\lambda(s_k)}$ is nonexpansive, i.e.,

$$\|\mathrm{Prox}_f^{\lambda(s_k)}(\boldsymbol{x}) - \mathrm{Prox}_f^{\lambda(s_k)}(\boldsymbol{y})\| \leq \|\boldsymbol{x} - \boldsymbol{y}\|, \quad \forall \boldsymbol{x}, \boldsymbol{y}.$$

Then we have

$$\|\boldsymbol{P}_{s_k}(\boldsymbol{w}_u) - \boldsymbol{P}_{s_k}(\boldsymbol{w}_{t_k})\| \leq \|\boldsymbol{A}(s_k, \boldsymbol{w}_u) - \boldsymbol{A}(s_k, \boldsymbol{w}_{t_k})\|. \tag{194}$$

For any $\boldsymbol{w}_1, \boldsymbol{w}_2$,

$$\begin{aligned}
\|\boldsymbol{A}(s_k, \boldsymbol{w}_1) - \boldsymbol{A}(s_k, \boldsymbol{w}_2)\| &= \left\|\frac{\boldsymbol{w}_1 - \boldsymbol{w}_2}{\mu(s_k)} - \lambda(s_k)\beta\left(\nabla f\left(\frac{\boldsymbol{w}_1}{\mu(s_k)}\right) - \nabla f\left(\frac{\boldsymbol{w}_2}{\mu(s_k)}\right)\right)\right\| \\
&\leq \frac{1}{\mu(s_k)}\|\boldsymbol{w}_1 - \boldsymbol{w}_2\| + \lambda(s_k)\beta\left\|\nabla f\left(\frac{\boldsymbol{w}_1}{\mu(s_k)}\right) - \nabla f\left(\frac{\boldsymbol{w}_2}{\mu(s_k)}\right)\right\|.
\end{aligned} \tag{195}$$

By the $L$-smoothness of $f$, we have

$$\left\|\nabla f\left(\frac{\boldsymbol{w}_1}{\mu(s_k)}\right) - \nabla f\left(\frac{\boldsymbol{w}_2}{\mu(s_k)}\right)\right\| \leq L\left\|\frac{\boldsymbol{w}_1 - \boldsymbol{w}_2}{\mu(s_k)}\right\| = \frac{L}{\mu(s_k)}\|\boldsymbol{w}_1 - \boldsymbol{w}_2\|. \tag{196}$$

Substituting (196) into (195), we have

$$\|\boldsymbol{A}(s_k, \boldsymbol{w}_1) - \boldsymbol{A}(s_k, \boldsymbol{w}_2)\| \leq \left(\frac{1}{\mu(s_k)} + \lambda(s_k)\beta\frac{L}{\mu(s_k)}\right)\|\boldsymbol{w}_1 - \boldsymbol{w}_2\| \leq \left(1 + \bar{\lambda}\beta L\right)\exp\left\{M_\mu T\right\}\|\boldsymbol{w}_1 - \boldsymbol{w}_2\|. \tag{197}$$

Combine (194) and (196) with (193), we have

$$\|\boldsymbol{b}_u^{(3)}(\boldsymbol{w}) - \boldsymbol{b}_u^{(4)}(\boldsymbol{w})\| \leq \frac{b^2(T-u)\mu(T-u)}{\sigma^2(T-u)}\|\boldsymbol{A}(s_k, \boldsymbol{w}_u) - \boldsymbol{A}(s_k, \boldsymbol{w}_{t_k})\| \leq \frac{b^2(T-u)\mu(T-u)}{\sigma^2(T-u)}G_{3,4}\|\boldsymbol{w}_u - \boldsymbol{w}_{t_k}\|, \quad (198)$$

where $G_{3,4} = \left(1 + \bar{\lambda}\beta L\right)\exp\{M_\mu T\}$.

**Combining the three terms.**

Summing the three estimates, we obtain

$$\begin{aligned}
\|\Delta \boldsymbol{b}_u(\boldsymbol{w})\| \leq & \|\boldsymbol{b}_u^{(1)}(\boldsymbol{w}) - \boldsymbol{b}_u^{(2)}(\boldsymbol{w})\| + \|\boldsymbol{b}_u^{(2)}(\boldsymbol{w}) - \boldsymbol{b}_u^{(3)}(\boldsymbol{w})\| + \|\boldsymbol{b}_u^{(3)}(\boldsymbol{w}) - \boldsymbol{b}_u^{(4)}(\boldsymbol{w})\| \\
\leq & \frac{Mb^2(T-u)}{\sigma^2(T-u)}\left(1 + \|\boldsymbol{w}_u\|\right) + \frac{b^2(T-u)\mu(T-u)}{\sigma^2(T-u)}G_{2,3}\gamma_k + \frac{b^2(T-u)\mu(T-u)}{\sigma^2(T-u)}G_{3,4}\|\boldsymbol{w}_u - \boldsymbol{w}_{t_k}\| \\
\leq & \left(M_\lambda M + \left(1 + \beta\bar{\lambda}L + m_\lambda\right)\exp\{M_\mu T\}\gamma_k\right)\|\boldsymbol{w}_u\| + M_\lambda\left(1 + \bar{\lambda}\beta L\right)\exp\{M_\mu T\}\|\boldsymbol{w}_u - \boldsymbol{w}_{t_k}\| + \\
& \left(\beta G_f\left(\bar{\lambda}M_\lambda + m_\lambda\right) + m_\lambda R_p\exp\{M_\mu T\}\right)\gamma_k + M_\lambda M.
\end{aligned} \quad (199)$$

Recall that from Lemma G.2 and (121), (122), we have

$$\begin{aligned}
\mathbb{E}[\|\boldsymbol{w}_u\|^2] &\leq R_k := \bar{\lambda}d + B\left(1/A + \delta\right), \\
\mathbb{E}[\|\boldsymbol{w}_u - \boldsymbol{w}_{t_k}\|^2] &\leq (A' + B')\gamma_k,
\end{aligned} \quad (200)$$

where

$$\begin{aligned}
A &= \frac{1}{2\ln 2}\left(1 + \frac{\gamma_k}{8\ln 2}\right) + (2M + \eta)(1 - \frac{\gamma_k}{4\ln 2})\exp\{M_\mu\delta\}M_\lambda + \exp\{2M_\mu\delta\}M_\lambda^2\gamma_k\left(2M^2 + 1 + (2M + \eta)\bar{\lambda}\right) \\
B &= \exp\{M_\mu\delta\}M_\lambda\frac{(M + \mathrm{diam}(\mathcal{X}))^2}{\eta}\left(1 - \frac{\gamma_k}{4\ln 2}\right) + \exp\{M_\mu\delta\}M_\lambda\gamma_k(2\eta + \bar{\lambda}) + \bar{\lambda}M_\lambda d, \\
A' &= \left\{\frac{\gamma_k}{(4\ln 2)^2} - \frac{2M + \eta}{4\ln 2}\exp\{M_\mu\delta\}M_\lambda + \exp\{2M_\mu\delta\}M_\lambda^2\gamma_k(2M^2 + 1 + (2M + \eta)\bar{\lambda})\right\}R_k \\
B' &= -\frac{\gamma_k}{4\ln 2}\exp\{M_\mu\delta\}M_\lambda\frac{(M + \mathrm{diam}(\mathcal{X}))^2}{\eta} + \exp\{2M_\mu\delta\}M_\lambda^2\gamma_k\frac{(2\eta + \bar{\lambda})(M + \mathrm{diam}(\mathcal{X}))^2}{\eta} + \bar{\lambda}M_\lambda d.
\end{aligned}$$

Substituting (200) into (199), denote

$$\begin{aligned}
W_{b,1} &= \exp\{M_\mu T\}\left[\left(1 + \beta\bar{\lambda}L + m_\lambda\right)\sqrt{R_k} + m_\lambda R_p\right] + \beta G_f\left(\bar{\lambda}M_\lambda + m_\lambda\right) \\
W_{b,2} &= M_\lambda(\sqrt{R_k} + 1) \\
W_{b,3} &= \exp\{M_\mu T\}M_\lambda(1 + \bar{\lambda}\beta L)\sqrt{A' + B'},
\end{aligned}$$

then we have

$$\mathbb{E}\left[\|\Delta\boldsymbol{b}_u((\bar{\boldsymbol{y}}_{s,v}^{\boldsymbol{x}})_{v\in[s,T]})\|\right] \leq W_{b,1}\gamma_k + W_{b,2}M + W_{b,3}\gamma_k^{1/2}.$$

$\square$

Lemma G.5 establishes a bound on the expected norm of the drift difference $\Delta\boldsymbol{b}_u$. It quantifies how three types of local approximations contribute to the error in the drift of the interpolated reverse process. The bound reveals that the total local drift error scales as $\mathcal{O}(M + \delta^{1/2})$ with weights increasing as $\mathcal{O}(\exp\{T\})$, which directly informs the choice of stepsize $\delta$ and score approximation accuracy $M$ relative to the time length $T$.

Compared with corresponding results in (De Bortoli, 2022), we obtain a tighter bound in (174) with the properties of proximal operator. This eliminates the $\mathcal{O}(\epsilon^{-2})$ scaling factor associated with the early-stopping time $\epsilon$, building the foundation for the strengthening of the convergence.

## G.4. Proof of Theorem 4.3

This subsection presents the complete proof of Theorem 4.3, which establishes the non-asymptotic convergence of the proximal diffusion sampler.

For simplicity, we denote $\bar{\boldsymbol{x}}_{t_k}^{EI}$ as $\boldsymbol{y}_k$, and then we have

$$\boldsymbol{y}_{k+1} = \alpha_{1,k}\boldsymbol{y}_k + \alpha_{2,k}\boldsymbol{P}_k + \alpha_{3,k}\boldsymbol{\xi}_k, \quad \boldsymbol{\xi}_k \sim \mathcal{N}(\boldsymbol{0}, I).$$

Denote the distribution of $\boldsymbol{y}_k$ as $\pi_\infty R_k$, where we choose $\boldsymbol{y}_0 \sim \pi_\infty$ and $R_k$ the transition kernel associated with the conditional distribution of $\boldsymbol{y}_k|\boldsymbol{y}_0$.

Our goal is to bound the Wasserstein-1 distance between $\pi_\infty R_K$ (the output of the algorithm after $K$ steps) and the target $\pi$. Denote $(P_t)_{t\in[0,T]}$ as the semi-group associated with $\boldsymbol{x}_t^{\rightarrow}$, and $(Q_t)_{t\in[0,T]}$ the semi-group associated with $\boldsymbol{x}_t^{\leftarrow}$. We decompose the error into two parts:

$$\mathcal{W}_1(\pi_\infty R_K, \pi) \leq \underbrace{\mathcal{W}_1(\pi_\infty R_K, \pi_\infty Q_T)}_{I_1} + \underbrace{\mathcal{W}_1(\pi_\infty Q_T, \pi)}_{I_2}, \tag{201}$$

where the first term $I_1$ corresponds to the discretization and score estimation error. The second term $I_2$ corresponds to the convergence of the continuous-time exact reverse process.

In contrast to existing convergence results in (De Bortoli, 2022; Li et al., 2023; 2024a), our framework eliminates the early-stopping error in the convergence result, i.e., the third term $I_3$ in the following decomposition:

$$\mathcal{W}_1(\pi_\infty R_K, \pi) \leq \underbrace{\mathcal{W}_1(\pi_\infty R_K, \pi_\infty Q_{T-\epsilon})}_{I_1} + \underbrace{\mathcal{W}_1(\pi_\infty Q_{T-\epsilon}, \pi P_\epsilon)}_{I_2} + \underbrace{\mathcal{W}_1(\pi P_\epsilon, \pi)}_{I_3}.$$

This error typically arises from partitioning the sampling interval into $[0, \epsilon]$ and $[\epsilon, T]$ to address nonsmoothness near $t = 0$. As $t$ approaches zero, the true score $\nabla \log \pi_t(\boldsymbol{x}_t)$ diverges, creating a singularity that must be avoided. In addition, one cannot simply force $\epsilon \to 0$, as the errors $I_1$ and $I_2$ are related with $\epsilon$ and tend to $\infty$ as $\epsilon \to 0$.

In our framework, this issue is naturally resolved by replacing the traditional score with the well-defined Moreau score $\nabla \log \pi_t^\lambda(\boldsymbol{x}_t)$, leading to enhanced error bounds of $I_1$ and $I_2$.

**Proposition G.6.** Under Assumption 3.3 and 4.1, we have

$$\mathcal{W}_1(\pi_\infty R_K, \pi_\infty Q_T) \leq \left(1 + \exp\left[-G_K\right]\lambda^{-1}\left(\frac{\operatorname{diam}^2(\mathcal{X})M_\lambda}{m_\lambda - M_\mu}\right)\right)\left(C'_{1,1}\gamma_k^{1/2} + C'_{1,2}M\right), \tag{202}$$

where

$$C'_{1,1} = W_{b,1}\gamma_k^{1/2} + W_{b,3}, \quad C'_{1,2} = W_{b,2},$$

and

$$W_{b,1} = \exp\left\{M_\mu T\right\}\left[\left(1 + \beta\bar{\lambda}L + m_\lambda\right)\sqrt{R_k} + m_\lambda R_p\right] + \beta G_f\left(\bar{\lambda}M_\lambda + m_\lambda\right)$$
$$W_{b,2} = M_\lambda(\sqrt{R_k} + 1)$$
$$W_{b,3} = \exp\left\{M_\mu T\right\}M_\lambda(1 + \bar{\lambda}\beta L)\sqrt{A' + B'}.$$

*Proof.* Consider the reverse diffusion process:

$$d\boldsymbol{x}_t^{\leftarrow} = \left\{a(t)\boldsymbol{x}_t^{\leftarrow} - b^2(t)\nabla_{\boldsymbol{x}_t^{\leftarrow}}\log\pi_t(\boldsymbol{x}_t^{\leftarrow})\right\}dt + b(t)d\bar{\boldsymbol{B}}_t,$$

and the proximal diffusion sampling process:

$$\bar{\boldsymbol{x}}_{k+1} = \alpha_{1,k}\bar{\boldsymbol{x}}_k + \alpha_{2,k}\boldsymbol{P}_k + \alpha_{3,k}\boldsymbol{\xi}_k,$$

where $\boldsymbol{\xi}_k \sim \mathcal{N}(\boldsymbol{0}, I)$, $\tau_k = T - t_k$, $\boldsymbol{P}_k$, $\alpha_{1,k}$, $\alpha_{2,k}$, and $\alpha_{3,k}$ are defined in (28) and (29), respectively.

Recall that the proximal diffusion sampling process corresponds to the interpolation process on the interval $t \in [t_k, t_{k+1}]$:

$$d\bar{\boldsymbol{x}}_t^{EI} = \left\{ -\left[ a(T-t) + \frac{b^2(T-t)}{\sigma^2(T-t)} \right] \bar{\boldsymbol{x}}_t^{EI} + \frac{b^2(T-t)\mu(T-t)}{\sigma^2(T-t)} \boldsymbol{P}_k \right\} dt + b(T-t) d\boldsymbol{B}_t.$$

From Lemma G.3, we have

$$\|\boldsymbol{x}_T^{\leftarrow} - \bar{\boldsymbol{x}}_K\| = \|\boldsymbol{x}_T^{\leftarrow} - \bar{\boldsymbol{x}}_T^{EI}\| \leq \int_0^T \|\nabla \boldsymbol{y}_{u,T}^{\bar{\boldsymbol{y}}_{0,u}}\| \|\Delta \boldsymbol{b}_u((\bar{\boldsymbol{y}}_{0,v})_{v \in [0,T]})\| du.$$

Combining this result and Lemma G.4, we have

$$\|\boldsymbol{x}_T^{\leftarrow} - \bar{\boldsymbol{x}}_K\| \leq \int_0^{\bar{t}} \|\nabla \boldsymbol{y}_{u,T}^{\bar{\boldsymbol{y}}_{0,u}}\| \|\Delta \boldsymbol{b}_u((\bar{\boldsymbol{y}}_{0,v})_{v \in [0,T]})\| du + \int_{\bar{t}}^T \|\nabla \boldsymbol{y}_{u,T}^{\bar{\boldsymbol{y}}_{0,u}}\| \|\Delta \boldsymbol{b}_u((\bar{\boldsymbol{y}}_{0,v})_{v \in [0,T]})\| du$$

$$\leq \int_0^{\bar{t}} \exp\left[-\bar{t}/2\right] \|\Delta \boldsymbol{b}_u((\bar{\boldsymbol{y}}_{0,v})_{v \in [0,T]})\| du + \int_{\bar{t}}^{t_K} \exp\left[-G_K\right] \|\Delta \boldsymbol{b}_u((\bar{\boldsymbol{y}}_{0,v})_{v \in [0,T]})\| du. \tag{203}$$

where

$$\bar{t} = T - \lambda^{-1} \left( \frac{\mathrm{diam}^2(\mathcal{X}) M_\lambda}{m_\lambda - M_\mu} \right),$$

$$G_K = \bar{t}/2 - (m_\lambda - M_\mu)(T - \bar{t}) \exp\left[M_\lambda(T - \bar{t})\right].$$

From the definition of Wasserstein-1 distance, we have

$$\mathcal{W}_1(\pi_\infty R_K, \pi_\infty Q_T) \leq \mathbb{E}\left[\|\boldsymbol{x}_T^{\leftarrow} - \bar{\boldsymbol{x}}_K\|\right] \leq (\bar{t} \exp\left[-\bar{t}/2\right] + \exp\left[-G_K\right](t_K - \bar{t})) \mathbb{E}\left[\|\Delta \boldsymbol{b}_u((\bar{\boldsymbol{y}}_{0,v})_{v \in [0,T]})\|\right] du. \tag{204}$$

Then with Lemma G.5, we have

$$\mathbb{E}\left[\|\Delta \boldsymbol{b}_u((\bar{\boldsymbol{y}}_{s,v}^{\boldsymbol{x}})_{v \in [s,T]})\|\right] \leq W_{b,1}\gamma_k + W_{b,2}M + W_{b,3}\gamma_k^{1/2}, \tag{205}$$

where

$$W_{b,1} = \exp\left\{M_\mu T\right\} \left[ \left(1 + \beta\bar{\lambda}L + m_\lambda\right) \sqrt{R_k} + m_\lambda R_p \right] + \beta G_f \left(\bar{\lambda}M_\lambda + m_\lambda\right)$$

$$W_{b,2} = M_\lambda(\sqrt{R_k} + 1)$$

$$W_{b,3} = \exp\left\{M_\mu T\right\} M_\lambda(1 + \bar{\lambda}\beta L)\sqrt{A' + B'}.$$

Substituting (205) into (204), we have

$$\mathcal{W}_1(\pi_\infty R_K, \pi_\infty Q_T) \leq (\bar{t} \exp\left[-\bar{t}/2\right] + \exp\left[-G_K\right](T - \bar{t})) \mathbb{E}\left[\|\Delta \boldsymbol{b}_u((\bar{\boldsymbol{y}}_{0,v})_{v \in [0,T]})\|\right] du$$

$$\leq (\bar{t} \exp\left[-\bar{t}/2\right] + \exp\left[-G_K\right](T - \bar{t})) \left(C'_{1,1}\gamma_k^{1/2} + C'_{1,2}M\right)$$

$$\leq \left(1 + \exp\left[-G_K\right]\lambda^{-1} \left(\frac{\mathrm{diam}^2(\mathcal{X})M_\lambda}{m_\lambda - M_\mu}\right)\right) \left(C'_{1,1}\gamma_k^{1/2} + C'_{1,2}M\right),$$

where $C'_{1,1}$ and $C'_{1,2}$ are defined in (202).

$\square$

Next, we control the second term $\mathcal{W}_1(\pi_\infty Q_T, \pi)$.

**Proposition G.7.** Under Assumption 3.3 and 4.1, we have

$$\mathcal{W}_1(\pi_\infty Q_T, \pi) \leq \exp\left[-G_K\right](\sqrt{d} + \mathrm{diam}(\mathcal{X})) \exp\left[-m_\mu T\right],$$

where

$$G_K = \bar{t}/2 - (m_\lambda - M_\mu)(T - \bar{t}) \exp\left[M_\lambda(T - \bar{t})\right].$$

*Proof.* Note that $\pi = \pi P_T Q_T$, with Lemma G.4, we have

$$\mathcal{W}_1(\pi_\infty Q_T, \pi) = \mathcal{W}_1(\pi_\infty Q_T, \pi P_T Q_T) \leq \exp[-G_K] \mathcal{W}_1(\pi_\infty, \pi P_T),$$

where $G_K = \bar{t}/2 - (m_\lambda - M_\mu)(T - \bar{t}) \exp[M_\lambda(T - \bar{t})]$.

To control $\mathcal{W}_1(\pi_\infty, \pi P_T)$, we use a synchronous coupling, i.e. we set $(\boldsymbol{Y}_t, \boldsymbol{Z}_t)_{t \in [0,T]}$ such that

$$\mathrm{d}\boldsymbol{Y}_t = a(t)\boldsymbol{Y}_t \mathrm{d}t + b(t)\mathrm{d}\boldsymbol{B}_t, \quad \mathrm{d}\boldsymbol{Z}_t = a(t)\boldsymbol{Z}_t \mathrm{d}t + b(t)\mathrm{d}\boldsymbol{B}_t,$$

where $(\boldsymbol{B}_t)_{t \in [0,T]}$ is a $d$-dimensional Brownian motion and $\boldsymbol{Y}_0 \sim \pi, \boldsymbol{Z}_0 \sim \pi_\infty$. We have that for any $t \in [0, T]$, $\boldsymbol{Z}_t \sim \pi_\infty$. In addition, denoting $u_t = \mathbb{E}[\|\boldsymbol{Y}_t - \boldsymbol{Z}_t\|]$ for any $t \in [0, T]$, we have

$$u_t \leq u_0 \exp\left\{\int_0^t a(s)ds\right\} = u_0\mu(t).$$

Therefore, combining this result and above, we get that

$$\mathcal{W}_1(\pi_\infty Q_{t_K}, \pi P_{T-t_K}) \leq \exp[-G_K]\,\mu(t_K)\mathcal{W}_1(\pi, \pi_\infty) \leq \exp[-G_K]\,(\sqrt{d} + \mathrm{diam}(\mathcal{X}))\exp[-m_\mu t_K].$$

$\square$

Finally, we give our main convergence result, which corresponds to Theorem 4.3.

**Theorem G.8.** Under Assumption 3.3 and 4.1, let $\mathrm{Law}(\bar{\boldsymbol{x}}_k) = \pi_\infty R_k$ denote the distribution of $\bar{\boldsymbol{x}}_k$ in (27), where $\bar{\boldsymbol{x}}_0 \sim \pi_\infty := \lim_{t \to \infty} \mathrm{Law}(\boldsymbol{x}_t^\rightarrow)$ and $R_k$ is the transition kernel associated with $p(\bar{\boldsymbol{x}}_k | \bar{\boldsymbol{x}}_0)$. Define the diameter of the constraint set as $\mathrm{diam}(\mathcal{X}) = \sup\{\|\boldsymbol{x} - \boldsymbol{x}'\| : \boldsymbol{x}, \boldsymbol{x}' \in \mathcal{X}\}$. Suppose that

$$T \geq \lambda^{-1}\left(\frac{\mathrm{diam}^2(\mathcal{X})M_\lambda}{m_\lambda - M_\mu}\right), \quad \beta \geq 2, \quad M \leq 1/6, \quad \delta \leq 2\ln 2.$$

Then there exist $D_1, D_2, D_3$ such that

$$\mathcal{W}_1\left(\mathrm{Law}(\bar{\boldsymbol{x}}_K), \pi\right) \leq D_1\exp[-m_\mu T] + D_2 M + D_3\delta^{1/2},$$

where

$$D_1 = W_e(\sqrt{d} + \mathrm{diam}(\mathcal{X})),$$

$$D_2 = \left(1 + W_e\lambda^{-1}\left(\frac{\mathrm{diam}^2(\mathcal{X})M_\lambda}{m_\lambda - M_\mu}\right)\right)W_\lambda,$$

$$D_3 = \left(1 + W_e\lambda^{-1}\left(\frac{\mathrm{diam}^2(\mathcal{X})M_\lambda}{m_\lambda - M_\mu}\right)\right)(W_f + W_\mu),$$

and

$$W_e = \exp[-\bar{t}/2 + (m_\lambda - M_\mu)(T - \bar{t})\exp[M_\lambda(T - \bar{t})]],$$

$$W_\lambda = \left(\sqrt{\bar{\lambda}d + B(1/A + \delta)} + 1\right)M_\lambda,$$

$$W_\mu = \left((1 + \beta\bar{\lambda}L + m_\lambda)\sqrt{R_k} + m_\lambda R_p + M_\lambda(1 + \bar{\lambda}\beta L)\sqrt{A' + B'}\right)\delta\exp\{M_\mu T\},$$

$$W_f = \beta\delta\left(\bar{\lambda}M_\lambda + m_\lambda\right)\max_{\|\boldsymbol{x}\| \leq \|\boldsymbol{w}_u\|}\|\nabla f(\boldsymbol{x})\|.$$

*Proof.* From (201), the total sampling error is decomposed into two parts. From the result in Proposition G.6, we have

$$I_1 = \mathcal{W}_1(\pi_\infty R_K, \pi_\infty Q_T) \leq \left(1 + \exp[-G_K]\lambda^{-1}\left(\frac{\mathrm{diam}^2(\mathcal{X})M_\lambda}{m_\lambda - M_\mu}\right)\right)\left(C'_{1,1}\gamma_k^{1/2} + C'_{1,2}M\right). \tag{206}$$

Further, from Proposition G.7, we have

$$I_2 = \mathcal{W}_1(\pi_\infty Q_T, \pi) \leq \exp[-G_K]\,(\sqrt{d} + \mathrm{diam}(\mathcal{X}))\exp[-m_\mu T]. \tag{207}$$

Substituting (206) and (207) into (201), we obtain the final result:

$$\mathcal{W}_1(\pi_\infty R_K, \pi) \leq \mathcal{W}_1(\pi_\infty R_K, \pi_\infty Q_T) + \mathcal{W}_1(\pi_\infty Q_T, \pi) \leq D_1 \exp\left[-m_\mu T\right] + D_2 M + D_3 \delta^{1/2}.$$

$\square$

This theorem establishes the main convergence result of the work, providing a comprehensive error analysis for the proposed proximal diffusion sampling algorithm. It demonstrates that the total sampling error decays exponentially with the diffusion horizon $T$, linearly with the score approximation error $M$, and as the square root of the maximum stepsize $\delta$.

