# OpenReview forum: "Proximal-Based Generative Modeling for Bayesian Inverse Problems"
_ICML.cc/2026/Conference — ICML 2026 regular_

### Official Review · Reviewer_LGfL · 2026-03-10

**Soundness:** 2
**Presentation:** 3
**Significance:** 3
**Originality:** 3
**Overall Recommendation:** 5
**Confidence:** 3

**Summary:**

The paper proposes a method for posterior sampling using diffusion models for inverse problems. Instead of relying on the DPS approximation of the intractable likelihood term $p(y | x_t)$ which is computationally costly due to backpropagation through the learned denoiser, it relies on an score approximation targeting the posterior distribution based on Moreau approximation. This expression of this so-called Moreau score involves the computation of a proximal operator of the full potential $U = \beta f_y + g$ (where $f_y$ is the known likelihood and $g$ is the prior), which is approximated through proximal splitting using the proximal operator associated with the prior $g$. Therefore the method requires to train a neural network to approximate this proximal operator. At inference, the sampling of the target posterior distribution follows a certain SDE whose drift depends on the computed approximation, without involving any backpropagation through the denoiser. The paper studies theoretical guarantees of their method under certain simplifying assumptions. Experimental results compares the proposed posterior sampling method to several zero-shot posterior sampling baseline methods on different image restoration tasks on various image datasets, showing a better trade-off between reconstruction quality and inference speed.

**Compliance With Llm Reviewing Policy:**

Affirmed.

**Final Justification:**

The authors addressed the concerns raised during my review.

**Key Questions For Authors:**

I think it is fine that the theoretical claims only hold for the moment on the convexity case, future work could explore how to extend to the nonconvex case. However, this means that the author should clearly discuss whether the assumptions of their theoretical claims hold in Section 5, and if not, this means that the soundness of the proposed method in Section 5 relies entirely on the rigor of the experimental protocol.

1. Could the author detail the protocol of their experiments in Section 5 to make sure that all methods are compared on equal foot?
2. Could the author further support their claims on inference speedup by comparing properly the tradeoff between reconstruction quality and the inference speed for the proposed methods compared to baselines?
3. Could the author discuss the significance of their theoretical claims, regarding the fact that some of their assumptions do not hold in practice?

**Limitations:**

Yes

**Strengths And Weaknesses:**

# Soundness
* *(Strength)* The theoretical claims seem to be sound. I did not check all the proofs in the appendices, but I do not see obvious counter examples for the claims. However, it is important to note that the claims rely on some simplifying assumptions, such as the fact that the prior function $g$ is convex (cf. Assumption 3.3). In practice, in the experiments in Section 5, the prior is a pretrained diffusion model, so it is nonconvex.
* *(Weakness)* To my view, unless if I missed some details, the experimental protocol is not detailed enough to allow one to judge whether the experimental comparison between different methods is sound. It is not clear to me whether all the methods were compared under the same setting for fair comparison. For instance, zero-shot posterior sampling methods typically rely on hyperparameter tuning (for instance the step size parameter in DPS), but I cannot assess whether the methods are tuned properly (on a validation set for instance) in Section 5.
* *(Weakness)* Similarly, when claiming empirical speedup of the method compared to previous baseline, it is necessary to compare the tradeoff between reconstruction quality vs. inference time and identifying Pareto front to conclude whether the method improves compared to baseline, by demonstrating higher speed at fixed reconstruction quality, or higher reconstruction quality at fixed inference time. To my opinion, combining the current Figure 3, Table 2 and 3 are not enough to assess the tradeoffs of the different methods.
* *(Strength)* The authors showed that the proposed method requires less memory usage compared to baselines, effectively illustrating the benefit proposing a score approximation that does not rely on backpropagation through the denoiser like in DPS

# Presentation
* *(Strength)* The paper is globally well structured; theory, method and experimental results are well written.
* *(Weakness)* As discussed above, the presentation can be improved by providing further experimental details to reproduce the results in Section 5.
* *(Weakness)* An explicit discussion about whether the assumptions of the theoretical claims are satisfied in practice should be included, otherwise the reader could think that the theoretical claims hold in the experimental settings of Section 5.

# Significance
* *(Strength)* The paper provides theoretical guarantees on their method.
* *(Strength)*  The paper bridges diffusion posterior sampling with proximal regularization, which to my sense gives an interesting insight, e.g., it offers another way to approximate the score in posterior sampling using Moreau score. Future work could further analyze the impact of such a connection, for instance by continuing clarifying the gap between the exact score and the Moreau score (and the impact in terms of reconstruction quality during posterior sampling) when the assumptions of the theoretical claims in the paper do not hold anymore (e.g., beyond the case where the prior $g$ is convex), whether through a theoretical or an experimental study.
* *(Strength)* It is valuable that Section 5 evaluates the method in a realistic setting with a pretrained diffusion prior.
* *(Weakness)* However, to better highlight the significance of the theoretical results, I think the paper should first include a simple experiment in which the assumptions of the theory hold, for instance a toy example in $\mathbb{R}^d$ with a quadratic prior $g$. This would allow the authors to numerically illustrate the predicted convergence behavior and compare it with theory, which would in turn clarify the practical meaning of the analysis. It would then be interesting to study how the method behaves once these assumptions no longer hold, and to characterize the gap between theory and practice.

# Originality
*(Strength)* To my understanding, the originality of the paper holds on its connection to proximal methods. Regarding this aspect, the authors clearly discuss the connection to existing algorithms in section 3.4 and claim that they are instances of their proximal-based generative modeling framework.

---

> ### Author Rebuttal · Authors · 2026-03-30
>
> We sincerely thank you for your thoughtful and constructive feedback, which allows us to clarify and further improve our work. Below we provide a point-by-point response to each of your comments.
>
> **Soundness of experimental comparison**:
>
> We apologize for the lack of a thorough explanation of our experimental protocol. To ensure relatively fair comparisons, each model uses the optimal parameter selections and checkpoints reported in its respective paper. Following your suggestion, we will detail our experimental protocol and compare with more advanced techniques in Appendix B.
>
> **Performance tradeoff**:
>
> We acknowledge that the reported experimental results on the FFHQ dataset, including memory and sampling usage (Table 1) and quantitative evaluation of samples (Table 2), represent only part of the Pareto front.
>
> Following your suggestion, we will provide the Pareto front to compare the tradeoff between reconstruction quality and inference time in the appendix. In the following table, we first show the least inference cost (Neural Function Evaluations, NFEs) of various methods for a fixed target reconstruction quality (PSNR) for the FFHQ Gaussian deblurring problem.
>
> | Method | Target PSNR 15 | Target PSNR 20 | Target PSNR 25 | Target PSNR 30 |
> |-|-|-|-|-|
> | DPS | 50 | 100 | 800 | - |
> | DiffPIR | 20 | 50 | 100 | - |
> | DAPS | 10 | 20 | 50 | 200 |
> | PGM (Our) | **8** | **10** | **20** | **80** |
>
> The experimental results indicate that, for a fixed PSNR target, the number of NFEs required by PGM is less than that of other methods, and this difference becomes more pronounced as the PSNR increases. This suggests that, compared to existing methods, PGM exhibits a better overall performance tradeoff.
>
> **Discussion on theoretical claims**:
>
> We thank the reviewer for the constructive feedback on our theoretical results. Most recent works on diffusion-based inverse problems assume smooth or Gaussian mixture priors to provide convergence, which may not hold for priors with bounded support. In this paper, we focus on one class of nonsmooth settings, namely convex priors, and provide non-asymptotic convergence analysis.
>
> We also agree that the assumptions used in the theory do not fully hold for the experiments in Section 5. In particular, the theoretical results are established mainly under convexity-type assumptions for analytical tractability, whereas the real-data experiments involve more complex and generally nonconvex settings. We will clarify this point explicitly in Section 5 and add additional experiments to validate our theoretical results. Further, in future work, we will extend the framework and theoretical results to general nonconvex scenarios.
>
> **Additional experiments for validation**:
>
> Thank you for this thoughtful suggestion. We agree that a simple toy example under the theoretical assumptions would help illustrate the practical meaning of our analysis more clearly. In the revision, we will provide a simple experiment in which all the assumptions hold to validate the theoretical results. We consider the optimization problem:
>
> $$
> \min_{x} \quad f(x) = \frac{1}{2}x^\top A x + b^\top x,\quad \text{s.t.} \quad x\in\mathcal{X}\triangleq\lbrace x:\|\|x\|\|_2 \leq r\rbrace,
> $$
>
> where the parameters $A,b,r$ are randomly generated and realizations are generated uniformly from $\mathcal{X}$. This corresponds to sampling from $\pi(x) \propto e^{-(\beta f(x)+\mathbb{I}_{\mathcal{X}}(x))}$. We choose $\mu(t)=1, \lambda(t)=\exp(10t-8)$. Then all the assumptions hold. To illustrate the convergence result, we vary the number of iterations $K$ and evaluate the feasibility rate, i.e., the proportion of samples satisfying the constraint, with 10000 samples. Some results are shown below:
>
> | Metric | $K=0$ | $K=1$ | $K=5$ | $K=10$ | $K=20$ |
> |-|-|-|-|-|-|
> | Feasibility | 0.00% | 7.43% | 97.43% | 99.61% | 99.73% |
>
> Based on the experimental results above, as $K$ increases, the sample distribution rapidly converges towards the constraint set, and the feasibility rate ultimately approaches 1. This is consistent with the convergence result in Theorem 4.3.
>
> Further, we test the influence of the inverse temperature parameter $\beta$ and calculate the optimality gap, i.e., the difference between the sample objective and the true optimum. Some results are shown below:
>
> | Metric | $\beta=0$ | $\beta=0.1$ | $\beta=1$ | $\beta=2$ | $\beta=10$ |
> |-|-|-|-|-|-|
> | Optimality | 0.236546 | 0.209489 | 0.070171 | 0.043825 | 0.016491 |
> | Feasibility | 99.90% | 99.88% | 99.78% | 99.66% | 99.61% |
>
> The above experimental results indicate that as $\beta$ increases, the density of the target distribution $\pi(x)$ concentrates around the optimal solution of the problem, which causes the samples to gradually approach the constraint boundary and converge to the optimizer.

---

> > ### Author Rebuttal · Reviewer_LGfL · 2026-04-02
> >
> > I thank the authors for their clarifications. They addresses my concerns. Therefore I raise my score to "5 Accept".

---

> > > ### Author Response · Authors · 2026-04-03
> > >
> > > Many thanks for acknowledging our rebuttal and for increasing your score. We are glad that our clarifications addressed your concerns. Your time and effort are greatly appreciated.

---

### Official Review · Reviewer_PQ88 · 2026-03-13

**Soundness:** 2
**Presentation:** 3
**Significance:** 2
**Originality:** 2
**Overall Recommendation:** 3
**Confidence:** 4

**Summary:**

This paper aims to address the bottleneck issue in the inverse problem of diffusion models: the time-varying likelihood function. Unlike existing methods such as DPS, it proposes a novel sampling mechanism driven by the Moreau fraction, which can be solved through a closed-form expression obtained via the proximal operator.

**Compliance With Llm Reviewing Policy:**

Affirmed.

**Key Questions For Authors:**

DiffPIR also employs the proximal operator to solve the inverse problem. Could you analyze its relationship with this paper?

**Limitations:**

yes

**Strengths And Weaknesses:**

**Strengths**
1. This article is well-written, with clear descriptions of the methods.
2. To my knowledge, this paper introduces the Moreau score into the field of diffusion inverse problems for the first time.

**Weaknesses**
1. The author primarily employs the DPS method for comparison and asserts that the approach presented in this paper does not require backpropagation through the scoring network. The DPS algorithm is an early-stage method, and numerous more advanced techniques have since emerged, such as DiffPIR[1], DDNM[2], RED-diff[3], and HRDIS[4]. These methods also do not necessitate backpropagation.

2. The benchmark data used in the experiment is too simplistic and lacks datasets such as Imagnet.

3. I notice that the method described in this paper performs observational operations on $x_t$, i.e., the noisy data. Does this imply that the method is difficult to apply to nonlinear inverse problems? Furthermore, the paper does not test such problems.


[1] Denoising Diffusion Models for Plug-and-Play Image Restoration

[2] ZERO-SHOT IMAGE RESTORATION USING DENOISING DIFFUSION NULL-SPACE MODEL

[3] A Variational Perspective on Solving Inverse Problems with Diffusion Models

[4] HYBRID REGULARIZATION IMPROVES DIFFUSIONBASED INVERSE PROBLEM SOLVING

---

> ### Author Rebuttal · Authors · 2026-03-30
>
> We sincerely thank you for your thoughtful and constructive feedback. Below we provide a point-by-point response to each of your comments.
>
> **Comparison with DiffPIR**:
>
> Thank you for the insightful suggestion. Both DiffPIR[1] and our PGM address Bayesian inverse problems with prior $p\propto e^{-g}$ and likelihood $e^{-f}$, and both are related to proximal-type optimization. However, the key difference lies in how the optimization problem is approximated and solved.
>
> From an optimization viewpoint, the target is
> $\min_x f(x)+g(x).$
> [1] adopts a half-quadratic splitting strategy and considers the augmented problem
> $\min_{x,z} f(x)+g(z)+\frac{1}{2\lambda/\mu}||x-z||^2,$
> which leads to the two subproblems
> $$z_k=\arg\min_z \frac{1}{2\lambda/\mu}||z-x_k||^2+g(z),$$
> $$x_{k+1}=\arg\min_x f(x)+\mu\sigma_n^2||x-z_k||^2.$$
> To solve the first subproblem, [1] further uses a first-order approximation $g(z)\approx g(x_k)+\nabla g(x_k)^\top(z-x_k)$, where the Stein score $s_\theta(x)$ is used to approximate $-\nabla g(x)$. This approximation relies on the smoothness of $g$, which does not hold in our nonsmooth setting.
>
> In contrast, PGM does not approximate the prior term $g$. Instead, we linearize the likelihood term $f(x)\approx f(x_k)+\nabla f(x_k)^\top(x-x_k)$, and apply a proximal-gradient step directly to the original problem:
> $$x_{k+1}=\arg\min_x \nabla f(x_k)^\top(x-x_k)+g(x)+\frac{1}{2\mu\sigma_n^2}||x-x_k||^2.$$
> This leads to the proximal splitting in Eqn. (20):
> $$x_{k+1}=\text{Prox}_{g}^{\lambda/\mu}\left(x_k-\frac{1}{2\mu\sigma_n^2}\nabla f(x_k)\right).$$
>
> We then learn a proximal network $\phi_\theta(x,\lambda)$ to approximate $\text{Prox}_g^{\lambda}(x)$, which naturally accommodates nonsmooth priors.
>
> Overall, while both methods are motivated by proximal optimization, [1] approximates and solves a split formulation by smoothing the prior subproblem, whereas PGM applies proximal gradient directly to the original objective and approximates the proximal operator itself. This difference in the approximation and optimization strategies leads to two distinct frameworks, suitable for different problem settings.
>
> **Additional results on ImageNet**:
>
> We acknowledge that DPS is an early-stage method for diffusion-based inverse problems, yet several of its key ideas have been carried over into subsequent work. As a classical baseline, DPS remains widely used for comparison in recent studies.
>
> We highlight that PGM does not require backpropagation to emphasize the sampling efficiency of our single-loop algorithm. In contrast, many recent methods incur additional computational costs to ensure convergence. For example, DDNM[2] is designed for linear inverse problems and requires additional pseudo-inverse calculations; RED-diff[3] and HRDIS[4] rely on refined estimates obtained by optimizing a regularized loss. We numerically validate the sampling efficiency of our method in Section 5 to show that PGM achieves a significant speedup in sampling time.
>
> Following your suggestion, we will add additional experiments on ImageNet and include comparisons with more works. Selected results are listed below:
>
> |ImageNet|SR 4× PSNR|SR 4× LPIPS|Gauss Deblur PSNR|Gauss Deblur LPIPS|
> |-|-|-|-|-|
> |PGM(Ours)|**26.6**|**0.26**|26.6|**0.20**|
> |DPS|21.1|0.36|20.3|0.39|
> |DDNM|23.9|0.47|**28.0**|0.27|
> |DiffPIR|23.1|0.37|22.8|0.35|
> |DAPS|25.8|0.27|26.1|0.25|
>
> Based on the above results, PGM outperforms existing methods, which is consistent with results on other datasets as presented in Section 5.
>
> **Nonlinear inverse problems**:
>
> We apologize for any confusion caused by our problem formulation. Our main task is to sample from posterior $\pi\propto e^{-(f+g)}$, where the likelihood term $f$ is not necessarily required to be in the form of least squares. The only information about $f$ required is its gradient $\nabla f$; therefore, the method is applicable to nonlinear inverse problems as well.
>
> Following your suggestion, we will add additional experiments on nonlinear inverse problems. Selected results are listed below.
>
> |Nonlinear Deblurring|Type|FFHQ PSNR|FFHQ SSIM|FFHQ LPIPS|ImageNet PSNR|ImageNet SSIM|ImageNet LPIPS|
> |-|-|-|-|-|-|-|-|
> |PGM (Ours)||28.4|0.75|0.23|26.3|0.72|0.28|
> |DPS||23.3|0.62|0.27|22.4|0.59|0.30|
> |RED-diff||**30.8**|**0.79**|0.16|**30.0**|**0.75**|0.21|
> |DAPS||28.2|0.78|**0.15**|27.7|0.72|**0.16**|
> |ReSample|Latent|28.2|0.74|0.18|26.2|0.65|0.20|
> |LatentDAPS||28.1|0.71|0.23|25.3|0.61|0.31|
>
> The above results indicate that, although the performance of PGM is superior to that of the classical DPS, it is inferior compared to the latest methods suitable for nonlinear inverse problems. This is caused by the approximation $f(x)\approx f(x_k)+\nabla f(x_k)^\top(x-x_k)$ in operator splitting. As nonlinear inverse problems often suffer from severe nonconvexity, the performance of PGM may degrade. In future work, we will investigate how to extend PGM to general nonconvex problems and explore its convergence results in such settings.

---

> > ### Author Rebuttal · Reviewer_PQ88 · 2026-04-04
> >
> > Thank you for your response. The additional experiments and comments are very helpful. I still have one question: the authors mention that their method can be applied to non-smooth priors, unlike DIFFPIR. Is there any practical significance to this? Which category do the priors commonly used in diffusion models fall into? Additionally, I recall that DIFFPIR also allows for likelihood linearization, as can be seen in their Equation 13.

---

> > > ### Author Response · Authors · 2026-04-06
> > >
> > > We sincerely thank the reviewer for acknowledging our rebuttal. We are glad that our clarifications were helpful. We hope the following responses address your remaining concerns.
> > >
> > > **1. Which category do the priors commonly used in diffusion models fall into?**
> > >
> > > Diffusion-model priors are typically represented implicitly through score networks, i.e., $s_{\theta}(x)\approx \nabla \log p(x)$, and are therefore usually treated as smooth at the model level. It should be emphasized that, denoising score matching requires regularity conditions on the prior and fails to converge to the real score in the nonsmooth case [4].
> > >
> > > To avoid this nonsmoothness, recent works typically resort to early-stopping, i.e., restricting $t\in[t_{min}, T]$ for some $t_{min}>0$. Specifically, the score function $s_{\theta}(x_t,t)$ is strictly evaluated away from the singularity at $t=0$, which leads to an additional truncation error. For instance, references [1–3] explicitly account for this truncation error in their convergence analyses.
> > >
> > > **2. Practical significance for nonsmooth priors.**
> > >
> > > The ability to handle nonsmooth priors is practically meaningful. In many Bayesian inverse problems, useful handcrafted or variational priors, such as sparsity-promoting, TV-type, or composite regularizers, are naturally nonsmooth. This nonsmoothness also appears in priors of natural images or videos, which inherently lie on a low-dimensional manifold.
> > >
> > > In order to demonstrate the practical significance for nonsmooth priors, we ran DiffPIR for a truncated Gaussian sampling task, i.e., $p(x) \propto e^{-(f(x)+\mathbb{I}_X(x))}$ with $f(x) = \frac{1}{2}x^\top A x + b^\top x$ and $X=\lbrace x:\|x\|_2 \leq r\rbrace$. Training data are generated uniformly from $X$. To illustrate the convergence result, we vary the number of iterations $K$ and evaluate the feasibility, i.e., the proportion of samples satisfying the constraint, and convergence, i.e., the $W_1$ distance between sample and target distributions, with 10000 samples.
> > >
> > > |Metric|Method|$K=0$|$K=1$|$K=2$|$K=5$|$K=10$|
> > > |-|-|-|-|-|-|-|
> > > |Feasibility ratio|PGM(Our)|0.00%|99.99%|99.99%|99.97%|99.78%|
> > > ||DiffPIR| 0.00%|48.03%|62.62%|83.02%|89.13%|
> > > |$W_1$ distance|PGM(Our)|2.0242|0.1771|0.0879|0.0750|0.0488|
> > > ||DiffPIR|2.0242|0.3149|0.2545|0.2151|0.1858|
> > >
> > > From the above results, PGM can rapidly concentrate on the prior region and converge to the target distribution while ensuring compliance with the prior. In contrast, DiffPIR may only learn a smoothed prior information, resulting in samples that perform poorly at the prior boundaries and worse feasibility ratio.
> > >
> > > **3. Position of our work.**
> > >
> > > The above discussion clearly demonstrates that the inherent nonsmoothness of the prior can significantly affect both practical performance and theoretical convergence. The truncation error may lead to significant failure in some highly precise application scenarios, such as robot control, protein folding, aeronautics, and astronautics. Therefore, we believe that addressing nonsmooth priors is an important and urgent challenge, and PGM offers a practical solution that not only exhibits comparable performance but also has convergence guarantees.
> > >
> > > Our point is not that all diffusion priors are nonsmooth in practice, but rather that PGM extends naturally to settings where the underlying prior is nonsmooth, while DiffPIR also relies on approximating the prior gradient and thus implicitly requires more smoothness.
> > >
> > > **4. Likelihood linearization of DiffPIR.**
> > >
> > > Thank you for this helpful point. We agree that DiffPIR also allows likelihood linearization, i.e.,
> > > $$
> > > x_{k+1}=x_k+\frac{\lambda}{\mu}s_\theta(x_k)-\frac{1}{2\mu\sigma_n^2} \nabla f(z_k).
> > > $$
> > > In contrast, our PGM iterates as
> > > $$
> > > x_{k+1}=\text{Prox}_{g}^{\lambda/\mu}\left(x_k-\frac{1}{2\mu\sigma_n^2}\nabla f(x_k)\right).
> > > $$
> > > The key difference is therefore not the linearization itself, but the approximation target and optimization framework.
> > >
> > > More specifically, DiffPIR uses likelihood linearization within a half-quadratic splitting scheme, and then approximates the prior subproblem via a gradient linearization of $g$. In contrast, our PGM applies proximal gradient directly to the original objective, linearizes only the likelihood term, and keeps $g$ in an exact proximal form. Thus, DiffPIR uses a prior gradient surrogate $s_{\theta}(x)\approx \nabla \log p(x)$, whereas PGM learns the proximal operator itself, which naturally accommodates nonsmooth priors.
> > >
> > > The corresponding references are listed in order:
> > >
> > > [1] Valentin De Bortoli. "Convergence of denoising diffusion models under the manifold hypothesis." TMLR. 2022.
> > >
> > > [2] Gen Li, and Yuchen Jiao. "Improved convergence rate for diffusion probabilistic models." ICLR. 2024.
> > >
> > > [3] Xingyu Xu, Ziyi Zhang, Yorie Nakahira, Guannan Qu, and Yuejie Chi. "Polynomial convergence of Riemannian diffusion models." arXiv. 2026.
> > >
> > > [4] Pascal Vincent. "A connection between score matching and denoising autoencoders." Neural Computation. 2011.

---

### Official Review · Reviewer_4nzX · 2026-03-17

**Soundness:** 3
**Presentation:** 3
**Significance:** 2
**Originality:** 2
**Overall Recommendation:** 4
**Confidence:** 3

**Summary:**

The paper considers the Bayesian inverse problems where the prior is a diffusion model. The known challenge in the literature is that the time-dependent likelihood score is intractable. They paper propose to replace it with an alternate score based on Moreau score which has a closed-form expression via proximal operators. They show that the Moreau score matching can be trained, and used via proximal diffusion model for posterior sampling. They offer theoretical analysis on their methods.

**Compliance With Llm Reviewing Policy:**

Affirmed.

**Final Justification:**

After rebuttal, the authors clarified their originality, explained their assumptions on the theory, and included new experiments. I have revised my score on soundness and significance as the new results report sampling efficiency and provide clarifications. I have increased my score accordingly.

**Key Questions For Authors:**

See above.

**Limitations:**

See above.

**Strengths And Weaknesses:**

Good points:

- It's important to cite prior work (before the era of modern generative models when discussing inverse problem, as it has a rich literature in signal processing, computational imaging, ...). I appreciate the authors for citing prior works prior to diffusion.

- Their goal is clear and properly stated. The theoretical analysis seems thorough. See my comments below.

- The paper is clear and organized well.


Points to improve:

- Literature on improving the diffusion-based prior inverse problems: Over the past two years, there has been numerous works on how to tackle the intractability of the likelihood portion of the score. The paper lacks proper reference to those literature. The aper only refers to DPS from 2022. I suggest to include and discuss more recent works. I provide three examples from 2023-2025 [1-6] (there are more). Reference to prior work is particularly missing in Section 2.2, to make it clear how this paper differs from prior work; a few of this approaches to projection/regularization, decoupling, or variational to not be depending on the sequential time component.


- Clarity: In the introduction when discussing Moreau score, it helps to give intuition and explain/define what it is. Otherwise the reader would get lost without knowing what it is and why it is being proposed. Moreover, at least from the introduction the connection between Moreau score and proximal-based generative modelling is not clearly explained. Please clarify.

- Missing important info on inverse problems: Please specify clearly that the inverse problems are generally ill-posed and discuss why we even need a prior in the first place. For example, the effective dim of y is lower than x, and that in the absence of prior the solution is not unique (section 2.2 or in the introduction).

- Originality: I wonder if the author can refer to the literature on prior work on usage of Moreau-Yosida in the context of generative modelling. Given that the authors can not cited related work for diffusion-based inverse problems, I worry that there might be related literature to what the authors are proposing but not cited here. Hence, I am not confident about the originality of the paper. Aside from this, the decomposition of the objective we see at (19) is closely related to [1] (and a few other works) decoupling strategy.

- Please compare your approach to DAPS [1] (Algorithm 1).

- Major issue: prior works, in different way, majority of them aims to approximate or decompose p(y |xt) into terms that has y|x0 and xt|x0. Could the author situate their framework with respect to prior work (example [1-7]). While you don't need to evaluate y|xt, you still require the time dependent component on the reverse diffusion for xt|x0.


- Major limitations in proposition 3.2: assumption 1 assume p(x0|xt, y) is symmetric unimodal which for early reverse iterations and complex real data this is very strong restrictive assumption. Otherwise, the Tweedie's formula would be suitable in the approximation of x0|xt for the purpose of computing the likelihood. However, in reality, x0|xt may lie outside of the data manifold. Second, majority of inverse problems are ill-posed and that's why there is a need for prior; hence, the m-strongly convex assumption on f_y is not a proper assumption, which implies unique solution. I can see how under these assumptions their score might be suitable, but under their assumption, the approximation from DPS may itself might be a good approximation for posterior sampling. The strongly convex assumption also appears in Assumption 3.3. Could the authors discuss this in the rebuttal?


- Computational efficiency: The proposed method requires you to train the proximal operator which is expensive. If one have access to clean data, then they might be train the diffusion conditionally with y. Can the authors discuss why one should decide to "train" their score as opposed to the fine-tuning of the conditional score. In reality, for inverse problems we don't have access to x0 ourselves most of the time; if one have access at the same time that they know which inverse problems they are interested in solving, they may then train a conditional score.

- The whole point of diffusion-based inverse problems is to use a "pre-trained prior" which is the unconditional diffusion. Almost all works in this field aims to design/develop methods used during posterior sampling (the reverse SDE). However, this work suggests to train a new score, which brings the question of how if one don't have access to data. It defeat the original purpose to use an already-trained network as prior. Please provide a discussion on this point. Please also discuss the computational cost (how much data, how much resources require to train the proposed score).

On Experiments:

- Experiment section is limited and does not compare to works published in the past two years such as DAPS (Latent-DAPS) [1], DiffStateGrad [2], SITCOM [3], ...

- Experiments (Table 1) mix reporting performance from both latent (PSLD, ReSample) with pixel-based (DPS, DMPS) which is not a fair comparison as they have different backbones. I suggest to separate latent and pixel and compressed models under same condition (similar backbone, ...). For the same reason comparison in Figure 3 is not valid or informative about usefulness of the method.


- Overall, the method is not fully studied in experiments, characterized, or compared to recent literature.

- It's not fully clear the significance of the approach.

[1] Zhang, Bingliang, Wenda Chu, Julius Berner, Chenlin Meng, Anima Anandkumar, and Yang Song. "Improving diffusion inverse problem solving with decoupled noise annealing." CVPR 2025.

[2] Zirvi, Rayhan, Bahareh Tolooshams, and Anima Anandkumar. "Diffusion State-Guided Projected Gradient for Inverse Problems." ICLR 2025.

[3] Alkhouri, Ismail, Shijun Liang, Cheng-Han Huang, Jimmy Dai, Qing Qu, Saiprasad Ravishankar, and Rongrong Wang. "SITCOM: Step-wise Triple-Consistent Diffusion Sampling For Inverse Problems." ICML 2025.

[4] Tolooshams, Bahareh, Aditi Chandrashekar, Rayhan Zirvi, Abbas Mammadov, Jiachen Yao, Chuwei Wang, and Anima Anandkumar. "EquiReg: Equivariance Regularized Diffusion for Inverse Problems." arXiv 2025.

[5] Boys, Benjamin, Mark Girolami, Jakiw Pidstrigach, Sebastian Reich, Alan Mosca, and Omer Deniz Akyildiz. "Tweedie Moment Projected Diffusions for Inverse Problems." TMLR 2024.

[6] Mardani, Morteza, Jiaming Song, Jan Kautz, and Arash Vahdat. "A Variational Perspective on Solving Inverse Problems with Diffusion Models." ICLR 2024.

[7] Song, Jiaming, Arash Vahdat, Morteza Mardani, and Jan Kautz. "Pseudoinverse-guided diffusion models for inverse problems." ICLR 2023.

- There is no broader impacts.


Minor points:

- I suggest to authors not to use the word "novel" for themselves.

---

> ### Author Rebuttal · Authors · 2026-03-30
>
> We sincerely thank the reviewer for the valuable comments and constructive feedback. We hope the following point-by-point responses address your concerns.
>
> **Originality and related work**:
>
> Thank you for your suggestion. We will incorporate the suggested references [1–6] into Section 2.2 and expand the discussion of related work in Appendix A.1.
>
> We agree that Moreau–Yosida regularization has appeared in the broader Bayesian inverse problem literature (e.g., Mou et al. 2022 and Pillai 2024 in References). However, those works typically assume that the prior is explicitly known. In contrast, our setting is fundamentally different: the prior is unknown and only accessible through samples. As also noted by reviewer PQ88, to the best of our knowledge, this is the first work to introduce the Moreau score into diffusion-based inverse problems.
>
> Regarding Eqn. (19), we agree that it is related to the decoupling strategy in DAPS[1]. However, a key difference lies in how the posterior is handled. [1] improves posterior sampling by decoupling adjacent diffusion steps, while it employs (19) together with a Gaussian approximation of $p(x_0|x_t)$ to estimate its posterior mean through sampling. In contrast, our PGM replaces the posterior mean with the posterior mode of $p(x_0\mid x_t)$, which leads naturally to the Moreau score defined in Eqn. (17). Leveraging properties of the proximal operator, we further derive the splitting scheme in Eqn. (20) and Proposition 3.4 to approximate this score. This avoids inner Langevin iterations/backpropagation and enables convergence analysis in nonsmooth settings.
>
> A series of works, including DPS, decomposes the posterior $p(x_t|y)\propto p(x_t)p(y|x_t)$, and then approximates the measurement matching term $p(y|x_t)\approx p(y|\hat{x}_0(x_t))$. Recent works in [2-6] attempt to maintain better measurement consistency with additional projection/optimization/regularization/higher-order information. Compared with [1-6], PGM uses the Moreau approximation $p^\lambda(x_t|y)$ of  $p(x_t|y)$ to replace the inconsistent decomposition $p(x_t)p(y|\hat{x}_0(x_t))$, which admits a proximal-based Moreau score in Eqn. (17). From this perspective, PGM utilizes proximal operator to retract samples towards the prior, thereby maintaining consistency. A detailed comparison will be added in Appendix A.
>
> **Clarity of Moreau score**:
>
> We apologize for the lack of intuition and discussion regarding the Moreau score. Motivated by proximal-based nonsmooth sampling methods, we introduce the Moreau score in Eqn. (17), which is equivalently characterized via the Moreau envelope of the negative log-prior. This provides the key intuition behind our approach and enables nonsmooth priors. In Appendix D, we provide a detailed discussion comparing the Stein score used in diffusion models and Moreau score.
>
> **Prior and assumptions**:
>
> Thank you for pointing out the missing information. We will add a clearer motivation for introducing a prior in Section 2.2. We agree that assumptions in Proposition 3.2 are relatively strong. Their main role is to facilitate the theoretical analysis of the relationship between Moreau score and Stein score, so that the connection can be characterized more explicitly.
>
> Except for the asymptotic equivalence in Proposition 3.2, all other results hold for $m=0$. Specifically, Proposition 4.2 and Theorem 4.3 admit $m=0$ with only influences on the bounded constants. In the proof, the strongly convexity constant used is $\beta m + \mu^2/\sigma^2$, and it does not affect the subsequent proof for $m=0$. Therefore, PGM remains convergent and applicable to some convex but ill-posed problems. In future work, we will focus on extending the theoretical results to nonconvex scenarios.
>
> **Pre-trained prior and training costs**:
>
> Thank you for this important comment. Our method still follows the pre-trained prior paradigm. The proposed Moreau score is learned only from prior samples in an unsupervised manner. Once the proximal operator is learned, it can be used as a pre-trained operator for any sampling processes or further fine-tuning.
>
> We agree that this introduces an additional offline training stage. However, this training is performed only once on prior samples and can be reused across different inverse problems, similar to a pre-trained unconditional diffusion model. We will add more details on the training process in Appendix B.
>
> **Weakness on experiments**:
>
> Following your suggestion, we will add additional experiments and compare with more recent works. Some results are shown below:
>
> |Method|Type|SR 4× PSNR|SR 4× LPIPS|Inpaint 70% PSNR|Inpaint 70% LPIPS|Gauss Deblur PSNR|Gauss Deblur LPIPS|
> |-|-|-|-|-|-|-|-|
> |PGM(Ours)|Pixel|**26.6**|0.26|**31.1**|0.11|26.6|**0.20**|
> |DAPS||25.8|0.27|28.4|0.13|26.1|0.25|
> |DiffStateGrad||26.4|**0.22**|29.7|**0.10**|25.8|0.24|
> |SITCOM||26.3|0.23|29.6|0.12|**27.4**|0.23|
> |ReSample|Latent|22.6|0.37|27.5|0.14|25.9|0.25|
> |LatentDAPS||25.0|0.27|27.5|0.16|25.0|0.34|

---

> > ### Author Rebuttal · Reviewer_4nzX · 2026-04-04
> >
> > I thank the authors for their elaborations. Their originality response has addressed my concern on that aspect.
> >
> > Could the author discuss my earlier comment on the unimodal assumption in Prop 3.2? The assumption is strong, and indeed the source of error when it does not hold for solving inverse problems.
> >
> > Thank you for including more comparison. Could the authors provide more information on the dataset (from the numbers I see they are from ImageNet of prior work)? Are the backbones are similar? DiffStateGrad is a module, so which method they have applied it to? Could you add one more decimal to the table for both PSNR and LPIPS (they are very close) and also add std? Is this random inpainting or box? Possible to add runtime to the table? The reported numbers on the table are taken from the baselines published set of table. Could you confirm this? Given the reported numbers, could you discuss whether there is benefit to the proposed method given runtime, the level of outperformance, and the train time of the proximal operator.

---

> > > ### Author Response · Authors · 2026-04-06
> > >
> > > We sincerely thank the reviewer for acknowledging our rebuttal and for the thoughtful follow-up questions. We are delighted that our initial response addressed your concerns, and we hope the following answers adequately address your additional questions.
> > >
> > > **1. Unimodal assumption in Proposition 3.2.**
> > >
> > > We apologize for not addressing your earlier comment more clearly. We agree that the unimodal assumption on $p(x_0|x_t,y)$ in Proposition 3.2(1) is strong. This assumption is mainly introduced to make the relation between the Moreau score and the Stein score theoretically explicit. Under unimodality, the posterior mean and mode are better aligned (see Appendix D.3); when unimodality fails, the discrepancy between them contributes directly to the sampling error, as quantified in Proposition 3.2(2).
> > >
> > > Importantly, Proposition 3.2(1) is intended mainly to provide intuition. Our main theoretical result, Theorem 4.3, does not rely on this assumption; its proof only uses Assumptions 3.3 and 4.1 (see Appendix G). We will revise the paper to make this distinction clearer.
> > >
> > > **2. Additional experimental details.**
> > >
> > > We greatly appreciate the opportunity to provide more experimental details in the discussion. In the following, we will provide more explanation of the experimental content.
> > >
> > > - **Dataset.** Due to the limited rebuttal period, we were unable to train our model on the full ImageNet dataset. Instead, we used the ImageNet-100 subset [5], consisting of 100 selected ImageNet classes.
> > >
> > > - **Backbone.** All baseline methods [1-3] use the same pre-trained backbone ADM [4] to ensure a fair comparison, which improves the U-Net architecture with time embedding, multi-head attention, and BigGAN residual blocks for upsampling/downsampling. PGM uses a similar U-Net backbone for the proximal network, which adopts $\lambda$ embedding and simpler CNN residual blocks.
> > >
> > > - **DiffStateGrad.** We use DiffStateGrad-DAPS for comparison.
> > >
> > > - **Inpainting setting.** We consider random inpainting. Specifically, each pixel is masked independently with probability 70\%.
> > >
> > > - **More decimals and std.** Following your suggestion, we will report PSNR/LPIPS with more decimals and present results as mean $\pm$ standard deviation over 100 fixed validation images. Some results are shown below, where performances of baseline methods are taken from [1-3]:
> > >
> > > | ImageNet | SR 4× PSNR | SR 4× LPIPS | Inpaint (70%) PSNR | Inpaint (70%) LPIPS | Gauss Deblur PSNR | Gauss Deblur LPIPS |
> > > |-|-|-|-|-|-|-|
> > > | PGM (Ours) | **26.60±2.90** | 0.264±0.060 | **31.14±3.28** | 0.116±0.045 | 26.65±2.93 | **0.206±0.050** |
> > > | DAPS | 25.67±0.73 | 0.256±0.067 | 28.44±0.45 | 0.135±0.052 | 26.12±0.78 | 0.245±0.022 |
> > > | DiffStateGrad-DAPS | 26.40±3.44 | **0.229±0.057** | 29.78±4.17 | **0.107±0.037** | 25.87±3.56 | 0.243±0.075 |
> > > | SITCOM | 26.35±1.21 | 0.232±0.038 | 29.60±0.78 | 0.127±0.039 | **27.40±0.45** | 0.236±0.039 |
> > >
> > > - **Runtime.** We will add results on runtime and NFEs, where all baseline methods and PGM are tested on 4 NVIDIA A800 GPUs:
> > >
> > > | Method | Model type | Model size | Training time | Sampling time | NFEs |
> > > |-|-|-|-|-|-|
> > > | PGM (Ours) | ProxNet (Ours) | 2055MB | 51h32min | **4s** | **100** |
> > > | DAPS | Pre-trained [4] | 2108MB | N/A | 33s | 1000 |
> > > | DiffStateGrad-DAPS | Pre-trained [4] | 2108MB | N/A | 37s | 1000 |
> > > | SITCOM | Pre-trained [4] | 2108MB | N/A | 21s | 600 |
> > >
> > > **3. Benefit of the proposed method.**
> > >
> > > Compared with the baseline methods, the main advantage of PGM is that it achieves substantially lower sampling time and fewer NFEs while remaining competitive in reconstruction quality. Another important benefit is that we establish a convergence theory for proximal-based generative modeling in inverse problems, which is shown in Theorem 4.3.
> > >
> > > We acknowledge that training the proximal operator requires additional computations, but this is a one-time cost. Once the proximal operator is learned, it can be used as a pre-trained network for any optimization/sampling algorithms or further fine-tuning.
> > >
> > > The corresponding references are listed in order:
> > >
> > > [1] Bingliang Zhang , Wenda Chu, Julius Berner, Chenlin Meng, Anima Anandkumar, and Yang Song. "Improving diffusion inverse problem solving with decoupled noise annealing." CVPR 2025.
> > >
> > > [2] Rayhan Zirvi, Bahareh Tolooshams, and Anima Anandkumar. "Diffusion state-guided projected gradient for inverse problems." ICLR 2025.
> > >
> > > [3] Ismail Alkhouri, Shijun Liang, Cheng-Han Huang, Jimmy Dai, Qing Qu, Saiprasad Ravishankar, and Rongrong Wang. "SITCOM: step-wise triple-consistent diffusion sampling for inverse problems." ICML 2025.
> > >
> > > [4] Prafulla Dhariwal, and Alexander Nichol. "Diffusion models beat gans on image synthesis." NeurIPS. 2021.
> > >
> > > [5] Bowen Zhao, Xi Xiao, Guojun Gan, Bin Zhang, and Shu-Tao Xia. "Maintaining discrimination and fairness in class incremental learning." CVPR. 2020.

---

### Decision · Program_Chairs · 2026-04-30

**Decision:**

Accept (regular)

**Comment:**

Reviewers agree that the proposed approach is clear and well-motivated, and poses an interesting connection with more traditional proximal methods / convex analysis. The empirical results are generally strong and provide a potential computational benefit by avoiding backprop through the denoiser.

One significant concern was the positioning of the work in relationship to the recent literature on inverse problems. Reviewers also noted insufficient details for reproducibility of the experiments. In the rebuttal period, reviewers generally appreciated the additional experimental results and details, which alleviated some concerns here. However, the new experimental results discussed in the rebuttal period constitute a significant revision to the paper. While I agree with the reviewers that these new experiment setups and results move the paper in the right direction in terms of relevant baselines and evaluations, these new experiments should be carefully conducted and properly incorporated into the main paper.